# Understanding Warmup-Stable-Decay Learning Rates: A River Valley Loss Landscape View

**Kaiyue Wen**[1]   **Zhiyuan Li**[2]   **Jason Wang**[1]   **David Hall**[1]   **Percy Liang**[1]   **Tengyu Ma**[1]
[1] Stanford University   [2] Toyota Technological Institute at Chicago
kaiyuew@stanford.edu   zhiyuanli@ttic.edu   jsywang@stanford.edu
dlwh@cs.stanford.edu   pliang@cs.stanford.edu   tengyuma@cs.stanford.edu

## Abstract

Training language models currently requires pre-determining a fixed compute budget because the typical cosine learning rate schedule depends on the total number of steps. In contrast, the Warmup-Stable-Decay (*WSD*) schedule uses a constant learning rate to produce a main branch of iterates that can in principle continue indefinitely without a pre-specified compute budget. Then, given any compute budget, one can branch out from the main branch at a proper time with a rapidly decaying learning rate to produce a strong model. Empirically, *WSD* generates an intriguing, non-traditional loss curve: the loss remains elevated during the stable phase but sharply declines during the decay phase. Towards explaining this phenomenon, we conjecture that pretraining loss exhibits a *river valley landscape*, which resembles a deep valley with a river at its bottom. Under this assumption, we show that during the stable phase, the iterate undergoes large oscillations due to the high learning rate, yet it progresses swiftly along the river. During the decay phase, the rapidly dropping learning rate minimizes the iterate's oscillations, moving it closer to the river and revealing true optimization progress. Therefore, the sustained high learning rate phase and fast decaying phase are responsible for progress in the river and the mountain directions, respectively, and are both critical. Our analysis predicts phenomenons consistent with empirical observations and shows that this landscape can naturally emerge from pretraining on a simple bi-gram dataset. Inspired by the theory, we introduce *WSD-S*, a variant of *WSD* that reuses previous checkpoints' decay phases and keeps only one main branch, where we resume from a decayed checkpoint. *WSD-S* empirically outperforms *WSD* and *Cyclic-Cosine* in obtaining multiple pretrained language model checkpoints across various compute budgets in a single run for parameters scaling from 0.1B to 1.2B.

## 1 Introduction

Pre-training large language models (LLMs) typically involves following a learning rate schedule that decreases over a pre-determined number of steps, such as a cosine schedule (Loshchilov & Hutter, 2017; Touvron et al., 2023), where the learning rate starts high and gradually decreases in a smooth curve following the shape of a cosine function. This inflexible approach makes it difficult to adapt to additional compute or data, as the learning rate schedule for all the data is not a natural continuation of the schedule used with past data. Additionally, fitting scaling laws is costly because each compute budget requires a retraining to adjust the learning rate schedule (Hoffmann et al., 2022).

In contrast to the cosine learning rate, recent work Hu et al. (2024) introduces the warmup-stable-decay (*WSD*) schedule, which does not require committing to a pre-specified total compute budget. After a standard warm-up period, the *WSD* schedule maintains a main "branch" using a constant learning rate indefinitely and branches off using a fast-decaying learning rate schedule to obtain intermediate checkpoints (see the second row of Figure 2b). Using the *WSD* schedule, one can continue training from a checkpoint in the main branch by resuming with the same constant learning rate and can obtain training losses for multiple compute budgets with a single run.

Empirically, the *WSD* schedule produces a non-traditional loss curve (see Figure 1): during the constant learning rate phase, the loss remains higher than the loss using other schedules like the cosine schedule; but during the decay phase, it drops sharply, often leading to better final performance compared to the cosine schedule. This raises the main question the paper aims to address:

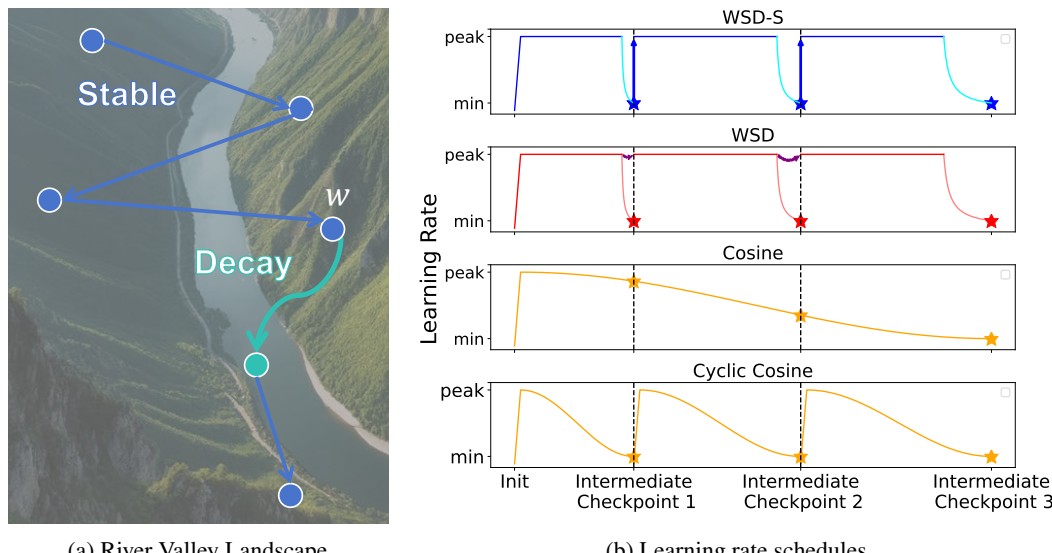

(a) River Valley Landscape        (b) Learning rate schedules

Figure 2: We demonstrate a **river valley** loss landscape in Figure 2a to explain the effectiveness of Warmup-Stable-Decay (*WSD*) schedule(demonstrated ). The stable phase adopts a large learning rate and the iterate will progress along the river while oscillating between the sharp hillsides. Due to the large oscillation caused by the large learning rate, the run will potentially show a higher loss compared to a run using a smaller learning rate in this phase. During the decay phase, the learning rate is dropped rapidly to ease the oscillation of the iterates, driving it closer to the river, revealing the optimization progress. Based on our theory, we propose *WSD-S*implified (*WSD-S*), an effective simplification of the *WSD* schedule in continual learning, where we start directly using a high learning rate from previous intermediate checkpoints. We visualize the learning rate schedule in Figure 2b. The arrow in the second row of Figure 2b indicates *WSD* reinitializes the checkpoint from the last checkpoint of the constant learning rate phase instead.

*Why does WSD work, especially with such a non-traditional loss curve? Specifically, why does a constant learning rate phase, characterized by slow loss improvements, eventually lead to superior performance?*

The first contribution of this paper is a theoretical framework to explain the underlying mechanism of *WSD*. We characterize a type of loss landscape, called the river valley landscape (Definition 3.1), and theoretically show that *WSD* has superior performance on such loss landscapes. We show that the river valley landscape can provide multiple theoretical predictions matching the empirical observations and hence can serve as a useful conceptual picture for understanding the pretraining optimization process.

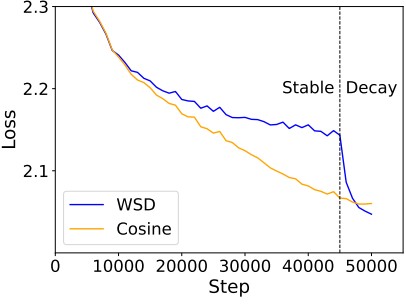

Figure 1: **The Non-traditional Loss Curve produced by *WSD*.** A constant learning rate phase, characterized by slow loss improvements, eventually leads to better validation loss after learning rate decay.

As the name suggests, a river valley landscape intuitively features steeply sloping hillsides with a river winding through the bottom of the gorge (see Figure 2a). During the stochastic gradient-based optimization process, the iterate bounces between the hillsides as it slowly and implicitly progresses along the river direction. The loss in this landscape can be decomposed into two components: the *river component*, which represents the primary loss along the river at the bottom of the hills, and the *hill component*, which accounts for the additional loss caused by deviations in height from the river's course. Progress is determined primarily by the river component in the long run. We demonstrate that when the loss function exhibits this type of landscape, a learning rate schedule should satisfy the following two key properties to effectively minimize the loss.

1. **Sustained high learning rate.** It is advantageous to maintain a large learning rate for as long as possible during training, even at the cost of less reduction in the loss. A large learning rate yields larger bouncing due to the stochasticity of the gradient, increasing the hill component of the loss, but it also makes faster progress in the river direction. In contrast, a small learning rate results in less bouncing, keeping the iterate close to the river, but progress along the river direction is slower. Therefore, a larger learning rate leads to faster fundamental progress in minimizing the river component, which is obscured by the oscillation in the hill component. This progress will be revealed by the decay phase discussed below.

2. **Final low learning rate.** As training nears completion, it becomes essential to reduce the learning rate. This decay minimizes the oscillations in the mountain direction to decrease the hill component and ensures that the iterates converge to a point close to the river, which has a lower loss than any nearby points up the hills.

In Section 3, we provide formal theoretical statements analyzing the trajectories of (stochastic) gradient descent on the river valley landscape, fleshing out the intuitions above. Among our synthetic and real-world studies supporting the river valley landscape hypothesis, an intriguing observation in language model pretraining is that the loss on the linear interpolation of two checkpoints in the stable phase exhibits a convex and unimodal shape, resembling a valley, whereas between two checkpoints in the decay phase, the loss shows a smooth monotone decay.

All the theoretical results above assume a river valley landscape. How likely does the next-token prediction loss follow this pattern, and why? We hypothesize the river valley landscape can naturally arise from the heterogeneity in the stochasticity of different tokens: highly deterministic tokens (which often involve facts and knowledge) contribute to the "river" direction, while uncertain tokens (which often involve flexibility and ambiguity in the language) create the steep hillsides. We demonstrate this insight by showing in Section 4 that under a bigram toy model, indeed the loss has a river valley landscape, and empirically, most properties of the loss curves under various learning rates on the real datasets are still seen in this toy model. We further show that the stable learning rate phase learns the deterministic tokens, whereas the decay phase learns better the stochastic tokens.

Finally, motivated by the theoretical insights, we propose a simplification and improved version of *WSD*, called **WSD-S** in continual learning. In *WSD*, after obtaining an intermediate checkpoint, the model and optimizer are rolled back to the end of the stable phase before continuing with a constant learning rate. However, our theory predicts that the decay phase also makes progress along the river direction and thus there is no reason to discard that part of the progress. Concretely, *WSD-S* immediately continues training from the intermediate checkpoint with a high constant learning rate, instead of rolling the model back to a checkpoint before decaying.

We evaluate the effectiveness of *WSD-S* with extensive experiments on LLMs from 0.1B to 1.2B parameters in a continual learning setting with 50B, 100B, and 200B tokens as the three target compute budgets. We empirically show that *WSD-S* has performance comparable with independent oracle runs with cosine learning rate schedules optimally tuned for each of the three budgets. Furthermore, *WSD-S* leads to a better validation loss than *WSD* under the same compute budgets due to the re-use of the decay period. We also show through ablation studies that the performance is relatively insensitive to the precise fraction of time spent decaying as long as it is near $10\%$ and the decay does not start shortly after a coincidental loss spike.

## 2 RELATED WORK

We discuss related work in two main areas: learning rate schedules and theoretical understandings of the loss landscape. We defer the detailed discussion to Appendix A.

**Learning Rate Schedules.** Prior research has explored various choices of learning rate schedules (Smith, 2017; Loshchilov & Hutter, 2017). Recent studies have focused on optimizing these schedules for language model pretraining (Hu et al., 2024; Raffel et al., 2023; Defazio et al., 2023).

**Theoretical Understanding on Loss Landscape.** A substantial body of research seeks to elucidate the properties of the loss landscape in deep learning. The closest ones related to our work include the impact of gradient noise and curvature (Zhang et al., 2020a; Pan & Li, 2023), the benefits of large learning rates for finding flatter minima (Kong & Tao, 2020; Wang et al., 2022), and the interplay between loss landscape geometry and feature learning dynamics (Nakkiran et al., 2019; Rosenfeld & Risteski, 2023). Among these works, Xing et al. (2018) has presented a similar conceptual picture with us, arguing that SGD locally bounces around the valley on top of a *valley floor*. The iterates will explore the uneven valley floor to find a more generalizable solution. In contrast, we focus on the optimization perspective and assume the existence of the *river* at the bottom of the hillsides, where the loss monotonously decreases. We build a formal theoretical framework on top of this picture, leading to multiple quantifiable theoretical predictions.

## 3 THEORETICAL ANALYSIS WITH RIVER VALLEY LOSS LANDSCAPES

### 3.1 SETTING AND ASSUMPTIONS

We prove in the theorem below that the gradient flow starting from $w$ will eventually converge near the river and remain close to it. Subsequently, if we project the iterate $w(T)$ onto the river,

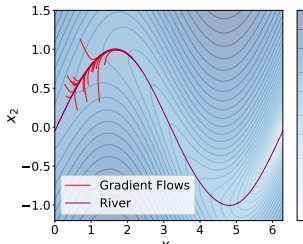 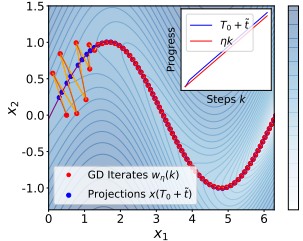 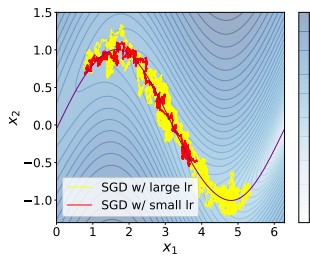

(a) Gradient Flow Dynamics  (b) Gradient Descent Dynamics  (c) Stochastic Gradient Descent

Figure 3: **Validation of Theory on a 2D Function.** We validate Theorems 3.2 to 3.4 using a 2D example with the loss function $L(x_1, x_2) = (x_2 - \sin(x_1))^2 + 0.2|10 - x_1|$. The blue curve represents the "river", where the gradient aligns with the minimal eigenvector of the Hessian. (1) On the left, we observe that multiple randomly initialized gradient flows converge near the river and follow it closely thereafter, consistent with Theorem 3.2. (2) In the middle, we show that discrete gradient descent with a learning rate $\eta = 0.6$ shows similar behavior: after initial oscillations, the gradient descent iterates align closely with their projections on the river. The inset illustrates that the $t$-th projection's progress along the reference flow (eq. (1)) approximately equals $\eta t$, as predicted by Theorem 3.3. (3) On the right, we further illustrate that stochastic gradient descent (SGD) also tracks the river. In contrast to the discrete-step gradient descent, the iterates oscillate around the river rather than staying on it. The trajectory with a larger learning rate exhibits both faster progress and greater oscillations compared to the trajectory with a smaller learning rate, as predicted by Theorem 3.4.

the projection will move along the river at a pace similar to the reference flow $x(\cdot)$ (eq. (1)). This phenomenon is visualized on Figure 3a. We will now formally present our theory. We use $w \in \mathbb{R}^d$ to denote the parameters and $L$ to denote the loss. Further, we use $\lambda_k(H)$ and $v_k(H)$ to denote the $k$-th largest eigenvalue and eigenvector of a matrix $H$, respectively. The "river" in the river valley is a 1-dimensional manifold $\mathcal{M}$ formalized below.

**Assumption 1.** *We assume the existence of a "river", which is a 1-dimensional manifold $\mathcal{M}$ such that any point $w \in \mathcal{M}$ has a gradient $\nabla L(w)$ that is in the same direction as the minimal eigenvector direction of the Hessian, $v_d(\nabla^2 L(w))$.*

Under this assumption, at every point on the river, the gradient $\nabla L(w)$ will align with the locally flattest direction, $v_d(\nabla^2 L(w))$, which we refer to as the *river direction*. All other directions orthogonal to the river direction are considered as the *mountain directions*, corresponding to the steep hillsides in our conceptual picture.

We will consider a neighborhood $U$ of the river $\mathcal{M}$ with the following technical assumptions.

**Assumption 2** (Regularity Assumption). *There exists an open set $U$ containing $\mathcal{M}$ satisfying the following assumptions:*

1. *Analyticity. $L(w)$ is analytic with respect to $w$.*
2. *Bounded Hessian. There exists a constant $\gamma_{\max} > 0$, such that $\forall w \in U, \|\nabla^2 L(w)\|_{\mathrm{op}} \leq \gamma_{\max}$.*
3. *Existence of Eigengap. There exist constants $\gamma_{\mathrm{flat}}, \gamma > 0$, such that $\forall w \in U, \lambda_{d-1}(\nabla^2 L(w)) > \gamma + 4\gamma_{\mathrm{flat}}, |\lambda_d(\nabla^2 L(w))| < \gamma_{\mathrm{flat}}$.*
4. *Slow Spinning of $v_d$. There exist constants $\Delta > \Delta_{\min} > 0, \kappa \in [0, 0.01)$, such that $\forall w \in U, \Delta_{\min} < \|\nabla L(w)\|_2 \leq \Delta$, and $\|\nabla v_d(\nabla^2 L(w))\|_{\mathrm{op}} \leq \kappa\gamma/(2\Delta)$. This means that the river direction $v_d$ changes slowly during optimization.*
5. *Uniqueness of $\mathcal{M}$. For any point $w \in U - \mathcal{M}$, the gradient $\nabla L(w)$ is not parallel to $v_d(\nabla^2 L(w))$.*
6. *Conservation of Gradient Flows. There exists an open subset $V \subset U$ and a constant $r > \frac{10\Delta}{\gamma}$ for $\gamma$ defined in Assumption 2.3 such that $\forall w \in V$, the $r$-neighborhood of the gradient flow starting from $w$ stays in $U$ for continuous time $T_{\max} \geq 10\log(2\Delta/(\kappa\Delta_{\min}))/\gamma$.*

Throughout the analysis, $\kappa$ should be treated as a *small* dimensionless constant, indicating the river spins slowly.

**Definition 3.1** (River Valley Landscape). *If a loss function $L$ satisfies Assumptions 1 and 2, then we will claim that the loss function is a river valley.*

One simple example of a river valley landscape is the quadratic loss $L(x_1, x_2) = \frac{\gamma x_1^2}{2} - x_2$ with $\kappa$ equals to 0. In this case, the river is simply the line $x_2 = 0$. However, the river valley landscape can also be more complex and non-convex, see Figure 3 for an illustration. We will prove that the iterates will follow the river with a predictable pace, which is characterized by the reference flow.

**Reference Flow.** We introduce a Riemannian gradient flow constrained to the river $\mathcal{M}$, serving as a reference in the following theorems. This flow intuitively represents the dynamics of iterates during a gradient flow on the loss constrained by the river. We will denote the projection to the tangent space of the river as $P_{\mathcal{M}}(w)$ for $w \in \mathcal{M}$ and choose an arbitrary starting point $x_0$ on the river. The reference flow is defined as

$$dx(T) = -P_{\mathcal{M}}\left(x\left(T\right)\right) \nabla L(x(T)) dT, x(0) = x_0. \tag{1}$$

Here, we use $x$ to represent a point on the river, distinguishing it from $w$, which denotes a weight in the original space. $T$ refers to the continuous time variable.

## 3.2 MAIN RESULTS

**Gradient Flow Dynamics.** We will now consider gradient flow in the river valley landscape starting from a point $w \in V$, with $V$ defined in Assumption 2.6:

$$dw(T) = -\nabla L(w(T)) dT, w(0) = w \in V. \tag{2}$$

**Theorem 3.2.** *If a loss $L$ is a river valley (Definition 3.1), for the gradient flow $w(T)$ defined in Equation* (2)*, the iterate will obey the following dynamics:*

1. *Converge to a neighborhood of the river after a constant time $T_{\text{converge}} = 2 \log(2\Delta/(\kappa \Delta_{\min}))/\gamma$.*

   $$\text{dist}(w(T_{\text{converge}}), \mathcal{M}) = \min_T \|x(T) - w(T_{\text{converge}})\|_2 \leq 2\kappa\Delta/\gamma.$$

2. *Track the river closely with the same pace as the reference flow. There exists a time shift $T_0$ depending on $w(T_{\text{converge}})$, such that for any $T \in [T_{\text{converge}}, T_{\max}]$ for $T_{\max}$ defined in Assumption 2.6, there exists a $\tilde{T} \in [(1-\epsilon)T, (1+\epsilon)T]$ satisfying that, $\|x(T_0 + \tilde{T}) - w(T)\|_2 \leq 2\kappa\Delta/\gamma$, for $\epsilon = 30\kappa$.*

The proof is deferred to Appendix C.5. In this theorem, the lower bound on $T$ represents the time required for the iterate to converge near the river. Here $x(T_0 + \tilde{T})$ can be viewed as a projection of $w(T)$ onto the river. As both the geometric error ($2\kappa\Delta/\gamma$) and the time-alignment error ($\epsilon$) vanish when $\kappa$ is small, this projection is not only close to $w(T)$ but also moves at nearly the same rate as the reference flow. Here the term $T_0$ acts as a shift, reflecting the dependence on the initialization, as optimization trajectories starting from different initial points will enter the river at distinct locations. The term $\tilde{T}$ represents the progress made along the river, consistent in the subsequent sections.

**Gradient Descent Dynamics** We will proceed to gradient descent with a discrete learning rate. Similar to the continuous case, an iterate far from the river will converge to the river (as visualized in the first few steps of Figure 3b). To ease our analysis, we will skip the convergence analysis and assume the starting point $w$ lies on the course of the river.

$$w_\eta(k+1) - w_\eta(t) = -\eta \nabla L(w_\eta(t)), w_\eta(0) = w \in \mathcal{M}. \tag{3}$$

Here we use $t$ to denote the discrete time step, in contrast to the continuous time variable $T$ used in the previous section. In this case, the progress along the reference flow over $t$ steps will be approximately $\eta t$, as shown in the following theorem.

**Theorem 3.3.** *If a loss $L$ is a river valley (Definition 3.1), when $\eta < \frac{\gamma}{2\gamma_{\max}^2}$, for the gradient descent $w_\eta(T)$ defined in Equation* (2) *with initialization $w$ on the river, there exists a time shift $T_0$ depending on $w$ and $\eta$, satisfying that for any $t \leq T_{\max}/\eta$, there exists a $\tilde{T} \in [(1-\epsilon)\eta t, (1+\epsilon)\eta t]$ satisfying that, $\|x(T_0 + \tilde{T}) - w_\eta(t)\|_2 \leq 10\kappa\Delta/\gamma$ for $\epsilon = 30\kappa + 4\eta\gamma_{\text{flat}}$.*

The proof is deferred to Appendix C.6. We observe that the distance of the iterates from the river remains on the same order as in Theorem 3.2. Finally, the Theorem 3.3 predicts that a larger learning rate $\eta$ will induce higher progress $\eta t$ down the river given the same number of steps $t$, which is verified in the inset of Figure 3b.

**Stochastic Gradient Descent Dynamics.** The above analysis holds for deterministic dynamics and we will now proceed to model the stochasticity in the optimization process. This stochasticity will stop the iterate from fully converging to the river and lead to oscillation in the mountain direction. To simplify the analysis, we will consider a special case where the river direction is a constant and the river reduces to a straight line.

**Assumption 3** (Straight River). *For $U$ in Assumption 2, $\forall w \in U, \|\nabla v_d(\nabla^2 L(w))\|_2 = 0$. In this case, the river is a straight line parallel to the direction of $v_d(\nabla^2 L(w))$.*

Under Assumption 3, $v_d(\nabla^2 L(w))$ is a constant vector for $w \in U$ and we will use $v_d$ to denote this vector. We will also assume that the update is deterministic in the direction of the river, which simplifies our proof while still capturing the essential dynamics of SGD. Consequently, we can express the SGD update as follows:

$$\tilde{w}(k+1) = \tilde{w}(t) - \eta_k \nabla L(\tilde{w}(t)) + \eta_k g_k, g_k \sim \mathcal{N}\left(0, \sigma^2\left(\mathcal{I}_d - v_d v_d^T\right)\right), \tilde{w}(0) = w \in \mathcal{M}. \quad (4)$$

Here $\mathcal{N}(\mu, \Sigma)$ indicates the normal distribution with mean $\mu$ and covariance $\Sigma$. Compared to deterministic gradient descent, the introduced noise $g_k$ causes the iterates to deviate from the river instead of fully converging to it (see the difference between Figure 3b and Figure 3c). Consequently, we need to impose additional assumptions (deferred to Assumption 5 in appendix) on the loss.

**Stable Phase.** We start with the stable phase, where the learning rate $\eta_k = \eta$ remains constant. This theorem provides a formal basis for decomposing the loss into its river and hill components.

**Theorem 3.4.** *Suppose a loss $L$ is a river valley (Definition 3.1) and satisfies Assumptions 3 and 5. Then, for any constants $\delta \in (0, 1)$ and $T \leq T_{\max}$, for sufficiently small learning rate $\eta$ depending on the regularity constants, (Deferred to Assumption 7 in Appendix) the SGD iterates (defined in Equation (4)) with $\eta_k = \eta$ satisfies that for any integer $t \in [1/\eta\gamma, T/\eta]$, there exists a $\tilde{T} \in [(1-\epsilon_t)\eta t, (1+\epsilon_t)\eta t]$ satisfying that, $\mathbb{E}[L(\tilde{w}(t))] - L(x(\tilde{T})) = (d-1)\eta\sigma^2/2 + \epsilon_L$ where $\epsilon_t = 4\eta\gamma_{\text{flat}}$ and $|\epsilon_L| \ll (d-1)\eta\sigma^2$ (defined in Appendix C.7).*

The proof is deferred to Appendix C.7. In Theorem 3.4, the error term in the approximation of the pace of the projection remains the same as in the deterministic case (Theorem 3.3). However, the stochasticity introduces an additional hill component $(d-1)\eta\sigma^2/2$ to the expected loss at the iterate. The hill component increases linearly with the learning rate. We conjecture that the theorem can be extended to a general setting and verify this conjecture on a toy loss (see Figure 3c).

**Decay Phase.** Finally, we will consider the decay phase in training and will show that a proper decaying schedule can reduce the hill component of the loss rapidly. We will first define our decaying schedule, starting from step $t_s = \lceil T/\eta \rceil$: $\eta_k = \frac{\eta}{2+(t-t_s)\eta\gamma}, t_s \leq t \leq 1.1t_s$. We choose this schedule to maximize the loss decrease rate on a quadratic function (see Appendix C.2) because we perform quadratic approximations of the loss near the river in our analysis. Our theorem predicts that the hill component of the loss will decrease linearly with the learning rate under this learning rate schedule, consistent with the empirical findings in Hu et al. (2024).

**Theorem 3.5.** *Under the setting of Theorem 3.4, the SGD iterates (defined in Equation (4)) with the decaying learning rate schedule satisfies that for any integer $t \in [t_s, 1.1t_s]$, there exists a $\tilde{T} \in [(1-\epsilon_t)T(t), (1+\epsilon_t)T(t)]$ satisfying that, $\mathbb{E}[L(\tilde{w}(t))] - L(x(\tilde{T})) \leq (d-1)\eta_k\sigma^2/2 + \epsilon_L$ with $T(t) = T + \sum_{k=t_s}^{t} \eta_k$.*

The formal proof is deferred to Appendix C.8. Compared with Theorem C.32, the hill component is now dominated by $(d-1)\eta_k\sigma^2/2$, scaling linearly with the decaying learning rate. When the oscillation level $\sigma$ is large compared to the loss changes along the river, the loss decrease can then appear faster in the decay phase than in stable phases (see Figure 4). Further, the decaying phase also makes progress along the river, which corresponds to the term $\sum_{k=t_s}^{t} \eta_k$ in the theorem. Finally, the terms used in our theorem match the scaling law formulation in the concurrent work (Tissue et al., 2024).

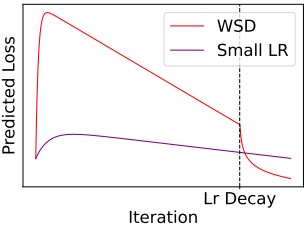

Figure 4: Predicted Loss Curve of SGD By Theorems 3.4 and 3.5 on Loss $L(x_1, x_2) = \gamma x_2^2/2 - x_1$.

### 3.3 Visualizing the River Valley.

We use a direct probing method to verify our theory. Our theory suggests that when the learning rate is large, the model will bounce back and forth between the sharp valleys. However, in the decay phase, the model will move downwards the hillside to approach the river. This suggests that if we connect two checkpoints in the stable phase, we should expect to see a projection of the valley, and if we connect two checkpoints in the decay phase, we should expect to see smooth decreasing curves.

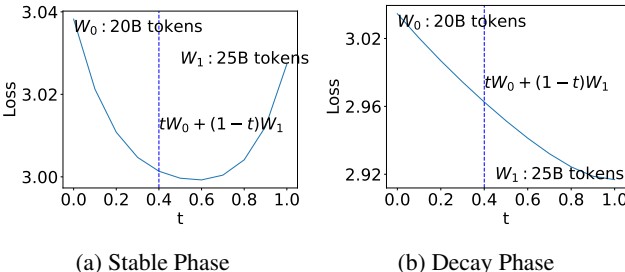

(a) Stable Phase     (b) Decay Phase

Figure 5: **Probing Loss Landscape.** We validate the river valley analogy by interpolating stable and decay phases in GPT-2 pretraining experiments. We observe that loss resembles a valley when constrained on the segment connecting two models during the stable phase and smoothly decreases when connecting two models during the decay phase.

To verify this, we pretrain a 124M GPT-2 model on OpenWebText. In the first run, we train the model with a constant learning rate for 25B tokens and interpolate between two checkpoints at 20B and 25B tokens (Figure 5a). In the second run, we branch off from the first run at 20B tokens and decay the learning rate for 5B tokens, and we interpolate between two checkpoints at 20B and 25B tokens (Figure 5b). The interpolation results closely resemble our theory. This observation is also consistent with Sanyal et al. (2023) which shows weight averaging improves model performance in the earlier part of the cosine training runs, where the learning rates are higher. Additionally, the smooth decreasing curves we observed when connecting two checkpoints in the decay phase are consistent with the findings in Hägele et al. (2024).

## 4 Uncertainty Variation in Data Distribution Shapes the River Valley Landscape

What causes the loss landscape to resemble a river valley structure? In this section, we propose and validate the hypothesis that variations in next-token uncertainty shape the loss landscape. When predicting a deterministic fact, a large learning rate can boost the model's confidence, accelerating learning. However, when the next token is inherently ambiguous—such as the continuation of a phrase like "I am"—the model must learn a calibrated distribution, which may necessitate a smaller step size. This variation in uncertainty leads to differences in sharpness across the loss landscape, resulting in the river valley structure.

**A Toy Bigram Language.** We formalize this intuition using a synthetic language composed of cities and names, where each city corresponds to a unique distribution of its citizens' names. For instance, one city might have a highly deterministic distribution, with most residents named "Ken", while another city may have a more diverse distribution of names. This synthetic language follows the structure in Allen-Zhu & Li (2024). The goal is to learn the distribution of names conditioned on each city. We show that cities with more deterministic name distributions align with flatter regions in the loss landscape (the "river"). In contrast, cities with more diverse name distributions correspond to sharper regions (the "hillsides").

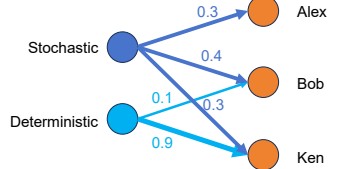

Figure 6: **Visualization of Toy Bigram Language.** We design a synthetic dataset where each city has a unique name distribution. The left shows the name distributions for two cities, one deterministic and one stochastic.

Formally, let the set of cities be represented by $\{1, \ldots, n\}$ and the set of names by $\{1, \ldots, m\}$. Data is generated by first selecting a city $i$ uniformly at random, then sampling a name $j$ according to the city's name distribution. The *name distribution* for city $i$ is parameterized by a categorical distribution $\text{Categorical}([P_{i,j}]_{j=1}^m)$, where $P_{i,j}$ represents the probability of selecting name $j$ in city $i$, and each $P_{i,j} > 0$. To quantify the uncertainty in each city's name distribution, we compute the Gini impurity of the distribution as: $U_i = \text{I}_{\text{G}}(\text{name} \mid \text{city} = i) = 1 - \sum_{j=1}^m P_{i,j}^2 \in \left[0, 1 - \frac{1}{m}\right).$

The value of $U_i$ reflects the uncertainty of city $i$'s name distribution. When the distribution is close to deterministic—i.e., there exists a $j$ such that $P_{i,j}$ is near 1—$U_i$ approaches its lower bound of 0. Conversely, for a nearly uniform distribution, $U_i$ approaches its upper bound of $1 - \frac{1}{m}$. Given this setup, we parameterize our model with $\Theta \in \mathbb{R}^{n \times m}$, where each row corresponds to a city and each column to a name. The model estimates the probability of name $j$ for city $i$ using the softmax function $\frac{\exp(\Theta_{i,j})}{\sum_{k=1}^m \exp(\Theta_{i,k})}$. We use sampled data to train this model with cross entropy loss. The population loss is given by: $L(\Theta) = \frac{1}{n} \sum_{i=1}^n \ell_i(\Theta_{i,:}), \ell_i(\Theta_{i,:}) = -\sum_{j=1}^m \mathcal{P}_{i,j} \log \frac{\exp(\Theta_{i,j})}{\sum_{k=1}^m \exp(\Theta_{i,k})}$. This loss is separable across different cities, meaning that the contribution from each city is independent. The loss component $\ell_i(\Theta_{i,:})$ captures the contribution from city $i$, and different name distributions across cities lead to different forms of $\ell_i$. Considering a parameter $\Theta^*$ that minimizes the loss $L$, we will

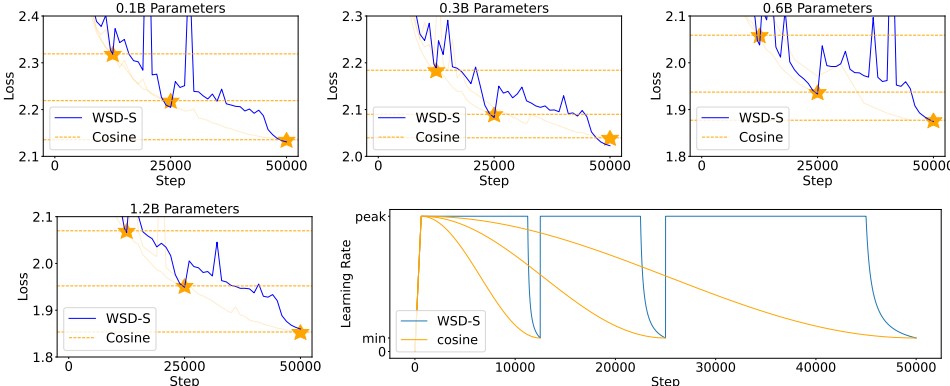

Figure 8: **Comparison with the Cosine Oracles.** We show that the *WSD-S* schedule can perform similarly to the Cosine schedules in a single run. The ⋆ in the graphs visualize the terminating validation loss of different Cosine runs. The largest validation loss gap between the *WSD-S* and the Cosine schedules is 6e-3. The lower right figure plots the learning rate curves used in this experiment.

show that cities with more stochastic name distributions correspond to sharper components in the loss landscape, as reflected by the average-direction sharpness of $\ell_i$.

**Lemma 4.1.** *The average-direction sharpness of loss component $\ell_i$ at $\Theta^*$ equals the uncertainty of the name distribution ($U_i$).* $\mathrm{Tr}(\nabla^2 \ell_i(\theta))\,|_{\theta=\Theta^*_{i,:}} = U_i$.

Lemma 4.1 demonstrates that at the global minimum, the sharpness associated with a city decreases as the city's name distribution becomes more deterministic. This aligns with the intuition that a deterministic token corresponds to a flatter loss direction. We can further establish the existence of a generalized river (Assumption 9) in this loss landscape under appropriate assumptions about $\mathcal{P}$ (see Theorem C.35). Along the river, the gradient remains nonzero only for the cities with more deterministic name distributions, reinforcing the connection between determinism and flatness in the loss landscape.

**Empirical Verification.** We empirically verify that the loss curve of *WSD* can be reproduced in our synthetic setting. The dataset used contains two types of cities: (1) a deterministic type with name distribution's entropy less than 0.2, and (2) a stochastic type with name distribution's entropy greater than 1. Each type contains 1.8k cities and there are 10 possible names. We train the toy model defined previously on this synthetic data and replicate the non-traditional loss curve of *WSD* (Figure 7).

We continue to show that the difference in uncertainty also shapes the loss landscape for Transformers. We convert the data into a synthetic language in the format "The resident of [CITY]: [NAME]" and fine-tune a 0.1B GPT-2 model, pretrained on OpenWebText, using this synthetic data. We experiment with two different learning rate schedules: a constant schedule (stable) and a decaying schedule (decay). We then calculate the difference in loss between the two models' predictions for the first token of "[NAME]". A significant Spearman correlation of 0.388 is observed between the loss difference and the ground truth entropy per city. This correlation indicates that the loss decrease is greater during the decay phase for more stochastic populations. Furthermore, although the decay phase achieves a lower overall loss, the mean loss for the deterministic sub-population is higher than in the stable run, suggesting that the stable run better learns the deterministic sub-population.

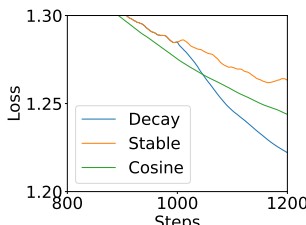

Figure 7: **Reproducing the Non-traditional Loss Curve.** We reproduce the non-traditional loss curve of *WSD* on this synthetic language.

## 5   *WSD*-S: A Simplification of the *WSD* Schedule

The goal of continual pretraining is to generate checkpoints that exhibit good performance at multiple compute budgets in one run. Formally, our goal is to achieve multiple intermediate checkpoints $\theta_{T_k}$, each corresponding to a computing budget (number of steps) $T_k$ for $k \in \{1, \ldots, K\}$.

A strong baseline to measure the performance of $\theta_{T_k}$ would be running cosine learning rates (Figure 8, lower right) for each budget $T_k$ separately, decay the learning rate linearly to the cosine function between $[0, \pi]$. We will dub this *oracle* method as **Cosine-Oracle**. However, *Cosine-Oracle* can't be done in a single run and will incur a high total compute budget $\sum_k T_k$. A simple modification to *Cosine-Oracle* is to use multiple consecutive cosine learning rates between $T_{k-1}$ and $T_k$ (Figure 2b,

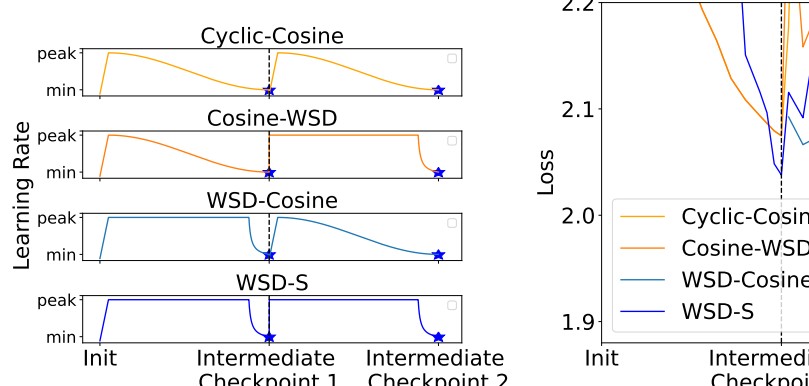

Figure 9: **Cosine Learning Rate Implicitly Hurts the Models for Future Continual Learning.** We show that while *WSD* and the cosine learning rate schedule may produce similar validation loss in a single run, a model trained with the cosine learning rate schedule is implicitly hurt compared to the model trained with *WSD* for future Continual learning. On the 0.6B models, after training the models for 50B tokens using both *WSD-S* and the cosine learning rate schedule, we continually train two models for another 50B tokens using both learning rates. We observe that the model trained with *WSD-S* consistently outperforms the model trained with the cosine learning rate when used as the starting point for further training.

last row), which we will dub as ***Cyclic-Cosine***. *Cyclic-Cosine* only requires a total compute budgets $T_K$ but it leads to non-negligible performance loss compared to *Cosine-Oracle* (Hu et al. (2024)).

Warmup-Stable-Decay (***WSD***) addresses this issue by maintaining a main branch that keeps using a constant learning rate after warmup process and branch off using a decaying learning rate to achieve intermediate checkpoints. One can then continue pretraining from a checkpoint in the main branch by resuming with the same constant learning rate. Formally, *WSD* introduces decay starting points $D_1, \ldots, D_k$ such that $T_{i-1} < D_i < T_i$. *WSD* will then correspond to the following process (Figure 2b, second row):(1) Get a main branch of checkpoints $\theta^{\mathrm{main}}$ by running a constant learning rate schedule for $D_K$ steps, and (2) For each $k$, run a decaying learning rate schedule for $T_k - D_k$ steps starting from $\theta^{\mathrm{main}}_{D_k}$ to get $\theta_{T_k}$. The above process reutilizes the main branch of checkpoints $\theta^{\mathrm{main}}$ for each $T_k$ and hence reduces the total compute budget to $T_K + \sum_k (T_k - D_k)$.

Recall that in the river valley landscape model, the Warmup-Stable-Decay (*WSD*) algorithm can be viewed as a combination of a large learning rate phase to speed up progress down the river and a rapid learning rate drop at the end to reduce the oscillation. Because the decay phase also makes progress along the river (see Theorem 3.5), we propose a simplified version of *WSD*, called Warmup-Stable-Decay-Simplified (***WSD-S***), that continues with another stable phase leaving off the end of the previous decay phase (see the first row of Figure 2b) without separating the training process into two branches. Formally, the *WSD-S* learning rate schedule is defined as follows:

$$\eta_k = \begin{cases} \mathrm{decay}(T_i - D_i, \eta_{\max}, \eta_{\min})[t - D_i] & \text{if } \exists i, D_i < t \le T_i; \\ \eta_{\max} & \text{otherwise.} \end{cases} \tag{5}$$

The key difference from our methods is the choice of initialization point when retraining starts. In *WSD*, the second stable phase uses the model before the decay phase, whereas we use the model after it. This process is more convenient to implement because it does not require rolling back to the main branch after each decay phase. Here the learning rate decay function $\mathrm{decay}$ can take many forms that decay the learning rate from $\eta_{\max}$ to $\eta_{\min}$ over $T_i - D_i$ steps. In this paper, we will use the following decay function $\frac{1}{\mathrm{decay}(T, \eta_{\min}, \eta_{\max})} = \left[ \frac{t}{T} \frac{1}{\eta_{\min}} + \left(1 - \frac{t}{T}\right) \frac{1}{\eta_{\max}} \,\middle|\, t \in \{0, 1, \ldots, T\} \right]$. for all experiments (visualized in Figure 2b, first two rows). This function is motivated by the analysis on quadratic functions in Theorem 3.5. The inverse of the learning rate linearly interpolates from the inverse of the maximum to the inverse of the minimum.

### 5.1 EXPERIMENTS

**Architecture and data.** We adopt the LLaMA architecture from Touvron et al. (2023), adjusting the hyperparameters to create four model sizes: 0.1B, 0.3B, 0.6B, and 1.2B. The exact hyperparameters are deferred to Appendix D. These models are trained on the Pile dataset (Gao et al., 2020) with a context length of 4096 and a batch size of 4M tokens.

Figure 10: **Comparison With *WSD*.** We show that *WSD-S* performs favorably compared with *WSD* when the total computes is fixed, achieving a consistent improvement over *WSD* on all the model sizes when trained for approximately 200B tokens.

**Implementation.** We use a standard Adam optimizer. We set the batch size to 1024 and fixed the peak learning rate for the same model size for all the methods. For the 0.1B and 0.3B models, we use a peak learning rate of 6e-4, and for the 0.6B and 1.2B models, we use a peak learning rate of 4e-4. These values are chosen following current empirical practice (e.g. see Groeneveld et al. (2024)). We set the minimal learning rate to 0.1 of the peak learning rate. We use a TPU v3-256 model to train the model with the Levanter framework in Jax (Bradbury et al., 2018; CRFM, 2024). The fraction of time spent decaying is chosen to be 10%. The only exception is that when running *WSD* on the 0.3B models, we encounter a loss spike after training for 22.5B tokens and decay at the checkpoint trained for 22B tokens instead. This change is in favor of *WSD* in our comparison between *WSD-S* and *WSD*. The detailed hyperparameters are deferred to Appendix D.

### 5.1.1 RESULTS

***WSD-S* performs competitively with *Cosine-Oracle*.** The three endpoints of *WSD-S* are set at 50B, 100B, and 200B tokens for all models. As shown in Figure 8, *WSD-S* delivers competitive results compared to *Cosine-Oracle* in a single run.

***WSD-S* significantly outperforms *Cyclic-Cosine*.** We compare the *Cyclic-Cosine* and the *WSD-S* on 0.6B models with a total token budget of 100B tokens. Both schedules reduce to a minimal learning rate at 50B tokens to obtain an intermediate checkpoint. Our results show that *WSD-S* outperforms *Cyclic-Cosine* with a significant performance gap of 4e-2 (Figure 9). A common belief is that loss spiking after increasing the learning rate is the main cause of the performance loss in *Cyclic-Cosine*. However, this belief does not explain the advantage of *WSD-S*. We hypothesize that a model trained with a small learning rate for too long, as with Cosine, is implicitly hurt compared to a model trained with a large learning rate for the majority of the run, as with *WSD* or *WSD-S*.

To show that the model trained with *WSD* is more suitable for continual training, we conducted ablation studies by interchanging the schedules in the latter half of the runs to create two new learning rates (Cosine-*WSD* and *WSD*-Cosine). Among the four runs, the model trained using *WSD* for the first half consistently achieved lower loss in continual learning, indicating that *WSD* produces models more suitable for continual learning, even after learning rate decay.

***WSD-S* matches (and slightly outperforms) *WSD* given the same total compute.** For *WSD*, we adopt the following comparison methodology: assuming a 10% decay portion, to get three checkpoints at 12.5k, 25k, and 50k steps, *WSD* then requires corresponding total steps of 12.5k, 26.25k, and 53.75k. Hence, we examine whether *WSD-S* can output three models of matching or better performance in the same corresponding steps (see Figure 10). Our results suggest that *WSD-S* consistently outperforms *WSD* when trained on 200B tokens and underperforms *WSD* only on the smallest scale experiments when we trained 0.1B models for 25k steps. As this is the smallest scale experiment, we conclude that *WSD-S* has a slight advantage over *WSD* when the total compute is fixed. This matches our intuition that *WSD-S* can reuse the decay phases of previous checkpoints, leading to a more efficient use of the total compute. As a simpler version of *WSD*, *WSD-S* is more user-friendly for open-source pretrained models, allowing users to continue training the final checkpoint without needing intermediate ones given that the pretrained models are trained with *WSD* or *WSD-S*.

***WSD* and *WSD-S* are not sensitive to the fraction of time spent decaying** We conclude with an ablation study on the fraction of time spent decaying, and the result is shown in Figure 15. The final performance matches tightly within the range of 8% to 12%, showing a small sensitivity to the choice of the decay portion. However, in our experiments, we observe that decaying near a loss spike can lead to a significant performance loss (Figure 15, right). With the large learning rate, the training runs tend to be very volatile and there are multiple loss spikes in the training (see Figure 8). If a decay happens closely *after* a loss spike and the loss has not yet decreased to its original level, it is typical that the final validation loss will be worse by 1e-2 or even more. We observe the same phenomenon for *WSD*, and when such a scenario happens, we suggest either running longer till the loss stabilizes or rolling back to a slightly earlier checkpoint before the loss spikes and decays from there.

ACKNOWLEDGEMENT

The authors would like to thank the support of NSF 2211780 and also would like to thank the Google TPU Research Cloud for the computing resources that enabled these experiments

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

# A RELATED WORK

**Learning Rate Schedules.** Learning rate schedules are crucial in deep learning, with previous studies exploring various options. Smith (2017) was the first to propose a cyclic triangular learning rate schedule that interleaves decreasing and increasing learning rates. Loshchilov & Hutter (2017) extended the idea to a cyclic cosine learning rate schedule. He et al. (2015) introduced the notion of warmup, which gradually increases the learning rate in the earlier training phase. Goyal et al. (2018); Hoffer et al. (2018); You et al. (2020) concluded that the learning rate should scale linearly with the batch size, which is further theoretically examined in Smith et al. (2020); Li et al. (2021); Malladi et al. (2023). You et al. (2019) performed an analysis on why learning rate schedules are helpful and suspected that the large learning rate at the beginning phase is mostly useful for avoiding memorization of noisy data, which is consistent with our analysis in Section 4.

In the LLM era, works including Hoffmann et al. (2022); DeepSeek-AI et al. (2024); Hu et al. (2024) examined how to choose learning rate schedules for pretraining. In particular, Hu et al. (2024) introduced a learning rate schedule called Warmup-Stable-Decay (*WSD*) that remains constant for the majority of the runs before decaying in language model pretraining, which were studied independently in Zhai et al. (2022); Ibrahim et al. (2024); Hägele et al. (2024). Raffel et al. (2023); Ibrahim et al. (2024) explored another possibility of using an inverse square root schedule to pretrain the language models. Defazio et al. (2023) proposes to use linear decay for the entire training run. Defazio et al. (2024) shows that with appropriate iterate averaging, a constant learning rate schedule can reach better performance than the cosine learning rate schedule. Rae et al. (2022); Gupta et al. (2023); Hu et al. (2024); Ibrahim et al. (2024) examined how to choose a learning rate schedule in a continual learning setting and verified that rewarming-up cosine learning rate brings performance drops that are costly to recover. A common belief is that the performance drop is due to the sudden increase in learning rate during rewarming-up. However, our work shows that increasing the learning rate after a short decay in *WSD* does not cause a similar performance drop as seen with the cosine learning rate, challenging the previous hypothesis. Instead, we suggest that the performance loss associated with rewarming-up cosine learning rate is due to the implicit damage it causes to the model, making it unsuitable for continual training. On the contrast, *WSD* avoids such damage by maintaining a high learning rate during the stable phase, hence the sudden increase in learning rate does not lead to performance drops in continual training.

**Continual Learning.** Continual learning, the process of updating the model with newly collected data, can improve the models' knowledge and capability. Previous continual learning research (Aljundi et al., 2019; Veniat et al., 2021; Cossu et al., 2022; Dyer et al., 2022; Harun et al., 2023; Mehta et al., 2023) assumed significant domain shift and aimed to avoid forgetting old knowledge while learning new knowledge. Recent works including Hernandez et al. (2021); Lesort et al. (2023) suggested that optimizers including SGD and Adam have a knowledge accumulation effect and the effect of catastrophic forgetting may be less significant than expected, especially when replay is applied. Our work mainly focuses on continual pre-training without necessarily a strong domain shift and hence does not touch upon the effect of covariance shift. Continual learning is also extensively employed in large language models such as LLaMA to extend their capabilities, such as handling longer contexts (e.g., see Tworkowski et al. (2023); Peng et al. (2023); Chen et al. (2023); Dubey et al. (2024) and references therein) or dealing with new languages and domains (e.g., see Azerbayev et al. (2024); Rozière et al. (2024); Cui et al. (2024) and references therein).

**Theoretical Understanding on Loss Landscape.** A long line of research aims to better understand the loss landscape in deep learning (e.g., see Freeman & Bruna (2017); Garipov et al. (2018); Li et al. (2020) and references therein). We will highlight several phenomena that are related to our findings.

(1) Ill-conditioned directional sharpness and heavy-tailed noise: Zhang et al. (2020a;b) examined the gradient noise in language modeling and observed that the noise is heavy-tailed in multiple dimensions. Pan & Li (2023); Liu et al. (2024) showed that the loss has vastly different curvatures in different dimensions. Pan et al. (2022) analyzes optimizing a quadratic function with skewed curvature theoretically. Our river valley landscape is consistent with these findings.

(2) Benefit of large learning rates: Large learning rates have a provable regularizing effect in finding flatter minima (Kong & Tao, 2020; Wang et al., 2022), and flatter minima typically have a better generalization effect, even in the pretraining setting (Jiang et al., 2019; Blanc et al., 2020; Liu et al., 2022; Li et al., 2022; Ma et al., 2022; Lyu et al., 2023; Andriushchenko et al., 2023).

(3) Connecting loss landscape with feature learning: Some recent works (Nakkiran et al., 2019; Rosenfeld & Risteski, 2023) tried to understand how the loss landscape is formed through the lens of feature learning. Rosenfeld & Risteski (2023) showed that a large learning rate will cause oscillation in learning subtle classification rules while continuing to learn other more deterministic features. Wang et al. (2024) studied how to improve generalization and convergence by amplifying the update provided by the optimizer in the flat direction of the loss landscape. Wu et al. (2024); Cai et al. (2024) studied gradient descent dynamics on logistic regression, showing that a large learning rate will cause oscillation in the earlier phase but will lead to higher progress later in training. Pagliardini et al. (2024) developed a modification of the Adam optimizer based on optimization analysis on the Rosenbrock function, which is a special case of the river valley landscape. Song et al. (2024) shows that when SGD update is projected to the dominant subspace of the Hessian, the model's optimization progress slows down and they conjecture the existence of *ill-conditioned valley* in the landscape, which can be viewed as a similar and simpler version of the river valley landscape discussed in this paper.

(4) Ravines in the Loss Landscape. Concurrently with our work, Davis et al. (2024) identified the existence of a ravine in the loss landscape—a manifold where every point has a vanishing gradient within the sharp eigenspace of the Hessian. This feature appears in any smooth loss function exhibiting fourth-order growth near minimizers. They also demonstrate the advantages of using adaptive step sizes in this context. The concept of a ravine aligns closely with the river structure described in our paper and can be considered a specific instance of it.

The landscape analysis described in these previous works matches our river valley picture at a high level.

## B  ADDITIONAL EXPERIMENT RESULTS

### B.1  PRETRAINING EXPERIMENTS ON DCLM

**WSD-S outperforms WSD.** We reran our experiments on another dataset called DCLM (Li et al. (2024)) with the 0.1B and 0.6B models for both WSD and WSD-S. We use learning rates 6e-4 and 5e-4 respectively for both models and a linear learning rate decay in the decay phase. The rest of the hyperparameters is the same as before. We observed that on this dataset, we no longer suffer from loss spikes and our results continue to hold. We also tested our final models on a sampled validation set of Penn Treebank, RedPajama, RefinedWeb, and the English subset of C4. The models trained with WSD-S continue to outperform the models trained with WSD.

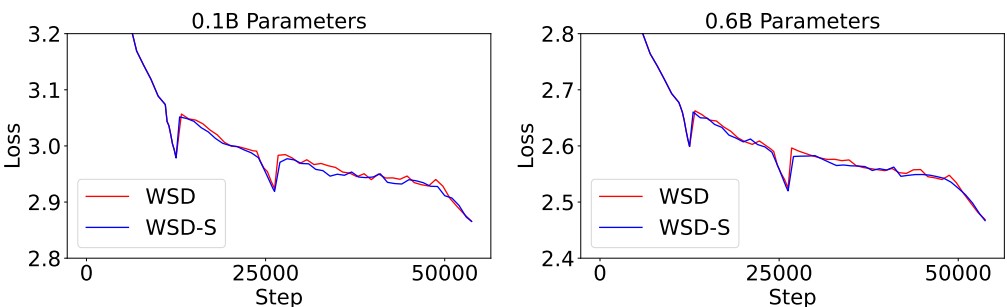

Figure 11: **Comparison With *WSD* on DCLM.** We show that *WSD-S* performs favorably compared with *WSD* when the total compute is fixed on the DCLM dataset, achieving a consistent improvement over *WSD* on all the model sizes.

**A learning rate sweep for WSD-S and Cyclic-cosine** We perform a learning rate sweep for the Cyclic-cosine method and WSD-S method on the DCLM dataset. Both methods are trained for 25000 steps and are decayed to a minimal learning rate at 12500 steps. The peak learning rate and corresponding final loss are shown in Table 1. We observe that WSD-S outperforms Cosine-Rewarmup for most choices of the learning rate and the best performance of WSD-S is also better.

| LR | 5E-4 | 1E-3 | 2E-3 | 4E-3 |
|---|---|---|---|---|
| **Cyclic-Cosine** | 2.54674 | 2.51853 | 2.49672 | 2.50063 |
| **WSD-S** | 2.52739 | 2.50944 | 2.49565 | 2.51052 |

Table 1: Comparison of methods across learning rates.

## B.2 ADDITIONAL MODE CONNECTIVITY RESULTS

**Ablations on the experiments in Section 3.3** We ablate the experiment results presented in Section 3.3, varying the starting point and the duration used for decay and stable phase in Figures 12 and 13. Our results continue to hold.

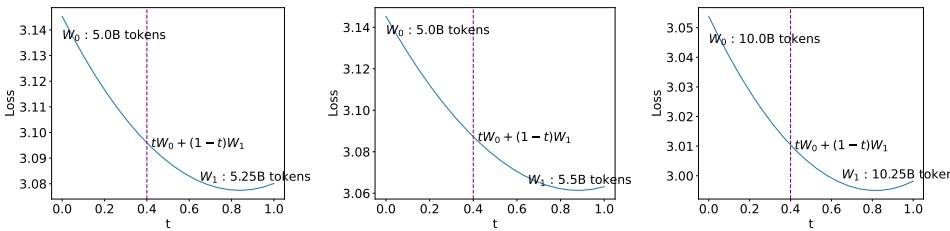

(a) Decay at 5B tokens for 0.25B tokens  (b) Decay at 5B tokens for 0.5B tokens  (c) Decay at 5B tokens for 0.5B tokens

Figure 12: **Loss smoothly decreases in decay phases** We vary the starting point of the decaying phase and the duration of the decaying phase and find that loss generally follows the smooth decreasing trend when connecting two models during the decay phase. The experiment setting is the same as Figure 5.

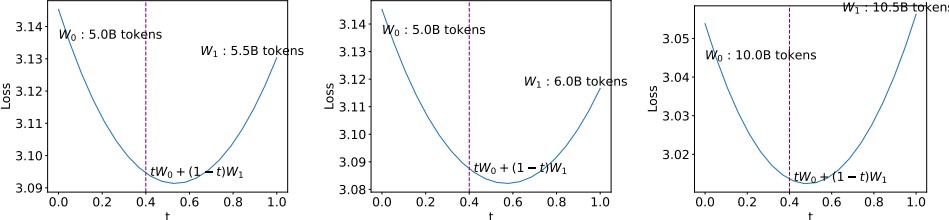

(a) Stable at 5B tokens for 0.5B tokens  (b) Stable at 5B tokens for 1B tokens  (c) Stable at 10B tokens for 0.5B tokens

Figure 13: **Losses exhibit valley shape in stable phase** We vary the starting point of the stable phase and the duration of the stable phase and find that loss generally exhibits a valley-like shape when connecting two models during the stable phase. The experiment setting is the same as Figure 5.

**2-dimensional visualization of loss.** Given a checkpoint $A$ trained using a constant learning rate, we decay the learning rate to obtain a decayed checkpoint $A'$. We then continue to train the checkpoint $A$ using a constant learning rate to obtain checkpoint $B$ and corresponding decayed checkpoint $B'$. Our assumption states that the loss is much sharper along the line $AA'$ (the sharp hillsides), then along the line $A'B'$ (the flat river). We present a visualization of the loss in this section, validating this assumption.

## C OMITTED PROOFS

### C.1 NOTATION.

To denote $a^T b$ for two vectors, we will $\langle a, b \rangle$. We will use the following function to denote the directional derivative of a mapping $F : \mathbb{R}^d \to \mathbb{R}^m$:

$$\nabla F(x)[v] = \lim_{\alpha \to 0} \frac{F(x + \alpha v) - F(x)}{\alpha}.$$

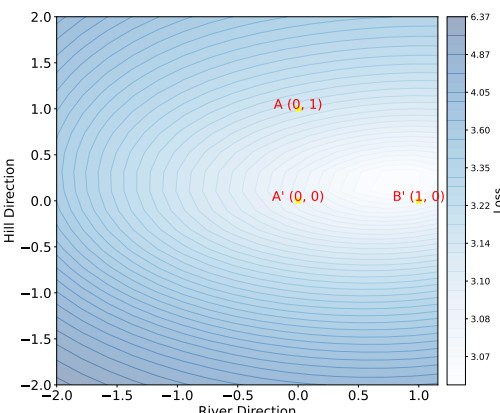

Figure 14: **2-dimensional Probing of Loss Landscape.** We choose $A$ to be a 0.1B GPT-2 model trained on OpenWebText for 5B tokens using constant learning rate 6e-4 and $A'$ being the model after decaying learning rate on 0.25B tokens. $B$ is a model trained on another 0.5B token using the constant learning rate after $A$ and $B'$ is the model after decaying learning rate on another 0.25B tokens. Our visualization shows that the loss is much flatter in the direction of $A'B'$ compared with the loss in the direction of $AA'$.

### C.2 A WARMUP ON THE QUADRATIC FUNCTION

We will first motivate the decaying function we choose using a simple example on quadratic function.

**Lemma C.1.** *Assuming that we are considering the following gradient descent*

$$y_{k+1} = y_k - \eta_k \nabla(\gamma y_k^2/2) - \eta g_k, g_k \in \mathcal{N}(0, \sigma^2 \mathcal{I}).$$

*Suppose $\eta_0 = \eta_{\max}$ and $y_0$ follows a normal distribution $\mathcal{N}(0, \eta_{\max} \frac{\sigma^2}{2\gamma - \eta_{\max}\gamma^2})$. Then the following two statements hold,*

*1. If $\forall t, \eta_k = \eta_0$, $y_k$ will follow the same distribution as $y_0$.*
*2. Consider all the learning rate schedule $\eta_k$, the following is the optimal*

$$\forall t \geq 1, \eta_k^* = \frac{1}{\gamma(k-1) + \frac{2}{\eta_{\max}}}$$

*in the sense that it yields the fastest expected loss decrease. Suppose $\eta_k^*$ corresponds to iterates variables $y_k^*$, for any $\eta_k$ and its corresponding iterates variables $y_k$,*

$$\mathbb{E}[\gamma y_k^2/2] \geq \mathbb{E}[\gamma(y_k^*)^2/2] = \frac{\sigma^2}{\gamma}\eta_k^*.$$

*Proof.* We will denote $\sigma_k = \mathbb{E}[y_k^2]$ and assume WLOG we start decayping at step $0$. Then we will have

$$\sigma_k = (1 - \eta_k \gamma)^2 \sigma_{k-1} + \eta_k^2 \sigma^2.$$

If we choose all $\eta_k = \eta_{\max}$, we can directly verify that $\sigma_k = \sigma$.

If we choose $\eta_k = \frac{\sigma_{k-1}\gamma}{\sigma_{k-1}\gamma^2 + \sigma^2}$ to minimize the right hand side, we will have that

$$\sigma_k = \frac{\sigma_{k-1}\sigma^2}{\sigma_{k-1}\gamma^2 + \sigma^2}.$$

$$\iff \frac{1}{\sigma_k} = \frac{1}{\sigma_{k-1}} + \frac{\gamma^2}{\sigma^2} = \frac{1}{\sigma_0} + \frac{\gamma^2 k}{\sigma^2}.$$

This implies $\sigma_k = \frac{1}{\frac{1}{\sigma_0} + \frac{\gamma^2 k}{\sigma^2}}$ and plugging into $\eta_k = \frac{\sigma_{k-1}\gamma}{\sigma_{k-1}\gamma^2 + \sigma^2}$ we have that

$$\eta_k^* = \frac{\gamma}{\gamma^2 + \frac{\sigma^2}{\sigma_{k-1}}} = \frac{\gamma}{\gamma^2 k + \frac{\sigma^2}{\sigma_0}} = \frac{\gamma}{\sigma^2}\sigma_k.$$

The optimality of $\eta_k^*$ can be easily inferred from the proof. $\qquad\square$

### C.3 Landscape Analysis

We will parameterize $P_F(x)$ as $v_d(x)v_d(x)^T$ and $P_S(x)$ to denote $\mathcal{I} - P_F(x)$. Throughout this section, we will assume $v_d(x)$ is continuous and pointing towards the direction of the gradient for all the $x$ on the river. The following technical lemmas will be used repetitively in the proof.

**Lemma C.2.** *Under Assumptions 1 and 2, the directional derivative of $P_S(x)$ and $P_F(x)$ exist and satisfied that*

$$\nabla P_F(x)[v] = -\nabla P_S(x)[v] = \nabla v_d(x)[v]v_d(x)^T + v_d(x)\nabla v_d(x)[v]^T.$$

*Further*

$$\nabla P_S(x)[v]P_S a = \langle \nabla v_d(x)[v], P_S a\rangle v_d(x),$$
$$\nabla P_S(x)[v]P_F a = \langle v_d, P_F a\rangle \nabla v_d(x)[v].$$
$$\|\nabla P_S(x)[v]\|_2 \le \frac{\gamma\epsilon}{\Delta}\|v\|_2.$$

*Proof.* As $\gamma_{\text{flat}}$ is an unique eigenvalue of $\nabla^2 L(x + vt)$ and $\nabla^2 L(x + vt)$ is analytical with respect to $t$, by Theorem 6.1 of Kato (1995), we know that $v_d(x + vt)$ is analytical with respect to $t$. Hence, the directional derivative exists.

The proof is by applying the chain rule and noticing that $\langle v_d(x), \nabla v_d(x)[v]\rangle = 0$ because $v_d(x)$ is always a unit vector. □

We will now define the projection of iterate to the river as the progress measure of the optimization dynamics.

**Definition C.3.** *For $U$ in Assumption 2 and any $w \in U$, we define the following ODE as the projection flow:*

$$\phi(w, 0) = w, d\phi(w, t) = -P_S(w)\nabla L(\phi(w, t))\, dt. \tag{6}$$

*When $\lim_{t\to\infty} \phi(w, t)$ is well defined, we will define $\Phi(w) = \lim_{t\to\infty} \phi(w, t)$ as the projection of $w$ to the river.*

The following lemma ensures that the projection function is well-defined and is close to $w$:

**Lemma C.4.** *Under Assumptions 1 and 2, for any $w$ satisfying that $\mathcal{B}(w, \frac{2\Delta}{\gamma}) \subset U$, $\Phi(w) \in \mathcal{M}$ exists and $\|w - \Phi(w)\|_2 \le \frac{2\|P_S(w)\nabla L(w)\|_2}{\gamma + 2\gamma_{\text{flat}}}$. Moreover, movement along the projection flow decays exponentially, $\|P_S(\phi(w, t))\nabla L(\phi(w, t))\|_2 \le \exp(-\gamma t/2)\|P_S(w)\nabla L(w)\|_2$.*

*Proof.* We will track $\|P_S(\phi(w, t))\nabla L(\phi(w, t))\|_2^2$ along the projection flow before $\phi(w, t)$ leaves $U$,

$$\frac{d\|P_S(\phi(w, t))\nabla L(\phi(w, t))\|_2^2}{dt}$$
$$= 2\langle P_S(\phi(w, t))\nabla L(\phi(w, t)), \frac{dP_S(\phi(w, t))}{dt}\nabla L(\phi(w, t)) + P_S(\phi(w, t))\frac{d\nabla L(\phi(w, t))}{dt}\rangle.$$

By Lemma C.2 and assumption 2, the first term can be bounded as

$$\langle P_S(\phi(w, t))\nabla L(\phi(w, t)), \frac{dP_S(\phi(w, t))}{dt}\nabla L(\phi(w, t))\rangle$$
$$= -\langle P_S(\phi(w, t))\nabla L(\phi(w, t)), \nabla P_S(\phi(w, t))[P_S(\phi(w, t))\nabla L(\phi(w, t))]\nabla L(\phi(w, t))\rangle$$
$$= -\langle P_S(\phi(w, t))\nabla L(\phi(w, t)), \nabla v_d(x)[P_S(\phi(w, t))\nabla L(\phi(w, t))]\rangle\langle v_d, P_F(\phi(x, t))\nabla L(\phi(x, t))\rangle$$
$$\le \|\nabla L(\phi(w, t))\|\|P_S(\phi(w, t))\nabla L(\phi(w, t))\|^2 \frac{\kappa\gamma}{\Delta} \le \kappa\gamma\|P_S(\phi(w, t))\nabla L(\phi(w, t))\|^2.$$

The second term is always negative

$$\langle P_S(\phi(w, t))\nabla L(\phi(w, t)), P_S(\phi(w, t))\frac{d\nabla L(\phi(w, t))}{dt}\rangle$$
$$= -\langle P_S(\phi(w, t))\nabla L(\phi(w, t)), P_S(\phi(w, t))\nabla^2 L(\phi(x, t))P_S(\phi(w, t))\nabla L(\phi(w, t))\rangle$$
$$\le -(\gamma + 4\gamma_{\text{flat}})\|P_S(\phi(w, t))\nabla L(\phi(w, t))\|^2.$$

Summing up the two terms,

$$\frac{d\|P_S(\phi(w,t))\nabla L(\phi(w,t))\|_2^2}{dt} \leq -(\gamma + 2\gamma_{\text{flat}})\|P_S(\phi(w,t))\nabla L(\phi(w,t))\|^2.$$

By Lemma C.36, we have $\|P_S(\phi(w,t))\nabla L(\phi(w,t))\|_2^2 \leq \exp(-(\gamma + 2\gamma_{\text{flat}})t)\|P_S(w)\nabla L(w)\|_2^2$. Hence $\forall t > 0$,

$$\begin{aligned}
\|\phi(w,t) - w\|_2 &\leq \int_0^\infty \|P_S(\phi(w,\tau))\nabla L(\phi(w,\tau))\|_2 d\tau \\
&\leq \|P_S(w)\nabla L(w)\|_2 \int_0^\infty \exp(-(\gamma + 2\gamma_{\text{flat}})t/2) \\
&\leq \frac{2\|P_S(w)\nabla L(w)\|_2}{(\gamma + 2\gamma_{\text{flat}})}.
\end{aligned}$$

As $\mathcal{B}(w, \frac{\Delta}{2\gamma}) \subset U$, the analysis hold along the trajectory and this shows that $\Phi(w) = \lim_{t\to\infty} \phi(w,t)$ exists and that $\|\Phi(w) - w\|_2 \leq \frac{2\|P_S(w)\nabla L(w)\|_2}{(\gamma + 2\gamma_{\text{flat}})}$.

Further $\Phi(w)$ satisfies that $P_S(\Phi(w))\nabla L(\Phi(w)) = 0$, and by Assumption 2, $\Phi(w) \in \mathcal{M}$. $\qquad\square$

The following lemmas focus on the properties of $\partial\Phi$.

**Lemma C.5.** *Under Assumptions 1 and 2, for any $w$ satisfying that $\mathcal{B}(w, \frac{2\Delta}{\gamma}) \subset U$, $\partial\Phi(w)$ is well-defined.*

*Proof.* Recall that $\Phi(w) = \lim_{n\to\infty} \underbrace{(\phi \circ \phi \circ \ldots \phi)}_{n \text{ times}}(w)$, as $\phi$ is differentiable (Lemma C.2) and $\mathcal{M}$ is the fixed point of $\Phi$, by Theorem 5.1 of Falconer (1983), we have that $\partial\Phi(w)$ is well-defined. $\qquad\square$

**Lemma C.6.** *Under Assumptions 1 and 2, for any $w$ satisfying that $\mathcal{B}(w, \frac{2\Delta}{\gamma}) \subset U$, it holds that $\partial\Phi(w)P_S(w)\nabla L(w) = 0$. Further for any $w = x(t) \in \mathcal{M}$, it holds that $\partial\Phi(w)P_S(w) = 0, \partial\Phi(w)\frac{dx(t)}{dt} = \frac{dx(t)}{dt}$ and that for any $v$, $\partial\Phi(w)v$ aligns with $\frac{dx(t)}{dt}$.*

*Proof.* According to Lemmas C.4 and C.6, $\Phi, \partial\Phi$ is well-defined when $\mathcal{B}(w, \frac{2\Delta}{\gamma}) \subset U$. Based on Definition C.3, we have that

$$\forall t, \Phi(\phi(w,t)) = \Phi(w).$$

Hence,

$$\frac{d\Phi(\phi(w,t))}{dt} \Big|_{t=0} = 0,$$

Therefore,

$$0 = \partial\Phi(w)\frac{d\phi(w,t)}{dt} \Big|_{t=0} = -\partial\Phi(w)P_S(w)\nabla L(w).$$

For any $w \in \mathcal{M}$ and any $v \in \mathbb{R}^d$, it holds that

$$\begin{aligned}
0 &= \frac{d\partial\Phi(w + \alpha v)P_S(w + \alpha v)\nabla L(w + \alpha v)}{d\alpha} \Big|_{\alpha=0} \\
&= \partial^2\Phi(w)[v]P_S(w)\nabla L(w) + \partial\Phi(w)\Big[\partial P_S(w)[v]\nabla L(w) + P_S(w)\nabla^2 L(w)v\Big] \\
&= \partial\Phi(w)\Big[\partial P_S(w)[v]\nabla L(w) + P_S(w)\nabla^2 L(w)v\Big]. \qquad (7)
\end{aligned}$$

Define $J_w(v)$ as the projection from $v$ to $\partial P_S(w)[v]\nabla L(w) + P_S(w)\nabla^2 L(w)v$.

**Lemma C.7.** *$J_w(v)$ is a linear projection and the range of $J_w$ is the range of $P_S$.*

*Proof.* Based on Lemma C.2, $P_S \partial P_S(w)[v] \nabla L(w) = \partial P_S(w)[v] \nabla L(w)$. Hence the range of $J_w$ is a subspace of the range of $P_S$. When $v = P_S(w)u \neq 0$, based on Assumption 2,

$$\|J_w(v)\|_2 \geq \|P_S \nabla^2 L(w) P_S(w)u\|_2 - \|\partial P_S(w)[P_S(w)u]\nabla L(w)\|_2 \geq \gamma\|u\|_2 - \gamma\kappa\|u\|_2 > 0.$$

Hence the range of $J_w$ has a dimension no smaller than the dimension of the range of $P_S(w)$. This concludes that the range of $J_w$ is the range of $P_S(w)$. $\qquad\square$

Hence by Equation (7) and lemma C.7, it holds that for $w \in \mathcal{M}$, $\partial\Phi(w)P_S(w) = 0$. This shows that the range of $\partial\Phi(w)$ has dimension 1.

Finally for any $w \in \mathcal{M}$, $\Phi(w) = w$. Hence,

$$\frac{d\Phi(x(t)) - dx(t)}{dt} = 0.$$

$$\partial\Phi(x(t))\frac{dx(t)}{dt} = \frac{dx(t)}{dt}.$$

Hence the range of $\partial\Phi(w)$ contains $\frac{dx(t)}{dt}$, this concludes the proof. $\qquad\square$

**Lemma C.8.** *Under Assumptions 1 and 2, for any $w$ satisfying that $\mathcal{B}(w, \frac{2\Delta}{\gamma}) \subset U$, it holds that*

$$\frac{\|P_F[\Phi(w)]\partial\Phi(w)\nabla L(w) - P_F(w)\nabla L(w)\|_2}{\|P_F(w)\nabla L(w)\|_2} \leq 5\kappa,$$

$$\frac{\|P_S[\Phi(w)]\partial\Phi(w)\nabla L(w)\|_2}{\|P_F(w)\nabla L(w)\|_2} \leq 5\kappa.$$

*Proof.* First, by Lemma C.6, it holds that $\partial\Phi(w)\nabla L(w) = \partial\Phi(w)P_F(w)\nabla L(w)$. Define

$$v = P_F(w)\nabla L(w)/\|P_F(w)\nabla L(w)\|_2,$$
$$s(t) = P_S(\phi(w,t))\partial\phi(w,t)[v],$$
$$f(t) = P_F(\phi(w,t))\partial\phi(w,t)[v],$$

it holds that $s(0) = 0$ and $f(0) = v$ as $\phi(w, 0) = w$.

We will bound the changes of $s(t)$ and $f(t)$. We will begin with calculating the time derivative of $\partial\phi(w,t)[v]$.

$$\begin{aligned}
\frac{d\partial\phi(w,t)[v]}{dt} &= \partial\left(\frac{d\phi(w,t)}{dt}\right)[v] \\
&= -\partial\left(P_S(\phi(w,t))\nabla L(\phi(w,t))\right)[v] \\
&= -\partial P_S(\phi(w,t))[\partial\phi(w,t)[v]]\nabla L(\phi(w,t)) \\
&\quad - P_S(\phi(w,t))\nabla^2 L(\phi(w,t))\partial\phi(w,t)[v] \\
&= -\partial P_S(\phi(w,t))[\partial\phi(w,t)[v]]\nabla L(\phi(w,t)) \\
&\quad - P_S(\phi(w,t))\nabla^2 L(\phi(w,t))s(t).
\end{aligned} \qquad (8)$$

We will now bound $\frac{d\|s(t)\|_2}{dt}$,

$$\begin{aligned}
\frac{d\|s(t)\|_2^2}{dt} &= 2\langle s(t), \frac{ds(t)}{dt}\rangle \\
&= 2\langle s(t), \nabla P_S(\phi(w,t))[\frac{d\phi(w,t)}{dt}]\partial\phi(w,t)[v] + P_S(\phi(w,t))\frac{d\partial\phi(w,t)[v]}{dt}\rangle \\
&= -2\langle s(t), \nabla P_S(\phi(w,t))[P_S(\phi(w,t))\nabla L(\phi(w,t))]\partial\phi(w,t)[v]\rangle \\
&\quad - 2\langle s(t), P_S(\phi(w,t))\frac{d\partial\phi(w,t)[v]}{dt}\rangle.
\end{aligned}$$

By Lemmas C.2 and C.4 and assumption 2, the first term satisfies that,

$$\langle s(t), \nabla P_S(\phi(w,t))[P_S(\phi(w,t))\nabla L(\phi(w,t))]\partial\phi(w,t)[v]\rangle \leq \gamma\kappa\|s(t)\|_2\|s(t) + f(t)\|_2.$$

By Equation (8) and assumption 2, the second term satisfies that,

$$
\langle s(t), P_S(\phi(w,t))\frac{d\partial\phi(w,t)[v]}{dt}\rangle
$$
$$
= -\langle s(t), \partial P_S(\phi(w,t))[\partial\phi(w,t)[v]]\nabla L(\phi(w,t))\rangle
$$
$$
- \langle s(t), P_S(\phi(w,t))\nabla^2 L(\phi(w,t))s(t)\rangle
$$
$$
\leq \gamma\kappa\|s(t)\|_2\|s(t)+f(t)\|_2 - \gamma\|s(t)\|_2^2.
$$

Hence $2\|s(t)\|_2\frac{d\|s(t)\|_2}{dt} = \frac{d\|s(t)\|_2^2}{dt} \leq -2\gamma\|s(t)\|_2^2 + 4\gamma\kappa\|s(t)\|_2\|s(t)+f(t)\|_2$ and we can conclude that

$$
\frac{d\|s(t)\|_2}{dt} \leq -\gamma\|s(t)\|_2/2 + 2\gamma\kappa\|f(t)\|_2. \tag{9}
$$

Similarly, we can provide a bound for $\|\frac{df(t)}{dt}\|_2$,

$$
\|\frac{df(t)}{dt}\| = 2\|\nabla P_F(\phi(w,t))[\frac{d\phi(w,t)}{dt}]\partial\phi(w,t)[v] + P_F(\phi(w,t))\frac{d\partial\phi(w,t)[v]}{dt}\|_2
$$
$$
= -2\|\nabla P_F(\phi(w,t))[P_S(\phi(w,t))\nabla L(\phi(w,t))]\partial\phi(w,t)[v]
$$
$$
+ P_F(\phi(w,t))\frac{d\partial\phi(w,t)[v]}{dt}\|_2.
$$

By Lemmas C.2 and C.4 and assumption 2, the first term satisfies that,

$$
\|\nabla P_F(\phi(w,t))[P_S(\phi(w,t))\nabla L(\phi(w,t))]\partial\phi(w,t)[v]\|
$$
$$
= \|\nabla P_S(\phi(w,t))[P_S(\phi(w,t))\nabla L(\phi(w,t))]\partial\phi(w,t)[v]\|
$$
$$
\leq \gamma\kappa\exp(-\gamma t/2)\|f(t)\|_2\|s(t)+f(t)\|_2.
$$

By Lemmas C.2 and C.4 and assumption 2, the second term satisfies that,

$$
\|P_F(\phi(w,t))\frac{d\partial\phi(w,t)[v]}{dt}\|
$$
$$
= \|P_F(\phi(w,t))\partial P_F(\phi(w,t))[\partial\phi(w,t)[v]]\nabla L(\phi(w,t))\rangle\|
$$
$$
= \|P_F(\phi(w,t))\partial P_F(\phi(w,t))[\partial\phi(w,t)[v]]P_S(\phi(w,t))\nabla L(\phi(w,t))\rangle\|
$$
$$
\leq \gamma\kappa\exp(-\gamma t/2)\|f(t)\|_2\|s(t)+f(t)\|_2.
$$

Hence we can conclude that

$$
\|\frac{df(t)}{dt}\|_2 \leq 2\gamma\kappa\exp(-\gamma t/2)(\|f(t)\|_2 + \|s(t)\|_2). \tag{10}
$$

By Equation (9), it holds that

$$
\frac{d(\exp(\gamma t/2)\|s(t)\|_2)}{dt} = \gamma\exp(\gamma t/2)\|s(t)\|_2/2 + \exp(\gamma t/2)\frac{d\|s(t)\|_2}{dt} \leq 2\gamma\kappa\exp(\gamma t/2)\|f(t)\|_2.
$$

Integrating the above equation from 0 to $t$, and we have

$$
\|s(t)\|_2 \leq 2\gamma\kappa\int_0^t \exp(\gamma(\tau-t)/2)\|f(\tau)\|_2 d\tau. \tag{11}
$$

By Equations (10) and (11), we have that

$$
\left|\frac{d\|f(t)\|_2}{dt}\right| \leq 2\gamma\kappa\exp(-\gamma t/2)\|f(t)\|_2 + 4\gamma^2\kappa^2\int_0^t \exp(\gamma(\tau-2t)/2)f(\tau)d\tau.
$$

This suggests that

$$
\|f(T)\|_2 \leq 1 + 2\gamma\kappa\int_0^T \exp(-\gamma t/2)\|f(t)\|_2 dt + 4\gamma^2\kappa^2\int_0^T\int_0^t \exp(\gamma(\tau-2t)/2)f(\tau)d\tau dt.
$$

Define $M(t) = \sup_{0 \leq \tau \leq t} f(t)$, then it holds that

$$\|M(T)\|_2 \leq 1 + \|M(T)\|_2 \left( 2\gamma\kappa \int_0^T \exp(-\gamma t/2)dt + 4\gamma^2\kappa^2 \int_0^T \exp(-\gamma t/2) \int_0^t \exp(\gamma(\tau-t)/2)d\tau dt \right).$$

$$\leq 1 + \|M(T)\|_2 \left( 4\kappa + 16\kappa^2 \right).$$

This implies that $\forall t, \|f(t)\|_2 \leq \|M(t)\|_2 \leq \frac{1}{1-4\kappa-16\kappa^2} \leq 1 + 5\kappa$. By Equation (11), this suggests that $\|s(t)\|_2 \leq 4\kappa(1+5\kappa) \leq 5\kappa$. Finally, returning to Equation (10), we have that

$$\|\frac{f(t)}{dt}\|_2 \leq 2\gamma\kappa \exp(-\gamma t/2)(\|f(t)\|_2 + \|s(t)\|_2)$$

$$\leq 2\gamma\kappa \exp(-\gamma t/2)(1 + 10\kappa).$$

Hence $\|f(t) - f(0)\|_2 \leq 2 \int_0^\infty 2\gamma\kappa \exp(-\gamma t/2)(1+10\kappa) \leq 2\kappa(1+10\kappa) \leq 5\kappa$.

We have that

$$\frac{\|P_F[\Phi(w)]\partial\Phi(w)\nabla L(w) - P_F(w)\nabla L(w)\|_2}{\|P_F(w)\nabla L(w)\|_2} = \lim_{t\to\infty} \|f(t) - f(0)\|_2 \in [0, 5\kappa],$$

and

$$\frac{\|P_S[\Phi(w)]\partial\Phi(w)\nabla L(w)\|_2}{\|P_F(w)\nabla L(w)\|_2} = \lim_{t\to\infty} \|s(t)\|_2 \in [0, 5\kappa],$$

The proof is then complete. $\qquad\square$

The following lemma generalizes Lemma C.8 to general direction instead of $\nabla L(w)$.

**Lemma C.9.** *Under Assumptions 1 and 2, for any $w$ satisfying that $\mathcal{B}(w, \frac{2\Delta}{\gamma}) \subset U$, it holds that*

$$\frac{\|P_F[\Phi(w)]\partial\Phi(w)u - P_F(w)u\|_2}{\|P_F(w)\nabla L(w)\|_2} \leq 5\kappa,$$

$$\frac{\|P_S[\Phi(w)]\partial\Phi(w)u\|_2}{\|P_F(w)u\|_2} \leq 5\kappa.$$

*Proof.* We only need to notice that $P_F(w)u$ aligns with $v_d(\nabla^2 L(w))$. Hence, it always holds that

$$P_F(w)u = P_F(w)\nabla L(w)\frac{\langle P_F(w)\nabla L(w), P_F(w)u \rangle}{\|P_F(w)\nabla L(w)\|_2^2},$$

$$\partial\Phi(w)u = \partial\Phi(w)P_F(w)u = \partial\Phi(w)\nabla L(w)\frac{\langle P_F(w)\nabla L(w), P_F(w)u \rangle}{\|P_F(w)\nabla L(w)\|_2^2}.$$

The proof is then complete. $\qquad\square$

The following lemma states that the angle between the gradient and the tangent direction is small for any point on the river.

**Lemma C.10.** *For any $w \in \mathcal{M}$, it holds that*

$$\|P_{\mathcal{M}}(w)\nabla L(w) - \nabla L(w)\|_2 \leq 4\kappa\|P_{\mathcal{M}}(w)\nabla L(w)\|_2.$$

*Proof.* Assume $w = x(T)$, we will denote $P_{\mathcal{M}}(w)\nabla L(w)$ by $v$.

It holds that

$$\nabla\big(P_S(w)\nabla L(w)\big)[v] = 0,$$

which can be simplified to

$$P_S(w)\nabla^2 L(w)v + \nabla P_S(w)[v]\nabla L(w) = 0.$$

The first term satisfies that $\|P_S(w)\nabla^2 L(w)v\|_2 \geq \gamma\|P_S(w)v\|$ and the second term satisfies that $\|\nabla P_S(w)[v]\nabla L(w)\|_2 \leq \gamma\kappa\|v\|$. This then suggests $\|P_S(w)v\|_2 \leq \kappa\|v\|_2$.

Therefore $\|P_F(w)v\|_2 \geq (1-\kappa)\|v\|_2$. As

$$v = \frac{dx(t)}{dt}\Big|_{t=T} = -P_{\mathcal{M}}(w)\,\nabla L(w)$$

We know that $\left|v_d^\top P_{\mathcal{M}}(w)v_d\right| \geq (1-\kappa)\|P_{\mathcal{M}}(w)v_d\|_2$, which suggests that $\left|v_d^\top \frac{P_{\mathcal{M}}(w)v_d}{\|P_{\mathcal{M}}(w)v_d\|_2}\right| \geq (1-\kappa)$. Hence we can conclude that $\|P_{\mathcal{M}}(w)v_d\|_2 \geq (1-\kappa)$. Hence, we know that

$$\|v + \nabla L(w)\|_2 \leq \sqrt{1-(1-\kappa)^2}\|\nabla L(w)\|_2 \leq 2\kappa\|\nabla L(w)\|_2 \leq \frac{2\kappa}{1-\kappa}\|v\|_2 \leq 4\kappa\|v\|_2.$$

This concludes the proof. □

The next lemma states that $P_F(w)\nabla L(w)$ and $\nabla L(\Phi(w))$ is always close.

**Lemma C.11.** *Under Assumptions 1 and 2, for any $w$ satisfying that $\mathcal{B}(w, \frac{2\Delta}{\gamma}) \subset U$, it holds that*

$$\|P_F(w)\nabla L(w) - \nabla L(\Phi(w))\|_2 \leq (\gamma\kappa + \gamma_{\text{flat}})\|w - \Phi(w)\|_2.$$

*Proof.* By Lemma C.4, the line segment from $\Phi(w)$ to $w$ lies in $U$. By Assumption 2 and lemma C.4,

$$\|P_F(w)\nabla L(w) - P_F(\Phi(w))\nabla L(\Phi(w))\|_2$$

$$= \|\int_0^1 \nabla P_F(\Phi(w) + t(w - \Phi(w)))[w - \Phi(w)]\nabla L(\Phi(w) + t(w - \Phi(w)))dt$$

$$+ \int_0^1 P_F(\Phi(w) + t(w - \Phi(w)))\nabla^2 L(\Phi(w) + t(w - \Phi(w)))(w - \Phi(w))dt\|_2$$

$$\leq (\gamma\kappa + \gamma_{\text{flat}})\|w - \Phi(w)\|_2.$$

This concludes the proof. □

The final theorem states that when $w$ is near the river, the movement of its projection has a similar value as the inherent speed at the river.

**Lemma C.12.** *Under Assumptions 1 and 2, when $\|w - \Phi(w)\|_2 \leq \frac{10\kappa\|P_F(w(t))\nabla L(w(t))\|_2}{\gamma + \gamma_{\text{flat}}}$,*

$$\|P_F(w)\nabla L(w) + \frac{dx(\tau)}{d\tau}\Big|_{\tau=T}\|_2 \leq 16\kappa\|\frac{dx(\tau)}{d\tau}\Big|_{\tau=T}\|_2.$$

$$\|\partial\Phi(w)\nabla L(w) + \frac{dx(\tau)}{d\tau}\Big|_{\tau=T}\|_2 \leq 30\kappa\|\frac{dx(\tau)}{d\tau}\Big|_{\tau=T}\|_2.$$

*Proof.* By Lemma C.8

$$\|\partial\Phi(w)\nabla L(w) - P_F(w)\nabla L(w)\|_2 \leq 10\kappa\|P_F(w)\nabla L(w)\|_2.$$

Combining Lemma C.11 and $\|w - \Phi(w)\|_2 \leq \frac{10\kappa\|P_F(w(t))\nabla L(w(t))\|_2}{\gamma + \gamma_{\text{flat}}}$, we have that

$$\|P_F(w)\nabla L(w) - \nabla L(\Phi(w))\|_2 \leq (\gamma\kappa + \gamma_{\text{flat}})\|w - \Phi(w)\| \leq 10\kappa\|P_F(w)\nabla L(w)\|_2.$$

By Lemma C.10, let $v = \frac{dx(\tau)}{d\tau}\Big|_{\tau=T}$,

$$\|v + \nabla L(\Phi(w))\|_2 \leq 4\kappa\|v\|_2.$$

Combining the three inequalities, we have that

$$\|P_F(w)\nabla L(w) - \nabla L(\Phi(w))\|_2 \leq \frac{10\kappa}{1-10\kappa}\|\nabla L(\Phi(w))\|_2 \leq \frac{1+4\kappa}{1-10\kappa}10\kappa\|v\|_2 \leq 12\kappa\|v\|_2.$$

This suggests that

$$\|P_F(w)\nabla L(w) - v\|_2 \le |P_F(w)\nabla L(w) - \nabla L(\Phi(w))\|_2 + \|v + \nabla L(\Phi(w))\|_2 \le 16\kappa\|v\|_2.$$

Hence

$$\|v + \partial\Phi(w)\nabla L(w)\|_2$$
$$\le \|P_F(w)\nabla L(w) - v\|_2 + \|P_F(w)\nabla L(w) - \nabla L(\Phi(w))\|_2$$
$$\le 16\kappa\|v\|_2 + 10\kappa\|P_F(w)\nabla L(w)\|_2$$
$$\le 30\kappa\|v\|_2.$$

This concludes the proof. $\square$

### C.4 RIVER EXISTS UNDER MILD ASSUMPTIONS.

In this subsection, we will provide two results stating the local existence of rivers under the existence of eigengap (Assumption 4). Recall we define the river as a smooth manifold of points with vanishing gradients in sharp directions.

1. River exists and every point in $V$ is close to some part of the river (Lemma C.13).
2. River is a 1-dimensional manifold. (Lemma C.14).

Combining the two statements, we can conclude that there is always a river near every point under the existence of eigengap.

**Assumption 4.** *There exists an open set $U$ satisfying the following assumptions:*

1. *Analyticity. $L(w)$ is analytic with respect to $w$.*
2. *Existence of Eigengap. There exist constants $\gamma_{\text{flat}}, \gamma > 0$, such that $\forall w \in U, \lambda_{d-1}\left(\nabla^2 L(w)\right) > \gamma + 4\gamma_{\text{flat}}, |\lambda_d\left(\nabla^2 L(w)\right)| < \gamma_{\text{flat}}$.*
3. *Slow Spinning of $v_d$. There exist constants $\Delta > \Delta_{\min} > 0, \kappa \in [0, 0.01)$, such that $\forall w \in U, \Delta_{\min} < \|\nabla L(w)\|_2 \le \Delta$, and $\|\nabla v_d\left(\nabla^2 L(w)\right)\|_{\text{op}} \le \kappa\gamma/(2\Delta)$. This means that the flat direction $v_d$ changes slowly during optimization.*
4. *Conservation of Gradient Flows. There exists an open subset $V \subset U$ and a constant $r > \frac{10\Delta}{\gamma}$ for $\gamma$ defined in Assumption 2.3 such that $\forall w \in V$, the $r$-neighborhood of the gradient flow starting from $w$ stays in $U$ for continuous time $T_{\max} \ge 10\log(2\Delta/(\kappa\Delta_{\min}))/\gamma$.*

We note that Assumption 4 is a strict subset of Assumption 2.

**Lemma C.13.** *Under Assumption 4, for every $w \in V$, there exists $w' \in U$, such that $\|w - w'\|_2 \le \frac{2\Delta}{\gamma}$ and $\|P_S[w']\nabla L(w')\|_2 = 0$.*

*Proof.* We will define $w' = \Phi(w)$ for $\Phi$ defined in the same way as in Definition C.3. The rest of the proof goes in the same line as Lemma C.4. We note that in the proof of Lemma C.4, our deduction does not depend on the existence of the river until the last line. $\square$

**Lemma C.14.** *Under Assumption 4, for every $x \in U$ satisfying $\|P_S[x]\nabla L(x)\|_2 = 0$, there exists a smooth 1-dimensional manifold $\mathcal{M}$ passing through $x$, such that for every point $u \in \mathcal{M}$, the projected gradient onto the sharp directions vanishes, i.e.,*

$$P_S(u)\nabla L(u) = 0.$$

*Proof.* To establish the existence of the river $\mathcal{M}$ as a smooth 1-dimensional manifold passing through $x$, we apply the Implicit Function Theorem to the system of equations defined by the vanishing of the projected gradient onto the sharp directions.

Fixing the coordinate vector as $e_1, ..., e_d$, we will assume the rotation rotating from $e_1$ to $v$, and keeping all the vectors orthogonal to $e_1$ and $v$ constant as $R(v)$. We then have $R(v)$ is a smooth function of $v$ as long as $v$ is not close to $e_1$. We can then assume $D(v) = \sum_{i=2}^{d} e_i'(R(v)e_i)^T \in \mathbb{R}^{(d-1)\times d}$ with $e_i'$ being the $(i-1)$-th coordinate vector in $\mathbb{R}^{d-1}$. $D(v)$ is then also a smooth function in $v$.

We will now assume without loss of generality $v$ does not align with $e_1$ and define $K(u) = D(v_d(u))$, then $K(u)$ is a smooth function in $u$ that maps the sharp component of each vector to $\mathbb{R}^{d-1}$.

Define the constraint function $F : \mathbb{R}^d \to \mathbb{R}^{d-1}$ by

$$F(u) = K(u)P_S(u)\nabla L(u).$$

We aim to show that the solution set $F^{-1}(0)$ near $x$ is a smooth 1-dimensional manifold. To apply the implicit function Theorem, we need to verify that:

1. Smoothness. The function $F$ is continuously differentiable in a neighborhood of $x$. Given that $L(u)$ is analytic (and hence smooth) by Assumption 4, and $P_S(u)$ is defined in terms of the continuously differentiable projection $P_F(u)$, it follows that $F$ is smooth.
2. Rank of the Jacobian is $d-1$ at $x$. The Jacobian matrix $DF(x) \in \mathbb{R}^{d \times (d-1)}$ must have rank $d-1$.
   The Jacobian of $F$ at point $x$ is given by:

$$DF(x) = K(x)[\partial P_S(x)\nabla L(x) + P_S(x)\nabla^2 L(x)] + \partial K(x)P_S(x)\nabla L(x).$$
$$= K(x)[\partial P_S(x)\nabla L(x) + P_S(x)\nabla^2 L(x)].$$

Consider $J_u$ defined in Lemma C.7, $DF(x) = K(x)J_x$. Applying the same argument shows that $J_x$ has rank $d-1$ and the range of $J_x$ is the range of $P_S(x)$, i.e., the sharp space. Therefore $D_f$ has rank $d-1$.

Since $F$ is smooth and the Jacobian $DF(\Phi(w))$ has rank $d-1$, the implicit function Theorem guarantees that the solution set $F^{-1}(0)$ near $\Phi(w)$ is a smooth manifold of dimension $d - (d-1) = 1$. Thus, there exists a smooth 1-dimensional manifold $\mathcal{M}$ passing through $\Phi(w)$ where the projected gradient vanishes:

$$P_S(u)\nabla L(u) = 0 \quad \forall u \in \mathcal{M}.$$

Furthermore, the smoothness of $F$ ensures that the manifold $\mathcal{M}$ is not only locally 1-dimensional but also smoothly parameterized. $\square$

## C.5 PROOF OF THEOREM 3.2

We will consider the following gradient flow:

$$dw(t) = -\nabla L(w(t))dt, w(0) \in V. \tag{12}$$

We will first prove that along the gradient flow trajectory, it holds that $\|P_S(w)\nabla L(w)\|_2$ is bounded.

**Lemma C.15.** *Under Assumptions 1 and 2, along the gradient flow Equation* (12)*, it holds that for $t \geq 2\log(2\Delta/(\kappa\Delta_{\min}))/\gamma$, $\|P_S(w(t))\nabla L(w(t))\|_2 \leq 2\kappa\|P_F(w(t))\nabla L(w(t))\|_2$.*

*Proof.* We will first compute how fast $\|P_S(w(t))\nabla L(w(t))\|_2$ can change

$$\frac{d\|P_S(w(t))\nabla L(w(t))\|_2^2}{dt} = -2\langle P_S(w(t))\nabla L(w(t)), \partial P_S(w(t))[\nabla L(w(t))]\nabla L(w(t))\rangle$$
$$-2\langle P_S(w(t))\nabla L(w(t)), P_S(w(t))\nabla^2 L(w(t))\nabla L(w(t))\rangle$$

By Lemma C.2 and assumption 2, the first term satisfies,

$$-2\langle P_S(w(t))\nabla L(w(t)), \partial P_S(w(t))[\nabla L(w(t))]\nabla L(w(t))\rangle$$
$$= -2\langle P_S(w(t))\nabla L(w(t)), \partial P_S(w(t))[\nabla L(w(t))]P_F L(w(t))\nabla L(w(t))\rangle$$
$$\leq 2\kappa\gamma\|P_S(w(t))\nabla L(w(t))\|_2\|P_F(w(t))\nabla L(w(t))\|_2$$

By Assumption 2, the second term satisfies,

$$-2\langle P_S(w(t))\nabla L(w(t)), P_S(w(t))\nabla^2 L(w(t))\nabla L(w(t))\rangle$$
$$\leq -2(\gamma + \gamma_{\text{flat}})\|P_S(w(t))\nabla L(w(t))\|_2^2.$$

Hence,

$$\frac{d\|P_S(w(t))\nabla L(w(t))\|_2}{dt} \leq \kappa\gamma\|P_F(w(t))\nabla L(w(t))\|_2 - (\gamma + \gamma_{\text{flat}})\|P_S(w(t))\nabla L(w(t))\|_2.$$

(13)

We then consider the corresponding $P_F(w(t))\nabla L(w(t))$.

$$\frac{d\|P_F(w(t))\nabla L(w(t))\|_2^2}{dt} = -2\langle P_F(w(t))\nabla L(w(t)), \partial P_F(w(t))[\nabla L(w(t))]\nabla L(w(t))\rangle$$
$$-2\langle P_F(w(t))\nabla L(w(t)), P_F(w(t))\nabla^2 L(w(t))\nabla L(w(t))\rangle$$

By Lemma C.2 and assumption 2, the first term satisfies,

$$-2\langle P_F(w(t))\nabla L(w(t)), \partial P_F(w(t))[\nabla L(w(t))]\nabla L(w(t))\rangle$$
$$= -2\langle P_F(w(t))\nabla L(w(t)), \partial P_F(w(t))[\nabla L(w(t))]P_S L(w(t))\nabla L(w(t))\rangle$$
$$\leq 2\kappa\gamma\|P_S(w(t))\nabla L(w(t))\|_2\|P_S(w(t))\nabla L(w(t))\|_2$$

By Assumption 2, the second term satisfies,

$$-2\langle P_F(w(t))\nabla L(w(t)), P_F(w(t))\nabla^2 L(w(t))\nabla L(w(t))\rangle$$
$$\leq 2\gamma_{\text{flat}}\|P_F(w(t))\nabla L(w(t))\|_2^2.$$

Hence, we have that

$$\left|\frac{d\|P_F(w(t))\nabla L(w(t))\|_2}{dt}\right| \leq \kappa\gamma\|P_S(w(t))\nabla L(w(t))\|_2 + \gamma_{\text{flat}}\|P_F(w(t))\nabla L(w(t))\|_2. \quad (14)$$

Choose $\alpha_\kappa = \frac{1-\sqrt{1-4\kappa^2}}{2\kappa} < 1.5\kappa$ as the solution to the quadratic equation $\kappa\alpha^2 - \alpha + \kappa = 0$.
Then combining Equations (13) and (14), it holds that

$$\frac{d\left(\|P_S(w(t))\nabla L(w(t))\|_2 - \alpha_\kappa\|P_F(w(t))\nabla L(w(t))\|_2\right)}{dt}$$
$$\leq \gamma(-1 + \kappa\alpha_\kappa)\|P_S(w(t))\nabla L(w(t))\|_2 + \kappa\gamma\|P_F\|_2$$
$$- \gamma_{\text{flat}}\left(|P_S(w(t))\nabla L(w(t))\|_2 - \alpha_\kappa\|P_F(w(t))\nabla L(w(t))\|_2\right).$$

Notice that

$$\frac{(-1 + \kappa\alpha_\kappa)}{\kappa} = \frac{-1}{\alpha_\kappa}$$

Hence,

$$\frac{d\left(\|P_S(w(t))\nabla L(w(t))\|_2 - \alpha_\kappa\|P_F(w(t))\nabla L(w(t))\|_2\right)}{dt}$$
$$\leq -\left(\gamma(1 - \kappa\alpha_\kappa) + \gamma_{\text{flat}}\right)\left(\|P_S(w(t))\nabla L(w(t))\|_2 - \alpha_\kappa\|P_F(w(t))\nabla L(w(t))\|_2\right).$$

By Lemma C.36, this suggests that

$$\|P_S(w(t))\nabla L(w(t))\|_2 - \alpha_\kappa\|P_F(w(t))\nabla L(w(t))\|_2$$
$$\leq \exp(-(\gamma(1 - \kappa\alpha_\kappa) + \gamma_{\text{flat}})t)\left(\|P_S(w(0))\nabla L(w(0))\|_2\right)$$
$$\leq \exp(-\gamma t/2)\|P_S(w(0))\nabla L(w(0))\|_2.$$

Hence,

$$\|P_S(w(t))\nabla L(w(t))\|_2 \leq 1.5\kappa\|P_F(w(t))\nabla L(w(t))\|_2 + \exp(-\gamma t/2)\left(\|P_S(w(0))\nabla L(w(0))\|_2\right).$$

$\square$

**Lemma C.16.** *Under Assumptions 1 and 2, along the gradient flow Equation* (12)*, it holds that* $w(t) \in U$ *and* $\|w(t) - \Phi(w(t))\|_2 \leq \frac{4\kappa\|P_F(w(t))\nabla L(w(t))\|_2 + 2\exp(-\gamma t/2)\Delta}{\gamma + \gamma_{\text{flat}}}$.

*Proof.* This is a direct combination of Lemmas C.4 and C.15. $\qquad\square$

**Lemma C.17.** *Under Assumptions 1 and 2, along the gradient flow Equation* (12)*, if* $T(t)$ *satisfies* $x(T(t)) = \Phi(w(t))$*, then*

$$\frac{dT(t)}{dt} \in [1 - 30\kappa, 1 + 30\kappa].$$

*Proof.* As $T(t)$ satisfies $x(T(t)) = \Phi(w(t))$, taking derivative on both sides yield,

$$\frac{dT(t)}{dt}\frac{dx(\tau)}{d\tau}\mid_{\tau=T(t)} = -\partial\Phi(w(t))\nabla L(w(t)).$$

By Lemmas C.11 and C.16, it holds that

$$\|\partial\Phi(w(t))\nabla L(w(t)) - \frac{dx(\tau)}{d\tau}\mid_{\tau=T(t)}\|_2 \leq 30\kappa\frac{dx(\tau)}{d\tau}\mid_{\tau=T(t)}.$$

We then have that

$$|\frac{dT(t)}{dt} - 1| \leq 30\kappa,$$

which concludes the proof. $\qquad\square$

*Proof of Theorem 3.2.* The proof is a direct combination of Lemmas C.16 and C.17. $\qquad\square$

## C.6    Proof of Theorem 3.3

We will consider the following gradient descent:

$$w_{k+1} - w_t = -\eta\nabla L(w_t), w_0 \in \mathcal{M}. \tag{15}$$

We will track the changes of $P_F(w(t))\nabla L(w(t))$ and $P_S(w(t))\nabla L(w(t))$, for simplicity, we will denote them us $fg(k)$ and $sg(k)$. Further, we will use the following denotation

$$w_{k,\tau} = (1 - \tau)w_t + \tau w_{k+1}$$

We will first prove some lemmas bounding the difference between gradient and projections at different points.

**Lemma C.18.** *Under Assumptions 1 and 2, when* $w_t \in V$*,* $\forall \tau \in (0,1)$*,* $w_{k,\tau} \in U$*.*

*Proof.* It holds that,

$$\|w_t - w_{k,\tau}\|_2 \leq \eta\Delta \leq \frac{\Delta}{2\gamma}.$$

$\qquad\square$

**Lemma C.19.** *Under Assumptions 1 and 2, when* $w_t \in V$*,* $\forall \tau, \tau' \in [0,1]$*, it holds that*
$$\|P_S(w_{k,\tau}) - P_S(w_{k,\tau'})\|_2 \leq \eta\gamma\kappa.$$

*Proof.* According to Lemma C.18, it holds that $w_{k,\tau}, w_{k,\tau'} \in U$. Assume without loss of generality $\tau > \tau'$,

$$\begin{aligned}
\|P_S(w_{k,\tau}) - P_S(w_{k,\tau'})\| &= \|\int_\tau^{\tau'} \nabla P_S(w_{k,\tau''})[\eta\nabla L(w)]d\tau''\|_2 \\
&\leq \int_{\tau'}^\tau \|\nabla P_S(w_{k,\tau''})[\eta\nabla L(w)]\|_2 d\tau'' \\
&\leq \eta\gamma\kappa.
\end{aligned}$$

$\qquad\square$

**Lemma C.20.** *Under Assumptions 1 and 2, when $w_t \in V$, $\forall \tau \in (0,1)$, it holds that*

$$\|P_S(w_{k,\tau})\nabla L(w_{k,\tau}) - P_S(w_{k,\tau})\nabla L(w_t)\|_2 \leq \eta\gamma_{\max}\|sg(k)\|_2 + 2\eta^2\gamma_{\max}\gamma\kappa\|\nabla L(w_t)\|_2.$$

*Proof.* According to Lemma C.18, it holds that $w_{k,\tau}, w_{k,\tau'} \in U$. Define $g(\tau') = \|P_S(w_{k,\tau})\nabla L(w_{k,\tau'}) - P_S(w_{k,\tau})\nabla L(w_t)\|_2$, then by Lagrange's Mean Value Theorem, there exists $\tau'$, such that

$$\|P_S(w_{k,\tau})\nabla L(w_{k,\tau}) - P_S(w_{k,\tau})\nabla L(w_t)\|_2 = g(\tau) - g(0)$$
$$=\tau g'(\tau') = \tau\frac{d\|P_S(w_{k,\tau})\nabla L(w_{k,\tau'}) - P_S(w_{k,\tau})\nabla L(w_t)\|_2}{d\tau'}$$
$$\leq\|\frac{dP_S(w_{k,\tau})\nabla L(w_{k,\tau'}) - P_S(w_{k,\tau})\nabla L(w_t)}{d\tau'}\|_2$$
$$=\eta\|P_S(w_{k,\tau})\nabla^2 L(w_{k,\tau'})\nabla L(w_t)\|_2$$

$$\leq\eta\|P_S(w_{k,\tau})\nabla^2 L(w_{k,\tau'})P_S(w_{k,\tau'})\nabla L(w_t)\|_2 + \eta\|P_S(w_{k,\tau})\nabla^2 L(w_{k,\tau'})P_F(w_{k,\tau'})\nabla L(w_t)\|_2$$
$$\leq\eta\gamma_{\max}\|P_S(w_{k,\tau'})\nabla L(w_t)\|_2 + \eta\|(P_S(w_{k,\tau}) - P_S(w_{k,\tau'}))\nabla^2 L(w_{k,\tau'})P_F(w_{k,\tau'})\nabla L(w_t)\|_2.$$

By Lemma C.19, it holds that

$$\gamma_{\max}\|P_S(w_{k,\tau'})\nabla L(w_t)\|_2 \leq \gamma_{\max}\|sg(k)\|_2 + \eta\gamma_{\max}\gamma\kappa\|\nabla L(w_t)\|_2.$$
$$\|(P_S(w_{k,\tau}) - P_S(w_{k,\tau'})\nabla^2 L(w_{k,\tau'})P_F(w_{k,\tau'})\nabla L(w_t)\|_2 \leq \eta\gamma\gamma_{\text{flat}}\kappa\|\nabla L(w_t)\|_2.$$

Summing up and the proof is complete. $\qquad\square$

**Lemma C.21.** $\forall \tau \in (0,1)$, *it holds that*

$$\|P_S(w_{k,\tau})\nabla L(w_{k,\tau}) - sg(k)\|_2 \leq \|sg(k)\|_2 + 3\eta\gamma\kappa\|\nabla L(w_t)\|_2.$$

*Proof.* This is a direct combination of Lemmas C.19 and C.20, with

$$\|P_S(w_{k,\tau})\nabla L(w_{k,\tau}) - P_S(w_t)\nabla L(w_t)\|_2$$
$$\leq\|P_S(w_{k,\tau})\nabla L(w_{k,\tau}) - P_S(w_{k,\tau})\nabla L(w_t)\|_2 + \|(P_S(w_{k,\tau}) - P_S(w_t))\nabla L(w_t)\|_2.$$

The proof is then complete. $\qquad\square$

**Lemma C.22.** $\forall \tau \in (0,1)$, *it holds that*

$$\|P_F(w_{k,\tau})\nabla L(w_{k,\tau}) - P_F(w_{k,\tau})\nabla L(w_t)\|_2 \leq \eta\gamma_{\text{flat}}\|fg(k)\|_2 + 2\eta^2\gamma_{\max}\gamma\kappa\|\nabla L(w_t)\|_2.$$

*Proof.* Define $g(\tau') = \|P_F(w_{k,\tau})\nabla L(w_{k,\tau'}) - P_F(w_{k,\tau})\nabla L(w_t)\|_2$, then by Lagrange's Mean Value Theorem, there exists $\tau'$, such that

$$\|P_F(w_{k,\tau})\nabla L(w_{k,\tau}) - P_F(w_{k,\tau})\nabla L(w_t)\|_2 = g(\tau) - g(0)$$
$$=\tau g'(\tau') = \tau\frac{d\|P_F(w_{k,\tau})\nabla L(w_{k,\tau'}) - P_F(w_{k,\tau})\nabla L(w_t)\|_2}{d\tau'}$$
$$\leq\|\frac{dP_F(w_{k,\tau})\nabla L(w_{k,\tau'}) - P_F(w_{k,\tau})\nabla L(w_t)}{d\tau'}\|_2$$
$$=\eta\|P_F(w_{k,\tau})\nabla^2 L(w_{k,\tau'})\nabla L(w_t)\|_2$$

$$\leq\eta\|P_F(w_{k,\tau})\nabla^2 L(w_{k,\tau'})P_F(w_{k,\tau'})\nabla L(w_t)\|_2 + \eta\|P_F(w_{k,\tau})\nabla^2 L(w_{k,\tau'})P_S(w_{k,\tau'})\nabla L(w_t)\|_2$$
$$\leq\eta\gamma_{\text{flat}}\|P_F(w_{k,\tau'})\nabla L(w_t)\|_2 + \eta\|(P_F(w_{k,\tau}) - P_F(w_{k,\tau'}))\nabla^2 L(w_{k,\tau'})P_S(w_{k,\tau'})\nabla L(w_t)\|_2.$$

By Lemma C.19, it holds that

$$\gamma_{\text{flat}}\|P_F(w_{k,\tau'})\nabla L(w_t)\|_2 \leq \gamma_{\text{flat}}\|fg(k)\|_2 + \eta\gamma_{\text{flat}}\gamma\kappa\|\nabla L(w_t)\|_2.$$
$$\|(P_F(w_{k,\tau}) - P_F(w_{k,\tau'})\nabla^2 L(w_{k,\tau'})P_F(w_{k,\tau'})\nabla L(w_t)\|_2 \leq \eta\gamma\gamma_{\max}\kappa\|\nabla L(w_t)\|_2.$$

Summing up and the proof is complete. $\qquad\square$

**Lemma C.23.** $\forall \tau \in (0,1)$, *it holds that*
$$\|P_F(w_{k,\tau})\nabla L(w_{k,\tau}) - fg(k)\|_2 \leq \|fg(k)\|_2 + 3\eta\gamma\kappa\|\nabla L(w_t)\|_2.$$

*Proof.* This is a direct combination of Lemmas C.19 and C.22, with
$$\|P_F(w_{k,\tau})\nabla L(w_{k,\tau}) - P_F(w_t)\nabla L(w_t)\|_2$$
$$\leq \|P_F(w_{k,\tau})\nabla L(w_{k,\tau}) - P_F(w_{k,\tau})\nabla L(w_t)\|_2 + \|(P_F(w_{k,\tau}) - P_F(w_t))\nabla L(w_t)\|_2.$$
The proof is then complete. $\qquad\square$

We will prove a discrete version of Lemma C.15.

**Lemma C.24.** *Under Assumptions 1 and 2, when $\eta < 1/\gamma_{\max}$, along the gradient flow Equation* (15)*, it holds that $\|P_S(w(t))\nabla L(w(t))\|_2 \leq 10\kappa\|P_F(w(t))\nabla L(w(t))\|_2$ as long as $w(\tau) \in U, \forall \tau \leq t$.*

*Proof.* We will first consider $sg(k)$, By Lagrange's Mean Value Theorem, there exists $\tau$, such that,
$$\|sg(k+1)\|_2^2 - \|sg(k)\|_2^2$$
$$= \|P_S(w_{k,1})\nabla L(w_{k,1})\|_2^2 - \|P_S(w_{k,0})\nabla L(w_{k,0})\|_2^2 = \frac{d\|P_S(w_{k,\tau})\nabla L(w_{k,\tau})\|_2^2}{d\tau}$$
$$= -\eta\langle P_S(w_{k,\tau})\nabla L(w_{k,\tau}), \partial P_S(w_{k,\tau})[\nabla L(w_t)]\nabla L(w_{k,\tau}) + P_S(w_{k,\tau})\nabla^2 L(w_{k,\tau})\nabla L(w)\rangle$$

The first term satisfies that
$$-\eta\langle P_S(w_{k,\tau})\nabla L(w_{k,\tau}), \partial P_S(w_{k,\tau})[\nabla L(w_t)]\nabla L(w_{k,\tau})\rangle$$
$$\leq \eta\gamma\kappa\|\nabla L(w_t)\|_2\|P_S(w_{k,\tau})\nabla L(w_{k,\tau})\|_2$$
$$\leq \eta\gamma\kappa\|\nabla L(w_t)\|_2(2\|sg(k)\|_2 + 3\eta\gamma\kappa\|\nabla L(w_t)\|_2).$$

The second term satisfies that
$$-\eta\langle P_S(w_{k,\tau})\nabla L(w_{k,\tau}), P_S(w_{k,\tau})\nabla^2 L(w_{k,\tau})\nabla L(w_t)\rangle$$
$$= -\eta\langle P_S(w_{k,\tau})\nabla L(w_t), P_S(w_{k,\tau})\nabla^2 L(w_{k,\tau})\nabla L(w_t)\rangle + \langle P_S(w_{k,\tau})(\nabla L(w_t) - \nabla L(w_{k,\tau})), \nabla^2 L(w_{k,\tau})\nabla L(w_t)\rangle$$
$$\leq -\eta(\gamma + 4\gamma_{\text{flat}})\|P_S(w_{k,\tau})\nabla L(w_t)\|_2^2 + \eta\gamma_{\max}\|P_S(w_{k,\tau})\nabla L(w_t)\|_2\|P_S(w_{k,\tau})(\nabla L(w_t) - \nabla L(w_{k,\tau}))\|_2$$

As we have that $\|a - b\|^2 \geq \frac{\|a\|^2}{2} - 4\|b\|^2$, by Lemma C.19, it holds that
$$-\|P_S(w_{k,\tau})\nabla L(w_t)\|_2^2$$
$$= -\|P_S(w_t)\nabla L(w_t) + (P_S(w_{k,\tau}) - P_S(w_t))\nabla L(w_t)\|_2^2$$
$$\leq -\eta\gamma\frac{\|P_S(w_t)\nabla L(w_t)\|_2^2}{2} + 4\eta\gamma\|(P_S(w_{k,\tau}) - P_S(w_t))\nabla L(w_t)\|_2^2$$
$$\leq -\frac{\|P_S(w_t)\nabla L(w_t)\|_2^2}{2} + 4\eta\gamma(\eta\gamma\kappa\|\nabla L(w_t)\|)^2.$$
$$= -\frac{\|P_S(w_t)\nabla L(w_t)\|_2^2}{2} + 4(\eta\gamma)^2(\kappa\|\nabla L(w_t)\|)^2.$$
$$\leq -\|sg(k)\|_2^2 + 2\eta\gamma\kappa^2\|\nabla L(w_t)\|^2$$

Hence
$$-\eta(\gamma + 4\gamma_{\text{flat}})\|P_S(w_{k,\tau})\nabla L(w_t)\|_2^2$$
$$\leq -\eta(\gamma + 4\gamma_{\text{flat}})\|sg(k)\|_2^2 + 2\eta^2(\gamma + 4\gamma_{\text{flat}})\gamma\kappa^2\|\nabla L(w_t)\|^2$$
$$\leq -\eta(\gamma + 4\gamma_{\text{flat}})\|sg(k)\|_2^2 + 2\eta\gamma\kappa^2\|\nabla L(w_t)\|^2$$

By Lemmas C.19 and C.20 and $\eta\gamma_{\max}^2 \leq \gamma/2$, it holds that
$$\eta\gamma_{\max}\|P_S(w_{k,\tau})\nabla L(w_t)\|_2\|P_S(w_{k,\tau})(\nabla L(w_t) - \nabla L(w_{k,\tau}))\|_2$$
$$\leq \eta^2\gamma_{\max}^2(\|sg(k)\|_2 + \eta\gamma\kappa\|\nabla L(w_t)\|_2)(\|sg(k)\|_2 + 2\eta\gamma\kappa\|\nabla L(w_t)\|_2)$$
$$\leq \eta\gamma(\|sg(k)\|_2 + \kappa\|\nabla L(w_t)\|_2/2)(\|sg(k)\|_2 + \kappa\|\nabla L(w_t)\|_2)/2.$$

Hence, we can conclude that

$$\|sg(k+1)\|_2^2 - \|sg(k)\|_2^2$$
$$- \eta((\gamma + 4\gamma_{\text{flat}})\|sg(k)\|_2^2 + 4\eta\gamma\kappa^2\|\nabla L(w_t)\|^2$$
$$+ \eta\gamma(\|sg(k)\|_2 + \kappa\|\nabla L(w_t)\|_2/2)(\|sg(k)\|_2 + \kappa\|\nabla L(w_t)\|_2)/2$$

Let $b = \kappa\|\nabla L(w_t)\|_2$ and $a = sg(k)$, as $b(2a + 3b/2) - a^2 + 4b^2 + \frac{1}{2}(a + \frac{b}{2})(a+b) \leq -\frac{a^2}{4} + 10b^2$, it holds that

$$\|sg(k+1)\|_2^2 - \|sg(k)\|_2^2 \leq -\eta\gamma\frac{\|sg(k)\|_2^2}{4} + 10\eta\gamma\kappa^2\|\nabla L(w_t)\|^2. - 4\eta\gamma_{\text{flat}}\|sg(k)\|_2^2 \quad (16)$$

Similarly, we can control the $fg(k)$ changes. By Lagrange's Mean Value Theorem, there exists $\tau'$, such that,

$$\|fg(k+1)\|_2^2 - \|fg(k)\|_2^2$$
$$= \|P_F(w_{k,1})\nabla L(w_{k,1})\|_2^2 - \|P_F(w_{k,0})\nabla L(w_{k,0})\|_2^2 = \frac{d\|P_F(w_{k,\tau})\nabla L(w_{k,\tau})\|_2^2}{d\tau}$$
$$= -\eta\langle P_F(w_{k,\tau})\nabla L(w_{k,\tau}), \partial P_F(w_{k,\tau})[\nabla L(w_t)]\nabla L(w_{k,\tau}) + P_F(w_{k,\tau})\nabla^2 L(w_{k,\tau})\nabla L(w)\rangle$$

The first term satisfies that

$$\eta\langle P_F(w_{k,\tau})\nabla L(w_{k,\tau}), \partial P_F(w_{k,\tau})[\nabla L(w_t)]\nabla L(w_{k,\tau})\rangle$$
$$\leq \gamma\eta\kappa\|\nabla L(w_t)\|_2\|P_F(w_{k,\tau})\nabla L(w_{k,\tau})\|_2.$$
$$\leq \gamma\eta\kappa\|\nabla L(w_t)\|_2(\|fg(k)\|_2 + \eta\gamma_{\text{flat}}\|fg(k)\|_2 + 2\eta^2\gamma_{\max}\gamma\kappa\|\nabla L(w_t)\|_2).$$
$$\leq 4\gamma\eta\kappa\|\nabla L(w_t)\|_2^2.$$

Similarly, the second term satisfies that

$$\eta\langle P_F(w_{k,\tau})\nabla L(w_{k,\tau}), P_F(w_{k,\tau})\nabla^2 L(w_{k,\tau})\nabla L(w)\rangle$$
$$\leq \eta\gamma_{\text{flat}}\|P_F(w_{k,\tau})\nabla L(w_t)\|_2\|P_F(w_{k,\tau})\nabla L(w_{k,\tau})\|_2$$
$$\leq \eta\gamma_{\text{flat}}(\|fg(k)\|_2 + \eta\gamma\kappa\|\nabla L(w_t)\|_2)(\|fg(k)\|_2 + \eta\gamma_{\text{flat}}\|fg(k)\|_2 + 2\eta^2\gamma_{\max}\gamma\kappa\|\nabla L(w_t)\|_2)$$
$$\leq 2\eta\gamma_{\text{flat}}(\|fg(k)\|_2 + \eta\gamma\kappa\|\nabla L(w_t)\|_2)^2$$
$$\leq 4\eta\gamma_{\text{flat}}\|fg(k)\|_2^2 + 4\eta^2\gamma^2\kappa^2\|\nabla L(w_t)\|_2^2$$

Summarizing and we have

$$\|fg(k+1)\|_2^2 - \|fg(k)\|_2^2 \geq -5\gamma\eta\kappa\|\nabla L(w_t)\|_2^2 - 4\eta\gamma_{\text{flat}}\|fg(k)\|_2^2 \quad (17)$$

Let $a_\kappa$ be the smaller positive solution of

$$5\kappa a^2 + (10\kappa^2 + 5\kappa - \frac{1}{4})a + 10\kappa^2 = 0.$$

Then $a_\kappa = \frac{(-10\kappa^2 - 5\kappa + \frac{1}{4}) - \sqrt{(-10\kappa^2 - 5\kappa + \frac{1}{4})^2 - 200\kappa^3}}{10\kappa} < 100\kappa^2$.

Then combining Equations (16) and (17)

$$\|sg(k+1)\|_2^2 - a_\kappa\|fg(k+1)\|_2^2$$

$$\leq (1 - 4\eta\gamma_{\text{flat}})(\|sg(k)\|_2^2 - a_\kappa\|fg(k+1)\|_2^2) - \eta\gamma(\frac{1}{4} + 10\kappa^2 - 5\kappa a_\kappa)\|sg(k)_2\|^2 + \eta\gamma(10\kappa^2 + 5\kappa a_\kappa)\|fg(k)_2\|^2$$

$$= (1 - 4\eta\gamma_{\text{flat}} - \eta\gamma(\frac{1}{4} + 10\kappa^2 - 5\kappa a_\kappa))(\|sg(k)\|_2^2 - a_\kappa\|fg(k)\|_2^2).$$

As $\|sg(0)\|_2^2 - a_\kappa\|fg(0)\|_2^2 < 0$, we have that $\|sg(k)\|_2^2 < a_\kappa\|fg(k+1)\|_2^2 < 100\kappa^2\|fg(k)\|_2^2$ for all the $t$. $\qquad\square$

Then we can show that gradient descent will also track the river closely.

**Lemma C.25.** *Under Assumptions 1 and 2, along the gradient flow Equation* (12)*, it holds that* $w(t) \in U$ *and* $\|w(t) - \Phi(w(t))\|_2 \leq \frac{10\kappa\|P_F(w(t))\nabla L(w(t))\|_2}{\gamma + \gamma_{\text{flat}}}$.

*Proof.* This is a direct combination of Lemmas C.4 and C.24. $\square$

Finally, we will show that the movement of the projection of the gradient flow moves approximately at the same rate as the river, a discrete version of Lemma C.16.

**Lemma C.26.** *Under Assumptions 1 and 2, along the gradient flow Equation* (12)*, along the gradient descent Equation* (15)*, if* $T(t)$ *satisfies* $x(T(t)) = \Phi(w_{[t], t-[t]})$ *where* $[t]$ *is the integer part of* $t$*, then for any* $t$ *that is not integer*

$$\frac{dT(t)}{dt} \in [\eta - (30\kappa + 4\eta\gamma_{\text{flat}})\eta, \eta + (30\kappa + 4\eta\gamma_{\text{flat}})\eta].$$

*Proof.* Let $[t] = k, t - [t] = \tau$, as $T(t)$ satisfies $x(T(t)) = \Phi(w(k, t - [t]))$, let $v = frac{dx(\tau)}{d\tau}|_{\tau=T(t)}$, taking derivative on both sides yield,

$$\frac{dT(t)}{dt}v = -\eta\partial\Phi(w_{k,\tau})\nabla L(w_k).$$

As the proof of Lemma C.22, there exists $\tau'$

$$\|P_F(w_{k,\tau})\nabla L(w_k) - P_F(w_{k,\tau})\nabla L(w_{k,\tau})\|_2$$
$$\leq \eta\|P_F(w_{k,\tau})\nabla^2 L(w_{k,\tau'})P_F(w_{k,\tau'})\nabla L(w_k)\|_2 + \eta\|P_F(w_{k,\tau})\nabla^2 L(w_{k,\tau'})P_S(w_{k,\tau'})\nabla L(w_k)\|_2$$
$$\leq \eta\gamma_{\text{flat}}\|P_F(w_{k,\tau'})\nabla L(w_k)\|_2 + \eta^2\gamma\kappa\gamma_{\max}\|\nabla L(w_k)\|_2$$

By Lemma C.19, it holds that

$$\|P_F(w_{k,\tau})\nabla L(w_k) - P_F(w_{k,\tau})\nabla L(w_{k,\tau})\|_2$$
$$\leq \eta\gamma_{\text{flat}}\|P_F(w_{k,\tau})\nabla L(w_k)\|_2 + 2\eta^2\gamma\gamma_{\max}\kappa\|\nabla L(w_k)\|_2$$
$$\leq \eta\gamma_{\text{flat}}\|P_F(w_{k,\tau})\nabla L(w_k)\|_2 + \kappa\|\nabla L(w_k)\|_2 \qquad (18)$$

By Lemma C.24,

$$\|\nabla L(w_k)\|_2 \leq \frac{1}{1 - 10\kappa}\|P_F(w_k)\nabla L(w_k)\|_2$$
$$\leq \frac{1}{1 - 10\kappa}\left(\|P_F(w_{k,\tau})\nabla L(w_k)\|_2 + \eta\gamma\kappa\|\nabla L(w_k)\|_2\right)$$

This shows that

$$\|\nabla L(w_k)\|_2 \leq \frac{1}{1 - 10\kappa - \eta\gamma\kappa}\|P_F(w_{k,\tau})\nabla L(w_k)\|_2 \leq (1 + 12\kappa)\|P_F(w_{k,\tau})\nabla L(w_k)\|_2$$

Combining with Equation (18), we have that

$$\|P_F(w_{k,\tau})\nabla L(w_k) - P_F(w_{k,\tau})\nabla L(w_{k,\tau})\|_2$$
$$\leq \eta\gamma_{\text{flat}}\|P_F(w_{k,\tau})\nabla L(w_k)\|_2 + \kappa(1 + 12\kappa)\|P_F(w_{k,\tau})\nabla L(w_k)\|_2$$
$$\leq (\eta\gamma_{\text{flat}} + 2\kappa)\|P_F(w_{k,\tau})\nabla L(w_k)\|_2$$

This shows that

$$\|P_F(w_{k,\tau})\nabla L(w_k) - P_F(w_{k,\tau})\nabla L(w_{k,\tau})\|_2 \leq \frac{(\eta\gamma_{\text{flat}} + 2\kappa)}{1 - (\eta\gamma_{\text{flat}} + 2\kappa)}\|P_F(w_{k,\tau})\nabla L(w_{k,\tau})\|_2$$
$$\leq (2\eta\gamma_{\text{flat}} + 3\kappa)\|P_F(w_{k,\tau})\nabla L(w_k)\|_2 \qquad (19)$$

By Lemma C.12

$$\|P_F(w_{k,\tau})\nabla L(w_{k,\tau}) + v\|_2 \leq 16\kappa\|v\|_2 \qquad (20)$$

Combining Equations (19) and (20),

$$\|P_F(w_{k,\tau})\nabla L(w_k) + v\|_2$$
$$\leq\|P_F(w_{k,\tau})\nabla L(w_k) - -P_F(w_{k,\tau})\nabla L(w_{k,\tau})\|_2 + \|P_F(w_{k,\tau})\nabla L(w_{k,\tau}) + v\|_2$$
$$\leq(2\eta\gamma_{\text{flat}} + 3\kappa)\|P_F(w_{k,\tau})\nabla L(w_k)\|_2 + 16\kappa\|v\|_2$$
$$\leq((2\eta\gamma_{\text{flat}} + 3\kappa)(1 + 16\kappa) + 16\kappa)\|v\|_2$$
$$\leq(19\kappa + 3\eta\gamma_{\text{flat}})\|v\|_2. \tag{21}$$

By Lemma C.9

$$\|\partial\Phi(w_{k,\tau})\nabla L(w_k) - P_F(w_{k,\tau})\nabla L(w_k)\|_2 \leq 10\kappa\|P_F(w_{k,\tau})\nabla L(w_k)\|_2. \tag{22}$$

Combining Equations (21) and (22), it holds that

$$\|\partial\Phi(w_{k,\tau})\nabla L(w_k) + v\|_2$$
$$\leq\|\partial\Phi(w_{k,\tau})\nabla L(w_k) - P_F(w_{k,\tau})\nabla L(w_k)\|_2 + \|P_F(w_{k,\tau})\nabla L(w_k) + v\|_2$$
$$\leq10\kappa\|P_F(w_{k,\tau})\nabla L(w_k)\|_2 + (19\kappa + 3\eta\gamma_{\text{flat}})\|v\|_2$$
$$\leq(30\kappa + 4\eta\gamma_{\text{flat}})\|v\|_2.$$

Hence

$$\frac{dT(t)}{dt} \in [\eta - (30\kappa + 4\eta\gamma_{\text{flat}})\eta, \eta + (30\kappa + 4\eta\gamma_{\text{flat}})\eta].$$

This concludes the proof. □

*Proof of Theorem 3.3.* The proof is a direct combination of Lemmas C.25 and C.26. □

## C.7 PROOF OF THEOREM 3.4

**Assumption 5** (Regularity Assumption for SGD). *In the setting of Assumptions 2, we assume in addition the following:*

1. *Bounded Hessian. There exists a constant $\tau > 0$, such that for any weight $w \in U$, the nuclear norm of the Hessian is bounded. $\|\nabla^2 L(w)\|_* = \sum_{i=1}^{d} |\lambda_i(\nabla^2 L(w))| \leq \tau$.*
2. *Bounded Third Order Information. There exist constants $\rho > 0, \kappa' \in [0, 0.01]$, such that $\|\nabla^3 L(w)\|_{\text{op}} \leq \rho, \Delta\rho \leq \kappa'\gamma^2$.*
3. *Bounded Loss. There exists a constant $M > 0$ such that $\forall w, L(w) < M$.*

In this assumption, we treat $\kappa'$ as a small constant, indicating that the influence of the third-order gradient is minimal. This suggests that the overall shape of the loss landscape is predominantly governed by the first and second-order information.

The error term in loss term $\epsilon_L$ satisfies that $|\epsilon_L| \leq \tau\eta^2\sigma^2 + \rho(Cd\eta\sigma^2/\gamma)^{3/2} + C\kappa'd\eta\sigma^2 + \delta(2M + \eta\sigma^2 d) \ll (d-1)\eta\sigma^2$ with $C = 200\log(64\gamma T/\delta)$ and the error can be decomposed into three parts: (1) $\tau\eta^2\sigma^2 + \rho(C\eta\sigma^2/\gamma)^{3/2}$ are higher order discretization effects of learning rate $\eta$; (2) $C\kappa'd\eta\sigma^2$ is caused by the change of the Hessian in the valley dimensions and will diminish when $\kappa'$ is small; (3) $\delta(2M + \eta\sigma^2 d)$ accounts for the small chances that the iterate will escape the neighborhood of the river due to the stochastic updates. While the theorem only considers the case where $v_d$ is a constant vector,

We will first show that under Assumption 3, the loss is separable within $U$.

**Lemma C.27.** *Under Assumptions 1 to 3, the river is a straight line parallel to $v_d$.*

*Proof.* In this case, the $\kappa$ in Assumption 2 is 0 and this is a direct corollary of of Lemma C.12. □

**Lemma C.28.** *Under Assumptions 1 to 3, there exists functions $g$ and $h$, such that for any $w \in U$ satisfying that $\mathcal{B}(w, \frac{2\Delta}{\gamma}) \subset U$, it holds that*

$$L(w) = g(\Phi(w)) + h(w - \Phi(w)).$$

*Furthermore, $h$ is a $\gamma$-strongly convex function when constrained on the range of $P_S$.*

*Proof.* We will choose $g$ as the constraint of $L$ on $\mathcal{M}$. Now $w - \Phi(w)$ will always fall in the range of $P_S$. Consider any $y$ in the range of $P_S$ and as $\nabla v_d(w)[v] = 0$, we have that

$$y^T \nabla^2 L(w) v_d = 0.$$

This then suggest that

$$\nabla[\langle \nabla L(w), y \rangle][v_d] = 0.$$

We then have for any $a \in \mathcal{M}$, by Lemma C.27,

$$L(w) - L(\Phi(w)) = \int_0^1 \langle (w - \Phi(w)), \nabla L(\Phi(w) + \tau(w - \Phi(w))) \rangle d\tau$$

$$= \int_0^1 \langle (w - \Phi(w)), \nabla L(a + \tau(w - \Phi(w))) \rangle d\tau.$$

We will then define $h(w - \Phi(w)) = \int_0^1 \langle (w - \Phi(w)), \nabla L(a + \tau(w - \Phi(w))) \rangle d\tau$. and this concludes the proof.

Now, as $h(w - \Phi(w)) = L(w) - L(\Phi(w))$, $\nabla^2 h(y)$ when constrained on the range of $P_S$ has an eigenvalue greater than $\gamma$. $\qquad\square$

We will first consider the mixing dynamics of the current SGD iterates on a strongly convex loss $h$ with a minimizer at $0$.

$$y(k+1) = y_k - \eta \nabla h(y_k) - \eta g_k, y(0) = 0, \mathbf{g_k} \sim \mathcal{N}(0, \sigma^2 \mathcal{I}) \tag{23}$$

We will define a coupling process $\tilde{y}_k$ as

$$\tilde{y}(k+1) = \tilde{y}_k - \eta H \tilde{y}_k - \eta g_k, w(0) = 0, g_k \sim \mathcal{N}(0, \sigma^2 \mathcal{I}), \tilde{y}(0) = 0. \tag{24}$$

Here $H = \nabla^2 h(0)$ is positive definite.

**Assumption 6** (Regularity of $h$). *We will assume the following for the function $h$, constant $\delta \in (0,1]$, learning rate $\eta$.*

1. *The smallest eigenvalue of $\nabla^2 h(y)$ within $\mathcal{B}(0,r)$ is at least $\gamma > 0$ and the largest eigenvalue for $H$ is at most $\gamma_{\max}$.*
2. $\forall y, h(y) \in [0, M]$.
3. $\forall y \in \mathcal{B}(0,r), \|\nabla h(y)\|_2 \leq \Delta, \|\nabla^3 h(y)\|_2 \leq \rho$.
4. $T > 1/\gamma$.
5. $\eta < 1/(2\gamma_{\max})$.
6. $\eta \rho^2 \sigma^2 \leq \gamma^3/(1600 d \log(8\gamma T/\delta))$.
7. $10 \frac{\sqrt{\eta}\sigma}{\sqrt{\gamma}} \sqrt{d \log(8\gamma T/\delta)} + 400 \eta \rho \sigma^2 d \log(8\gamma T/\delta)/\gamma^2 \leq r$.

We will first show that $\tilde{y}_k$ will be bounded with a high probability for $T/\eta$ steps.

**Lemma C.29.** *For any $\delta \in (0,1]$, with probability $1 - \delta$, for $\tilde{y}_k$ defined in Equation (24), under Assumption 6, it holds that for any $k \leq T/\eta$,*

$$\|\tilde{y}_k\|_2 \leq \frac{10\sqrt{\eta}\sigma}{\sqrt{\gamma}} \sqrt{d \log(8\gamma T/\delta)}.$$

*Proof.* For integer $K = \lceil \gamma T \rceil$. We first have that for $k \leq K$

$$\tilde{y}_{k\lceil \frac{1}{\eta\gamma} \rceil} = (1 - \eta\gamma)^{\lceil \frac{1}{\eta\gamma} \rceil} \tilde{y}_{(k-1)\lceil \frac{1}{\eta\gamma} \rceil} + \eta \sum_{\tau=0}^{\lceil \frac{1}{\eta\gamma} \rceil} (1 - \eta\gamma)^{\lceil \frac{1}{\eta\gamma} \rceil - t} g_{(k-1)\lceil \frac{1}{\eta\gamma} \rceil + \tau}.$$

Denote $\bar{g}_k = \eta \sum_{\tau=0}^{\lceil \frac{1}{\eta\gamma} \rceil} (1 - \eta\gamma)^{\lceil \frac{1}{\eta\gamma} \rceil - t} g_{(k-1)\lceil \frac{1}{\eta\gamma} \rceil + \tau}$, then $g_k$ is a normal vector with variance

$$\eta^2 \sum_{\tau=0}^{\lceil \frac{1}{\eta\gamma} \rceil} (1 - \eta\gamma)^{2(\lceil \frac{1}{\eta\gamma} \rceil - t)} \sigma^2 \mathcal{I} \leq \frac{\eta\sigma^2}{2\gamma - \eta\gamma^2} \leq \frac{\eta\sigma^2}{\gamma}.$$

Further, denote $Y_k = y_{k\lceil \frac{1}{\eta\gamma}\rceil}$ and $e_\gamma = (1-\eta\gamma)^{\lceil\frac{1}{\eta\gamma}\rceil} < \frac{1}{e}$, then

$$Y_k = e_\gamma Y_{k-1} + \bar{g}_{k-1} = \sum_{i \leq k-1} e_\gamma^{i-1} g_{k-i}$$

Then each variable $Y_k$ is also a Gaussian variable with variance smaller than

$$\sum_{i \leq k-1} e_\gamma^{2(i-1)} \mathbb{E}[\bar{g}_{k-i}\bar{g}_{k-i}^T] \leq \frac{1}{1-1/e^2}\frac{\eta\sigma^2}{\gamma}\mathcal{I} \leq \frac{2\eta\sigma^2}{\gamma}\mathcal{I}.$$

Hence, by Lemma C.37, for each $k$, it holds that

$$\mathbb{P}(\left|Y_k\right| > \frac{2\sqrt{\eta}\sigma}{\sqrt{\gamma}}\sqrt{d\log(4K/\delta)})) < \delta/2K.$$

Using union bound,

$$\mathbb{P}(\exists k \leq K, \left|Y_k\right| > \frac{2\sqrt{\eta}\sigma}{\sqrt{\gamma}}\sqrt{d\log(4K/\delta)}) < \delta/2.$$

We now proceed to bound the distance of $y_k$ compared with close $Y_k$, without loss of generality, considering $k = 0$, we will define a new process called $m_k$ satisfying that

$$m_k = \sum_{k \leq t}(1-\eta\gamma)^{\lceil\frac{1}{\eta\gamma}\rceil-k} g_k.$$

Then $m_k$ is a martingale and each $m_k$ is a Gaussian vector. In particular, $m_{\lceil\frac{1}{\eta\gamma}\rceil} = \bar{g}_1$. This further suggests that $\|m_k\|^2$ is a super martingale

$$\mathbb{E}[\|m_k\|_2^2 \mid m_{k-1}] \geq \|m_{k-1}\|_2^2.$$

By Doob's lemma (Lemma C.38)

$$\mathbb{P}(\sup_{k \leq \lceil\frac{1}{\eta\gamma}\rceil}\|m_k\|_2^2 > C^2) \leq \mathbb{P}(\sup_{k \leq \lceil\frac{1}{\eta\gamma}\rceil}\exp(\lambda\|m_k\|_2^2) > \exp(\lambda C^2))$$

$$\leq \mathbb{E}[\exp(\lambda\|m_{\lceil\frac{1}{\eta\gamma}\rceil}\|_2^2 - \lambda C^2)]$$

$$= \mathbb{E}[\exp(\lambda\|\bar{g}_1\|_2^2 - \lambda C^2)].$$

Following the same line of proof as Lemma C.37, we have that

$$\mathbb{P}(\sup_{k \leq \lceil\frac{1}{\eta\gamma}\rceil}\|m_k\|_2 > \frac{2\sqrt{\eta}\sigma}{\sqrt{\gamma}}\sqrt{d\log(4K/\delta)}) \leq \delta/2K$$

We further note that $\left|y_k - Y_0\right| \leq (1-\eta\gamma)^{-\lceil\frac{1}{\eta\gamma}\rceil}m_k \leq 4m_k$. We have that for any $k < K$

$$\mathbb{P}(\sup_{k \leq \lceil\frac{1}{\eta\gamma}\rceil}\|y_k - Y_0\|_2 > \frac{8\sqrt{\eta}\sigma}{\sqrt{\gamma}}\sqrt{d\log(4K/\delta)}) \leq \delta/2K$$

Combining with the bound on $Y_k$, we have that

$$\mathbb{P}(\sup_{0 \leq t \leq T}|y_k| > \frac{10\sqrt{\eta}\sigma}{\sqrt{\gamma}}\sqrt{d\log(8\gamma T/\delta)})) \leq \delta.$$

The proof is then complete. $\qquad\square$

The following lemma states that $y_k$ and $\tilde{y}_k$ are close with high probability.

**Lemma C.30.** *Assume function $h(y)$ is $\gamma$-strong convex in $\mathcal{B}(0, r)$ and has a minimizer at $0$,then for $\delta \in (0, 1)$, under Assumption 6, it holds that with probability $1 - \delta$,*

$$\forall k < T/\eta, \|\tilde{y}_k - y_k\|_2 \leq 400\eta\rho\sigma^2 d\log(8\gamma T/\delta)/\gamma^2, y_k \in \mathcal{B}(0, r), \tilde{y}_k \in \mathcal{B}(0, r)$$

*Proof.* By Lemma C.29, with probability $1 - \delta$,

$$\forall k < T/\eta, \|\tilde{y}_k\|_2^2 \leq \frac{100\eta\sigma^2}{\gamma}d\log(8\gamma T/\delta)$$

We will use $C$ as a shorthand for $100d\log(8\gamma T/\delta)$. Under such scenario, define $\nu_k = \tilde{y}_k - y_k$, we will prove by induction for $k \leq T/\eta$ that

$$\|\nu_k\|_2 \leq 4\eta\rho\sigma^2 C/\gamma^2, y_k \in \mathcal{B}(0, r). \tag{25}$$

Clearly $\nu_0 = 0$, satisfies the induction hypothesis. Assuming Equation (25) hold for $t$, then

$$\begin{aligned} y(k+1) &= y_k - \eta\nabla L(y_k) - \eta g_k \\ &= y_k - \eta\nabla^2 L(0)y_k + e_k - \eta g_k \\ &= \tilde{y}_k(1 - \eta\nabla^2 L(0)) - \eta g_k + \nu_k(1 - \eta\nabla^2 L(0)) + e_k. \end{aligned}$$

Here $\|e_k\| = \| - \eta(\nabla L(y_k) - \eta\nabla^2 L(0)y_k)\|_2 \leq \eta\rho\|y_k\|_2^2 \leq 2\eta\rho(\|\tilde{y}_k\|_2^2 + \|\nu_k\|_2^2)$. Hence we have that

$$\|\nu_{k+1}\|_2 \leq (1 - \eta\gamma)\|\nu_k\|_2 + 2\eta\rho(\|\tilde{y}_k\|_2^2 + \|\nu_k\|_2^2).$$

As $\|\nu_k\| \leq 4\frac{\eta\rho\sigma^2 C}{\gamma^2} \leq \frac{\gamma}{4\rho}$, we have that $2\rho\|\nu_k\|_2^2 \leq \eta\gamma\|\nu_k\|_2^2/2$.

Hence

$$\begin{aligned} \|\nu_{k+1}\|_2 &\leq (1 - \eta\gamma/2)\|\nu_k\|_2 + 2\eta\rho\frac{\eta\sigma^2 C}{\gamma} \\ &= (1 - \eta\gamma/2)\|\nu_k\|_2 + 2\frac{\eta^2\rho\sigma^2 C}{\gamma} \end{aligned}$$

By induction $\|\nu_k\|_2 \leq 4\eta\rho\sigma^2 C/\gamma^2$. It is then easy to check $\|\nu_{k+1}\|_2 \leq 4\eta\rho\sigma^2 C/\gamma^2$. $\qquad\square$

The following lemma tracks the changes of $\mathbb{E}[h(y_k)]$.

**Lemma C.31.** *Assume function $h(y)$ is $\gamma$-strong convex in $\mathcal{B}(0, r)$ and has a minimizer at $0$,then for $\delta \in (0, 1)$, denote $100\log(8\gamma T/\delta)$ as $C$, under Assumption 6, it holds that $\forall t \in [1/\eta\gamma, T/\eta]$,*

$$\left|\mathbb{E}[h(\tilde{y}_k)] - \eta\sigma^2 d/2\right| \leq \eta^2\sigma^2\text{Tr}(H) + \frac{\Delta\rho C}{\gamma^2}d\eta\sigma^2 + \rho\left(\frac{d\eta\sigma^2 C}{\gamma}\right)^{3/2} + 2\delta M + \delta\eta\sigma^2 d/2$$

*Proof.* By Lemma C.30, with probability $1 - \delta$,

$$\|\tilde{y}_k - y_k\|_2 \leq 4\eta\rho d\sigma^2 C/\gamma^2, y_k \in \mathcal{B}(0, r), \tilde{y}_k \in \mathcal{B}(0, r).$$

Define this event as $\mathcal{E}_1$.

Hence

$$\begin{aligned} &\left|\mathbb{E}[h(y_k)] - \mathbb{E}[h(\tilde{y}_k)]\right| \\ &\leq \left|\mathbb{E}[h(\tilde{y}_k) - h(y_k) \mid \mathcal{E}_1]\mathbb{P}(\mathcal{E}_1) + \mathbb{E}[h(\tilde{y}_k) - h(y_k) \mid \mathcal{E}_1^c]\mathbb{P}(\mathcal{E}_1^c)\right| \\ &\leq 4\frac{\Delta\rho C}{\gamma^2}\eta d\sigma^2 + \delta M. \end{aligned}$$

For $\|y\|_2 < r$, it holds that

$$\|h(y) - y^T \nabla^2 h(0)y\|_2 \leq \rho \|y\|_2^3.$$

By Lemma C.29, with probability $1 - \delta$, for $\eta < 1/\gamma_{\max}$, it holds that for any $k \leq T/\eta$,

$$\|\tilde{y}_k\|_2^2 \leq \frac{d\eta\sigma^2 C}{\gamma} < r^2.$$

Define this event as $\mathcal{E}_2$.

Denote $H = \nabla^2 h(0)$, we have that,

$$\left| \mathbb{E}[h(\tilde{y}_k)] - \mathbb{E}[(\tilde{y}_k)^T H \tilde{y}_k] \right|$$

$$= \left| \mathbb{E}[h(\tilde{y}_k) \mid \mathcal{E}_2]\mathbb{P}(\mathcal{E}_2) - \mathbb{E}[(\tilde{y}_k)^T H \tilde{y}_k] + \mathbb{E}[h(\tilde{y}_k) \mid \mathcal{E}_2^c]\mathbb{P}(\mathcal{E}_2^c) \right|$$

$$\leq \delta M + \left| \mathbb{E}[h(\tilde{y}_k) \mid \mathcal{E}_2]\mathbb{P}(\mathcal{E}_2) - \mathbb{E}[(\tilde{y}_k)^T H \tilde{y}_k] \right|$$

$$\leq \delta M + \rho \left( \frac{\eta d\sigma^2 C}{\gamma} \right)^{3/2} + \mathbb{E}[(\tilde{y}_k)^T H \tilde{y}_k \mid \mathcal{E}_2^c]$$

Combining the both and we have that

$$\left| \mathbb{E}[h(y_k)] - \mathbb{E}[(\tilde{y}_k)^T H \tilde{y}_k] \right| \leq 4\frac{\Delta\rho C}{\gamma^2}\eta d\sigma^2 + \rho \left( \frac{\eta d\sigma^2 C}{\gamma} \right)^{3/2} + 2\delta M + \mathbb{E}[(\tilde{y}_k)^T H \tilde{y}_k \mid \mathcal{E}_2^c].$$

Here the covariance of $\tilde{y}_k$, denoted as $\Sigma_k$ satisfies that

$$\Sigma_{k+1} = (\mathcal{I} - \eta H)^2 \Sigma_k + \sigma^2 \eta^2 \mathcal{I}.$$

Therefore

$$\Sigma_k - \eta\sigma^2(2\eta H - \eta^2 H^2)^{-1} = (\mathcal{I} - \eta H)^2(\Sigma_{k-1} - \eta\sigma^2(2\eta H - \eta^2 H^2)^{-1}).$$

$$\Sigma_k = \sigma^2(2H - \eta H^2)^{-1}(\mathcal{I} - (\mathcal{I} - \eta H)^{2k}).$$

Hence assuming the eigenvalues of $H$ is $\gamma_1, \ldots, \gamma_d$

$$\mathbb{E}[(\tilde{y}_k)^T H \tilde{y}_k] = \text{Tr}(\Sigma_k H) = \eta\sigma^2 \sum_{i=1}^d \frac{1}{2 - \eta\gamma_i}(1 - (1 - \eta\gamma_i)^{2k}).$$

When $t \geq \frac{1}{\eta\gamma_i}, \eta\gamma_i < 1/2$, it holds that

$$\left| \eta\sigma^2 \frac{1}{2 - \eta\gamma_i}(1 - (1 - \eta\gamma_i)^{2k}) - \eta\sigma^2/2 \right|$$

$$= \eta\sigma^2 \frac{(1 - (1 - \eta\gamma_i)^{2k})}{2 - \eta\gamma_i} - \eta\sigma^2/2$$

$$\leq \eta\sigma^2 \left( \frac{1}{2 - \eta\gamma_i} - \frac{1}{2} \right)$$

$$\leq \eta^2\sigma^2\gamma_i/2.$$

Hence

$$\left| \mathbb{E}[(\tilde{y}_k)^T H \tilde{y}_k] - d\eta\sigma^2/2 \right| \leq \text{Tr}(H)\eta^2\sigma^2/2.$$

Further, let $u_k = \Sigma_k^{-1/2}\tilde{y}_k$, under $\mathcal{E}_2^c$, we have that

$$\|u_k\|_2^2 \geq \lambda_{min}(\Sigma_k^{-1})\|\tilde{y}_k\|_2^2 = \lambda_{min}\left( (2H - \eta H^2)(\mathcal{I} - (\mathcal{I} - \eta H)^{2k})^{-1} \right)\|\tilde{y}_k\|_2^2/\sigma^2 \geq d\sigma^2 C.$$

As $u_k$ is isometric Gaussian,

$$
\begin{aligned}
\mathbb{E}[(\tilde{y}_k)^T H \tilde{y}_k \mid \mathcal{E}_2^c] &\leq \mathbb{E}[u_k^T (\Sigma_k^{1/2})^T H \Sigma_k^{1/2} u_k \mid \|u_k\|_2^2 \geq d\sigma^2 C] \\
&= \mathbb{E}[u_k^T (\Sigma_k^{1/2})^T H \Sigma_k^{1/2} u_k] \frac{\mathbb{E}[\|u_k\|_2^2 \mid \|u_k\|_2^2 \geq d\sigma^2 C]}{\mathbb{E}[\|u_k\|^2]} \\
&\leq d\eta\sigma^2 \frac{\mathbb{E}[\|u_k\|_2^2 \mid \|u_k\|_2^2 \geq d\sigma^2 C]}{\mathbb{E}[\|u_k\|^2]}
\end{aligned}
$$

Plugging in the density function of $\|u_k\|_2$, we have that

$$
\frac{\mathbb{E}[\|u_k\|_2^2 \mid \|u_k\|_2^2 \geq d\sigma^2 C]}{\mathbb{E}[\|u_k\|_2^2]} = \frac{\int_{\sqrt{dC}\sigma}^{\infty} r^{d+1} e^{-r^2/(2\sigma^2)} dr}{\int_0^{\infty} r^{d+1} e^{-r^2/(2\sigma^2)} dr}
$$

Let $r' = \sqrt{\frac{d}{d+1}} r$, then

$$
\begin{aligned}
\int_{\sqrt{dC}\sigma}^{\infty} r^{d+1} e^{-r^2/\sigma^2} dr &= \left(\frac{d+1}{d}\right)^{\frac{d+2}{2}} \int_{d\sigma^2 C}^{\infty} r'^{d+1} e^{-(r')^2(d+1)/(2d\sigma^2)} dr' \\
&\leq 4 \int_{\sqrt{dC}\sigma}^{\infty} r'^{d+1} e^{-(r')^2/(2\sigma^2)} e^{-(r')^2/(2d\sigma^2)} dr' \\
&\leq 4 e^{-C/2} \int_0^{\infty} r^{d+1} e^{-r^2/(2\sigma^2)} dr.
\end{aligned}
$$

Hence, we have that

$$
\mathbb{E}[(\tilde{y}_k)^T H \tilde{y}_k \mid \mathcal{E}_2^c] \leq 4 e^{-C/2} d\eta\sigma^2 \leq \delta d\eta\sigma^2 / 2.
$$

Putting together, we have that,

$$
\left| \mathbb{E}[h(\tilde{y}_k)] - \eta\sigma^2 d/2 \right| \leq \eta^2 \sigma^2 \mathrm{Tr}(H) + \frac{\Delta\rho C}{\gamma^2} d\eta\sigma^2 + \rho \left( \frac{d\eta\sigma^2 C}{\gamma} \right)^{3/2} + 2\delta M + \delta d\eta\sigma^2/2.
$$

The proof is then complete. $\qquad\square$

We will now state the complete version of Theorem 3.4.

**Assumption 7** (Sufficient Small Learning Rate). *We will assume the following for constant $\delta \in (0,1]$ and learning rate $\eta$:*

1. $\eta < 1/(2\gamma_{\max})$.
2. $\eta \leq \gamma^3/(1600\rho^2\sigma^2 d \log(8\gamma T/\delta))$.
3. $10\frac{\sqrt{\eta}\sigma}{\sqrt{\gamma}}\sqrt{d \log(8\gamma T/\delta)} + 400\eta\rho\sigma^2 d \log(8\gamma T/\delta)/\gamma^2 \leq r$.

**Theorem C.32** (Complete version of Theorem 3.4). *If a loss $L$ is a river valley (Definition 3.1) and satisfies Assumptions 3 and 5, for any constants $\delta \in (0,1)$ and $T > 1/\gamma$, for sufficiently small learning rate $\eta$ satisfying Assumption 7, the iterate defined in Equation (4) with $\eta_k = \eta$, satisfies that for any integer $t \in [1/\eta\gamma, T/\eta]$, there exists a $\tilde{T}$ satisfying that,*

$$
\mathbb{E}[L(\tilde{w}(t))] - L(x(\tilde{T})) = (d-1)\eta\sigma^2/2 + \epsilon_L
$$

*where $\epsilon_t = 4\eta\gamma_{\mathrm{flat}}$ and $|\epsilon_L| \leq \tau\eta^2\sigma^2 + \rho(Cd\eta\sigma^2/\gamma)^{3/2} + C\kappa' d\eta\sigma^2 + \delta(2M+\eta\sigma^2 d) \ll (d-1)\eta\sigma^2$ with $C = 200 \log(64\gamma T/\delta)$.*

*Proof.* By Lemma C.28, we can write

$$
L(w) = h(w - \Phi(w)) + L(\Phi(w)).
$$

Hence we can separate the dynamics of Equation (4) into two parts, namely $w = \Phi(w) + (w - \Phi(w))$. It is easy to check that when constrained on range of $P_S$, $h(y)$ satisfies Assumption 6. Hence, we can use Lemma C.31 to control $h(w_t - \Phi(w_t))$. For $\Phi(w_t)$, the iterates is running a gradient descent with learning rate $\eta$ on $\mathcal{M}$ and we can use proof analogous to the proof of Theorem 3.3 to show that if $\Phi(w_t) = x(\tilde{T}(t,\eta))$, then there exists $T_0$, such that

$$
\tilde{T}(t,\eta) \in [T_0 + (1 - 4\eta\gamma_{\mathrm{flat}})\eta t, T_0 + (1 + 4\eta\gamma_{\mathrm{flat}})\eta t].
$$

This completes the proof. $\qquad\square$

## C.8 PROOF OF THEOREM 3.5

We will first state the complete version of Theorem 3.5.

**Theorem C.33.** *Under the setting of Theorem C.32, the SGD iterates (defined in Equation (4)) with the decaying learning rate schedule satisfies that for any integer $t \in [t_s, 1.1t_s]$, there exists a $\tilde{T} \in [(1 - \epsilon_t)T(t), (1 + \epsilon_t)T(t)]$ satisfying that,*

$$\mathbb{E}[L(\tilde{w}(t))] - L(x(\tilde{T})) \leq (d-1)\eta_k\sigma^2/2 + \epsilon_L$$

*with $T(t) = T + \sum_{k=t_s}^{t} \eta_k$.*

*Proof.* The proof is analogous to Theorem C.32 and Lemma C.31. We will omit the detail derivation and only focus on deriving the variance of corresponding $\tilde{y}_k$.

$$\tilde{y}(k+1) = \tilde{y}_k - \eta_k H\tilde{y}_k - \eta_k g_k, \, w(0) = 0, \mathbf{g_k} \sim \mathcal{N}(0, \sigma^2\mathcal{I}), \tilde{y}(0) = 0.$$

Here the covariance of $\tilde{y}_k$, denoted as $\Sigma_k$ satisfies that

$$\Sigma_{k+1} = (\mathcal{I} - \eta_k H)^2\Sigma_k + \sigma^2\eta_k^2\mathcal{I}.$$

If we consider $i$-th eigenvector of $H$ as $v_i$, and denote $\sigma_{k,i} = v_i^\top \Sigma v_i$.

Analogous to the proof of Theorem C.32, $\left|\sigma_{k_s,i} - \frac{\eta\sigma^2}{\gamma_i}\right| \leq \frac{4\eta^2\sigma^2}{\gamma_i}$.

We further have that

$$\sigma_{k_s+r+1,i} = (1 - \frac{\eta}{2 + r\eta\gamma}\gamma_i)^2\sigma_{k_s+r,i} + \sigma^2\frac{\eta^2}{(2 + r\eta\gamma)^2}.$$

Then by induction, we can prove that for $r \geq 0$

$$\sigma_{k_s+r+1,i} \leq \frac{\sigma^2}{\gamma_i}\frac{\eta}{2 + r\eta\gamma} + \frac{4\eta^2\sigma^2}{\gamma_i} = \frac{\sigma^2}{\gamma_i}\eta_{t_s+r+1} + \frac{4\eta^2\sigma^2}{\gamma_i}.$$

The rest follows the proof of Theorem C.32. $\qquad\square$

## C.9 PROOF OF LEMMA 4.1 AND THEOREM C.35

In this section, we will denote $\frac{\exp(\Theta_{i,j})}{\sum_{j=1}^{m} \exp(\Theta_{i,j})}$ as $\mathcal{Q}_{i,j}$

We will study this loss

$$L(\Theta) = \frac{1}{n}\sum_{i=1}^{n} \ell_i(\Theta_{i,:}), \quad \ell_i(\Theta_{i,:}) = -\sum_{j=1}^{m} \mathcal{P}_{i,j}\log\frac{\exp(\Theta_{i,j})}{\sum_{k=1}^{m} \exp(\Theta_{i,k})}. \tag{26}$$

**Lemma C.34.** *The loss defined $L$ in Equation (26) satisfies that*

$$(\nabla L(\Theta))_{(i,j)} = \mathcal{P}_{i,j} - \mathcal{Q}_{i,j}.$$
$$(\nabla^2 L(\Theta))_{(i,j),(i',j')} = \mathbf{1}(i = i')(\mathcal{Q}_{i,j}\mathbf{1}(j = j') - \mathcal{Q}_{i,j}\mathcal{Q}_{i,j'}).$$

*Proof.* The loss satisfies that

$$L(\Theta) = \sum_{i=1}^{n}(\sum_{j=1}^{m} \mathcal{P}_{i,j}\Theta_{i,j}) - \log(\sum_{j'=1}^{m} \mathcal{P}_{i,j'}).$$

Hence,

$$(\nabla L(\Theta))_{(i,j)} = \mathcal{P}_{i,j} - \mathcal{Q}_{i,j}.$$

Taking differentiation for another time yields the desired result. $\qquad\square$

*Proof of Lemma 4.1.* This can be done by directly summing diagonal entries in Lemma C.34. $\quad\square$

**Assumption 8.** *We will assume there exists constant $\gamma$ and positive integer $n' < n$ such that $\mathcal{P}$ satisfies the following assumption,*

1. *For any $i \leq n'$, $\forall j, \mathcal{P}_{i,j} > 8\gamma$.*
2. *For any $i > n'$, there exists $j_i$, $\mathcal{P}_{i,j'} > 1 - \gamma$.*

**Assumption 9.** *We assume the existence of a "generalized river", which is a p-dimensional manifold $\mathcal{M}$ such that any point $w \in \mathcal{M}$ has a gradient $\nabla L(w)$ lies in the eigenspace spanned by the last $k$ eigenvectors' direction of the Hessian, $\{v_i \left(\nabla^2 L(w)\right) \mid i \in [d - p + 1, d]\}$.*

**Theorem C.35.** *Under Assumption 8, a generalized river with dimension $n'm + (n - n')$ exists in the loss landscape defined by $L$ in Equation* (26).

*Proof.* According to Lemma C.34, the Hessian for $L$ is block-diagonal. Now fixing a city $i$, we will analyze the eigenvalue distribution in this block. Let $q = [\mathcal{Q}_{i,j'}]_{j' \in [m]})$, then this block is $\mathrm{diag}(q) - qq^T$.

For all non-zero eigenvalue $\lambda$ for this block, there exists $v$ such that

$$\mathrm{diag}(q)v - q^T v q = \lambda v.$$

Hence, we have that

$$v_j = \frac{q_j q^T v}{q_j - \lambda}$$

This implies $\sum_{j=1}^{m} \frac{q_j^2}{q_j - c} = 1$. We then have $\lambda \geq 0$ and there exists only one eigenvector corresponding to $\lambda = 0$. For the rest nonzero eigenvalue, we have that $\lambda > \min q_i$.

Now if we consider the manifold $\mathcal{M}$ defined as

$$\mathcal{M} = \{\Theta \mid \forall i \leq n', \mathcal{Q}_{i,j} = \mathcal{P}_{i,j}; \forall i \geq n', \mathcal{Q}_{i,j_i} > 1 - \gamma\}.$$

Then for all $\Theta \in \mathcal{M}$, we have that the gradient is zero for all dimensions $(i, j)$ with $i \leq n'$. Further, we know all the nonzero eigenvalues for these dimensions are at least $8\gamma$ by Assumption 8. For the rest of dimensions $(i, j)$ with $i > n'$, by Lemma 4.1, the largest eigenvalue is bounded by $1 - (1 - \gamma)^2 < 2\gamma$. This shows that the gradient falls in the eigenspace spanned by the last $n'm + (n - n')$ eigenvectors, which concludes the proof. $\quad\square$

## C.10 Technical Lemma

**Lemma C.36.** *If a function $F(t)$ satisfies that*

$$\frac{dF(t)}{dt} \leq -AF(t),$$

*then $F(t) \leq e^{-At} F(0)$.*

*Proof of Lemma C.36.* Consider $G(t) = F(t)e^{At}$, then

$$dG(t) = e^{At} dF(t) + A e^{At} F(t) \leq 0.$$

Hence $G(t) \leq G(0)$. $\quad\square$

**Lemma C.37.** *If a random vector $g \sim \mathcal{N}(0, \Sigma)$ and $\delta \in (0, 1)$, then it holds that*

$$\mathbb{P}(\|g\|_2 \geq 2\sqrt{\mathrm{Tr}(\Sigma)}\sqrt{\log(2/\delta)}) \leq \delta$$

*Proof of Lemma C.37.* Assume $\Sigma = Q\Lambda^2 Q^T$ with $Q$ being an orthonormal matrix and $\Lambda$ being diagonal with diagonal $\lambda_i$ for $i \in [d]$, further let $g'$ being a standard gaussian random vector, then $g$ follows the same distribution as that of $\Lambda Q^T g'$, which is further identical to $\Lambda g'$.

$$\mathbb{P}(\|g\|_2 \geq C) = \mathbb{P}(\|\Lambda g'\|_2 \geq C) = \mathbb{P}(\sum_{i=1}^d \Lambda_i^2 (g_i')^2 \geq C^2) \leq \mathbb{E}[\exp(t \sum_{i=1}^d \Lambda_i^2 (g_i')^2 - tC^2)].$$

It is well known that the moment-generating function of $(g_i')^2$ is

$$E[\exp(t\Lambda_i^2 (g_i')^2)] = \frac{1}{\sqrt{1 - 2t\Lambda_i^2}}.$$

Hence $\mathbb{P}(\|g\|_2 \geq C) \leq e^{-tC^2} \prod_{i=1}^d \frac{1}{\sqrt{1-2t\Lambda_i^2}} \leq \frac{e^{-tC^2}}{\sqrt{1-2t\mathrm{Tr}(\Sigma)}}$.

With $t = \frac{1}{4\mathrm{Tr}(\Sigma)}$, it holds that $\mathbb{P}(\|g\|_2 \geq C) \leq 2e^{-\frac{C^2}{4\mathrm{Tr}(\Sigma)}}$. This concludes the proof. $\square$

**Lemma C.38** (Doob's Inequality). *Let $X_1, \ldots, X_n$ as a positive submartingale adapted to filtration $\mathcal{F}_1, \ldots, \mathcal{F}_n$, which means $X_i \leq \mathbb{E}[X_{i+1} \mid \mathcal{F}_i]$, then*

$$\mathbb{P}(\sup_{i \leq n} X_i > C) \leq \frac{\mathbb{E}[X_n]}{C}.$$

# D OMITTED EXPERIMENTS DETAILS

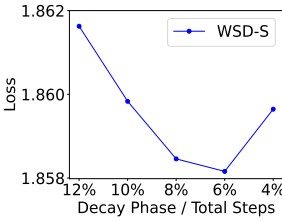

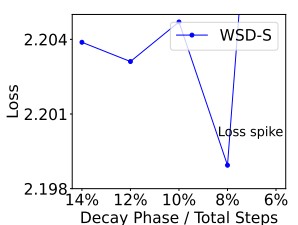

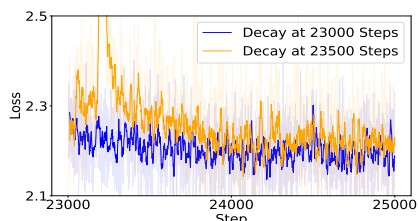

(a) 1.2B Models on 200B tokens

(b) 0.1B Models on 100B tokens

(c) 0.1B Models on 100B Tokens. Decay Near a Loss Spike (6%)

Figure 15: **Ablation Study on the Sensitivity of Fraction of Time Decaying**. This study examines two settings: a smaller scale with 0.1B parameters trained on 100B tokens (middle figure) and a larger scale with 0.6B parameters trained on 200B tokens (left figure). The results indicate that the final performance is similar when the decay phase is 8%-12% of the total training steps. However, the right figure demonstrates a significant performance loss when decaying near a loss spike. It compares two training loss curves with decay phases of 8% and 6% of the total compute on the 0.1B models, where the latter starts immediately after a loss spike, leading to a validation loss increase of 2e-2.

We train LLaMA models with 4 parameter sizes using the Levanter framework for our study on *WSD-S*. For our theoretical study, we pretrain a 124M GPT-2 using the nanoGPT framework with a learning rate 6e-4 and train it with a batch size of 0.5M for 100k steps with warmup steps of 2k.

We hereby provide all the hyperparameters we used for the LLaMA and GPT-2 models training.

| Model | Hidden Dim | Intermediate Dim | Num Layers | Num Heads | Peak LR |
|---|---|---|---|---|---|
| 0.1B LLaMa | 768 | 3072 | 12 | 12 | 6e-4 |
| 0.3B LLaMa | 1024 | 2048 | 24 | 16 | 6e-4 |
| 0.6B LLaMa | 1536 | 6144 | 24 | 32 | 4e-4 |
| 1.2B LLaMa | 2048 | 8096 | 16 | 32 | 4e-4 |
| 0.1B GPT-2 | 768 | 3072 | 12 | 12 | 6e-4 |

Table 2: Specifications for Different Sizes of LLaMa Models

We decay the model for the last $10\%$ of the training runs with one exception for 0.3B model using *WSD* method near 25k steps to avoid loss spikes. We outline the decaying and resuming point (the unit is 1k steps) we choose here:

| Model | 1st Decay Starts/Resume | 2nd Decay Starts/Resume | 3rd Decay Starts |
|-------|-------------------------|-------------------------|------------------|
| 0.1B LLaMa | 11.25 / 12.5 | 22.5 / 25 | 48.75/ 53.75 |
| 0.3B LLaMa | 11.25 / 12.5 | 22.5 / 25 | 48.75/ 53.75 |
| 0.6B LLaMa | 11.25 / 12.5 | 22.5 / 25 | 48.75/ 53.75 |
| 1.2B LLaMa | 11.25 /12.5 | 22.5 / 25 | 48.75/ 53.75 |

Table 3: Specifications for Decaying Steps for *WSD-S* Method

| Model | 1st Decay Starts/Ends | 2nd Decay Starts/Ends | 3rd Decay Starts | Total Steps |
|-------|-----------------------|-----------------------|------------------|-------------|
| 0.1B LLaMa | 11.25 / 12.5 | 22.5 / 25 | 45/ 50 | 53.75 |
| 0.3B LLaMa | 11.25 / 12.5 | 22 / 25 | 45/ 50 | 54 |
| 0.6B LLaMa | 11.25 / 12.5 | 22.5 / 25 | 45/ 50 | 53.75 |
| 1.2B LLaMa | 11.25 /12.5 | 22.5 / 25 | 45/ 50 | 53.75 |

Table 4: Specifications for Decaying Steps for *WSD* Method

