# OpenReview forum: "Understanding Warmup-Stable-Decay Learning Rates: A River Valley Loss Landscape View"
_ICLR.cc/2025/Conference — ICLR 2025 Poster_

### Official Review · Reviewer_bscX · 2024-10-17

**Soundness:** 2
**Presentation:** 2
**Contribution:** 2
**Rating:** 5
**Confidence:** 4

**Summary:**

This paper is composed of three main sections.  In the first section, they derive some analytical results for a specific type of loss landscape, which they call the “river valley landscape.”  The river valley landscape corresponds to a kind of loss function where the loss is reduced by moving in the direction of lowest curvature (the river direction), and there are steep hills on both sides of the river.  In such a situation, with a number of assumptions, it can be shown for gradient flow (continuous time), gradient descent (full batch gradients) and stochastic gradient descent (with approximate gradients), that the parameters will follow the river within a given bound.  Moreover, in the stochastic case, the loss is also bound by a term related to the variance of the gradients multiplied by the learning rate (which they associate with movement of the parameters up the hills).  A Warmup-Stable-Decay (WSD)-style schedule that has a decay at the end can, in theory, move along the river during the constant phases, and subsequently move down the hill directions during LR annealing.  The next section describes a bigram model where different initial tokens have different amounts of uncertainty in their following token.  The degree of uncertainty is shown (at an optimum) to correspond to the degree of curvature in the loss, suggesting that the “river” direction corresponds to learning more deterministic relationships, while the hill direction corresponds to more uncertain predictions.  Decayed LRs are shown to do relatively better on the uncertain bigrams in a small experiment.  Finally, the paper considers the case where multiple checkpoints are desired at intervals during training at a given model scale (e.g., to make a data-size-based scaling law).  If using the WSD schedule, the decaying happens in separate branches off the main sequence of checkpoints.  The paper makes the observation that even during these decaying phase, progress is made along the river direction.  This motivates the WSD-Simplified schedule (WSD-S), where rather than moving back to the pre-decay checkpoint in the main branch, we continue training directly after the decay phase.  This amounts to running cycles of WSD-style training.  Some experimental results dig into whether WSD-S is effective compared to WSD and to traditional schedules like Cosine decay or cyclic Cosine decay.

**Strengths:**

The WSD learning rate schedule has recently drawn a lot of attention in LLM training.  WSD allows us to obtain high-quality intermediate models by decaying in “branches” from checkpoints in the main training process.  Perhaps the key insight of the paper is that if using WSD, we must acknowledge that the decay portions are “wasting” a significant amount of compute in that the compute spent during these decay portions does not contribute toward the training of future checkpoints.  Prior work has recommended up to 20% of total steps in decaying, so decaying 5 times (e.g., to obtain a scaling law) could double the total compute spent training a model.  We can think of the proposed WSD-S approach as a philosophy: if you do decay, you should resume from *after* the decay, rather than from before it.  In this way, we seem to have the same amount of flexibility as WSD (with the exception that with WSD-S, we can’t run the decay in parallel with continuing training in the main branch, as you can with WSD).  Testing this philosophy is well-motivated, and of solid interest to the community.

The idea of specifying a loss landscape of a certain type (river-valley), and deriving different bounds on optimization given that landscape, and then determining whether different real-world training scenarios fit such landscapes, also seems well-motivated.
In terms of clarity, I found the figures fairly helpful in most cases.  For work like this, pictures of the LR schedules can tell the whole story and so Figure 2 was good, and the little arrows in Figure 2(b) (and even littler ones in Figure 10) were very helpful.  Figure 3 was helpful as well.  I also realized that after reading the paper, I wasn’t really confused about anything, so the work must have been explained somewhat clearly.

I thought the material on decay after loss spikes was interesting and fairly important.

**Weaknesses:**

The paper in its current form seems incomplete in many ways.  For one, the paper has not undergone a proper proofreading/editing/review process.  This makes it much harder for the reviewers to assess the quality of the work – I felt like I was being asked to help prepare an early draft of a paper written by my co-author, rather than reviewing a paper that is already ready for submission.  Some typos are probably just honest mistakes, e.g., repeatedly spelling “decaying” as “decayping” – but if you pasted the paper text into ChatGPT, or ran a simple spell checker, it would find many of these typos!  It’s unsettling to see typos in the abstract (!) “one can branch out from the main branch at a proper at any time”, in main theorems “with the learning decaying learning rate schedule”, in Figure captions ‘Figure 7: textbfReproducing [sic] the Nontraditional Loss Curve.”  There are missing LaTeX references (I assume) “We will first motivate the decaying function we choose ?? [sic] using a simple example”.  Figure 7 is blocking part of the caption of Figure 6.  I feel like all of these would have been picked up if a co-author had simply reviewed the paper prior to submission (and writing could have been improved in general with such review).

Moreover, the paper has a very non-traditional style: the main body of the paper is missing Related Work, Discussion, and Conclusion sections (although there is Related Work in the appendix, which reviewers are not required to read).  This makes it difficult to contextualize these results compared to prior work (missing the Related Work), and understand the key insights that we draw from the experiments and theoretical sections (missing Discussion/Conclusion).  For example, does this paper present a negative result?  I.e., is WSD just as good as WSD-S, even if we count the decay portions as part of the overall compute?  This seems to be what Figure 10 is showing.  It would be interesting to learn (from a conclusion) whether the authors feel the same way.

There is also some missing discussion of related work.  E.g., consider the following passage: “We find that the loss interpolation between parameters before and after each training iteration’s update is roughly convex with a minimum (valley floor) in between for most of the training. Based on this and other metrics, we deduce that for most of the training update steps, SGD moves in valley like regions of the loss surface by jumping from one valley wall to another at a height above the valley floor. This ’bouncing between walls at a height’ mechanism helps SGD traverse larger distance … this exploration above the valley floor allows SGD to quickly travel far away from the initialization point”.  Actually, this passage is not from the current paper, but from Xing et al., “A walk with SGD” (2018) (arXiv: 1802.08770v4), which is not cited here.  Contextualizing the findings of Xing et al (and other works) within the main body of the paper is essential.

I think another weakness is that, of the three main sections of the paper, none of them really went into enough depth to convince me of the main arguments.

For the theoretical section, it’s not clear, of all the many assumptions, which apply to modern LLM training.  Like, is the 4x “eigengap” commonly found?  How important is it?  Also, what is the point of departure here from other theoretical work in SGD, for example from Bottou et al, “Optimization methods for large-scale machine learning”. SIAM review, 60(2):223–311, 2018?  The benefits of decaying LRs has a long history in optimization, and decay has previosly been motivated by similar considerations to those used here (moving from initial conditions early, reducing stochastic noise later on).  If you frame your findings in terms of this earlier work, the contrast can help us understand the benefits of your analysis.  Like, maybe the stochastic noise plays a greater role if the loss landscape has a river-valley form (motivating greater decay than we would use otherwise), and we can see this in contrast to what we’d see without all the river valley assumptions.  These other works often derive a bound from the global optimum, but here we seem to be bounded from a point x “further down the river” – what is the significance of this?  This could indeed be a paper all in itself.

Moreover, I am not convinced by the probes in Figure 5, given that the two points are sampled 5 *billion* tokens apart.  Does this mean that the river has no bends?  Like, if you interpolated between two points in Figure 3, you can clearly see that you would typically travel up the hill and back down the other side.  Does this invalidate the river analogy?  If not, how does Figure 5 “validate” it?  Figure 2(a) says the iterate will “oscillate” between the hillsides – why not check this over fewer than 5 billion tokens?  Plotting Figure 5(a) at a single pair of checkpoints is not sufficiently rigorous, from my perspective.

The empirical verification in Section 3 lacks a lot of details – “we first train a toy model” – but what is this model?  A neural network?  Next we fine-tune a pre-trained GPT2 model, but of what size?  Why is it pre-trained?  What are the objectives of doing this?  There are no ablations here, nothing is varied and repeated, there are no error bars.  Basically, it’s not clear why the design choices were made for these experiments, and what the objectives were, and what the implications of these results really are.

For Section 4, my main objection is that I suspect the average learning rate to be a confounder.  Think of your Theorem 5: progress throughout training is partly controlled by the sum of the step sizes (this theorem is for decay, but it’s true during the stable phase as well).  Then look at Figure 9: WSD-S has a higher average LR than Cyclic-Cosine.  Would cyclic cosine work better than WSD-S if we simply increased the LR slightly?  The paper says, “We hypothesize that a model trained with a small learning rate for too long, as with Cosine, is implicitly hurt compared to a model trained with a large learning rate for the majority of the run, as with WSD or WSD-S.”  Exactly!  So why not train Cosine with a higher LR?

**Questions:**

Why doesn’t the blue line go as far as the red line in the “0.3B Parameters” plot in Figure 10?  The red line seems to stop short of 50,000.

The introduction says, “WSD-S leads to a better validation loss than WSD under the same compute budgets due to the re-use of the decay period”.  Is Figure 10 validation loss? (it doesn’t say).  For the 0.6B parameter model trained to 50,000 steps, don’t they obtain basically the same loss at the end?  And by definition, they always have the same loss on the first checkpoint, right?  I mean, with the loss spikes seeming to affect the intermediate results, can we really conclude that WSD-S gets better validation loss under the same compute budgets?  Do we need to soften this statement in the introduction?

Do you think WSD-S, since it has a slightly lower summed LR than WSD (lower total LR “area”) might actually do better than WSD if we increased the peak LR slightly?  Do you think Cyclic-Cosine could also improve if we increased the peak LR?

---

> ### Author Response · Authors · 2024-11-27
>
> We thank the reviewer for the helpful suggestions for us to improve the paper. We have updated our paper following the advice and would like to address the concerns here.
>
> **W1:** The paper contains typos and lacks proper proofreading.
>
> We apologize for the mistakes and have fixed the typos in the updated draft.
>
> **W2:** The non-traditional structure omits essential sections like Related Work or Contribution.
>
> We have added a shortened version of related work in the main text. We have outlined the contribution in the introduction and hence omitted the conclusion section.
>
> **W3:** Is WSD-S better or equivalent to WSD?
>
> While WSD-S only slightly outperforms WSD, our experiments show that this improvement is very robust as we demonstrated it over different scales of parameters and data. We have also added another set of experiments on a different dataset (DCLM) and the improvement continues to hold.
>
> **W4:** We didn’t cite and discuss relevant related work, such as Xing et al. (2018).
>
> We thank the reviewer for pointing us to this related work. We have added it to discuss it in our paper. They presented a similar conceptual picture with us, arguing that SGD locally bounces around the valley on top of \\emph{valley floor}. In the conceptual model they propose, the valley floor is uneven, and the iterates explore the valley floor to find a more generalizable solution. In contrast, we focus on the optimization perspective and assume the existence of the \\emph{river} at the bottom of the hillsides, where the loss monotonously decreases. We build a formal theoretical framework on top of this picture, leading to multiple quantifiable theoretical predictions. For example, Theorem 2.5 predicts that the loss drop will be linear concerning the learning rate decay in the decay phase.
>
> **W5:** In the theoretical section, it is unclear which assumptions apply to modern LLM training. For example, the eigengap may not exist
>
> As argued in our introduction, our work provides a *conceptual picture.* The goal of the river valley landscape is not to claim that the pretraining loss **is** a river valley. Instead, we argue that river valley is a useful abstraction in which WSD can obtain near-optimal performance and we can provide useful predictions.
>
> Therefore we believe that the criterion for our assumptions should not be whether they hold **exactly** in LLM training. Instead, the criterion should be whether the assumption captures properties in the LM pretraining loss. For example, the eigengap assumption \[1\] is an abstraction for the skewed hessian that typically appears in optimization. While we admit the constant 4 and even the eigengap assumption itself may be weakened, we disagree that this impacts the major contribution of our theory.
>
> \[1\] An Investigation into Neural Net Optimization via Hessian Eigenvalue Density
>
> **W6: It is not clear how the work departs from existing theoretical studies on SGD.**
>
> 1. In our paper (Figure 4), we introduce a simple quadratic function to illustrate the high-level concept of the river-valley loss landscape and explain why the WSD learning rate schedule can achieve strong performance. Despite the simplicity of the example, we are not aware of any exact reference for this. We would appreciate it if the reviewer knew such a reference and could point it to us.
>
> 2. More importantly, deep learning loss functions are inherently non-convex, and it is not immediately evident whether such landscapes and mechanisms extend to non-convex settings. Standard analyses of non-convex stochastic optimization typically aim to find approximate first- or second-order stationary points, often prescribing a  $1\\sqrt{t}$  decaying learning rate schedule. These analyses, however, do not account for or justify the effectiveness of the WSD schedule.
>
> Our main contribution is a clear and elegant characterization of the “river” structure hidden within general non-convex losses, along with a set of sufficient conditions—such as the slow spinning of the bottom eigenspace and the presence of an eigengap—that explains why the WSD schedule works effectively. These mechanisms are conceptually similar to those observed in the quadratic case. Additionally, we present a simple non-convex 2D function (Figure 3\) as a concrete example, demonstrating that our sufficient conditions are meaningful and non-trivial. To the best of our knowledge, our theoretical framework is the first to justify the effectiveness of WSD schedules in the non-convex setting.

---

> > ### Author Response · Authors · 2024-11-27
> >
> > **W7:** The probes in Figure 5 are not convincing and may undermine the river valley analogy due to the large sampling distance between points. Does this mean that the river has no bends?
> >
> > 1. We believe it is a strength rather than a weakness to have a large sampling distance. One of our theoretical assumptions is that the river bends a little throughout training so we choose two iterates that are 5B apart to stress test this theory. The results suggest that even for training durations over 5B tokens, the iterates are likely to be in the same part of the valley.
> > 2. We have also included experiments with smaller training durations in our updated paper in Appendix B and we show that the results still hold.
> >
> > **W8:** The empirical verification in Section 3 lacks detailed descriptions of the models and experimental design, making the results unclear and the design choices unmotivated.
> >
> > We have updated Section 3 to better state the details and the implications of the experiments. We perform the GPT-2 experiments to show that the difference in the uncertainty of the next token will shape the loss landscape of a (pretrained) Transformer in a similar way that it shapes the loss landscape for the toy model defined in Section 3\.
> >
> > **W8 (\&Q3):** Section 4 may have confounding factors such as average learning rate, questioning the fairness of comparisons between WSD-S and Cyclic-Cosine schedules.
> >
> > To remove this potential confounding, we perform a new set of experiments sweeping the learning rate of 600M models over training durations of 50B tokens. We show that after sweeping the learning rate, WSD-S still outperforms Cyclic-cosine. The results are presented in Appendix B.
> >
> > ### **Questions**
> >
> > **Q1:** Why doesn’t the blue line go as far as the red line in the “0.3B Parameters” plot in Figure 10?
> >
> > As mentioned on line 497, we decay the WSD learning rate for the 0.3B models at 22B tokens instead of 22.5B tokens due to encountering a loss spike. Therefore, we trained the 0.3B parameters model with WSD-S for a slightly shorter training length. However, this is to the advantage of the baseline method WSD and we found out that WSD-S outperforms WSD already.
> >
> > **Q2:** Is Figure 10 showing validation loss, and can the introduction's claim that WSD-S achieves better validation loss under the same compute budgets be supported?
> >
> > Yes, Figure 10 shows the validation set. While the absolute improvement’s magnitude is not large, this order of improvement is typical when comparing pretraining loss. As argued above in response to **W3,** the improvement is consistent considering the choices of the dataset and different scales of models.

---

> > > ### Comment · Reviewer_bscX · 2024-12-02
> > >
> > > I thank the authors for addressing the points raised in my review.  I believe these changes and additions improve the paper significantly and will adjust my rating accordingly.

---

### Official Review · Reviewer_H6GK · 2024-10-31

**Soundness:** 2
**Presentation:** 3
**Contribution:** 2
**Rating:** 3
**Confidence:** 3

**Summary:**

The purpose of the paper is two-fold, 1) understand the success of WSD by postulating that the loss plane is analogous to a river-valley. 2) They attempt to provide more theoretical justification and empirical support to this hypothesis, resulting in optimizing WSD to WSD-S under their assumptions.

**Strengths:**

They clearly articulate their idea of a river loss landscape and provide some theoretical guidance to establish it. They provide some benchmarks with various smaller models and evaluate the loss trajectories. They also provide an attempt to try and leverage their knowledge to propose WSD-S.

**Weaknesses:**

The main weaknesses observed are the following:
1. The assumption of a river valley isn't guaranteed and is not strongly supported. Additional theoretical or experimental results supporting this conclusion are necessary.
2. Results are demonstrated on very small models, with identical architectures, identical batch-sizes, and relatively small corresponding datasets. All of which could influence the loss region.

**Questions:**

Have the authors examined different architectures, datasets, learning rates, or attempted to examine other possible explanations for the loss curve region?

Did you evaluate the models on benchmarks post-training?

Provided the hypothesis, why do we have divergence randomly during training of large models on stable plains?

---

> ### Author Response · Authors · 2024-11-27
>
> We thank the reviewer for the advice to improve our paper. We would like to address the concerns here.
>
> **W1: More results supporting the existence of river valleys are necessary.**
>
> Thank the reviewer for the suggestions. We have added the following evidence to our paper.
>
> 1. **Theoretical justification**. We prove that the river must exist when (1) the Hessian has an eigengap, and (2) the flattest direction spins slowly. The new result is listed in Appendix C.
> 2. **Empirical evidence.** Beyond the original mode connectivity result, we present a 2-dimensional loss landscape visualization, showing that on the line segment connecting decayed checkpoints, the losses are consistently low and decrease slowly. In the direction connecting checkpoints in stable and decay checkpoints, the losses change sharply. This is consistent with our definition of flat rivers and sharp hillsides.
>
> **W2 & Q1.1: Results are demonstrated on small models and small datasets with identical hyperparameters**
>
> 1. **Model sizes.** We have scaled our models from 0.1B to 1.2B, which is a standard set of model sizes in studies on language modeling \[1,2\].
> 2. **Architecture choices.** The main body of our experiments studied the LLaMA architecture, which is a standard choice in current language modeling. We have also trained a GPT-2 architecture model in Section 3 and we observed a similar phenomenon.
> 3. **Dataset sizes.** We disagree that we are training on a small data size. We have used the Pile dataset in our experiments and trained for 200B tokens for each model, which is 10 times larger than the Chinchilla prediction \[3\] for models with a size of 1.2B.
> 4. **Dataset choices.** We have added experiments to train our models on a different pretraining dataset DCLM and our results continue to hold. The results are shown in Appendix B.
> 5. **Other hyperparameter choices.** For each model with the same scale, we use a fixed peak learning rate and keep the other hyperparameters constant in all the experiments. This is also consistent with previous studies \[1,2\]. While it is interesting and important to understand the interplay of hyperparameters in the experiments, it is beyond the scope of our paper and computationally infeasible for us to run all the experiments.
>
> \[1\] xLSTM: Extended Long Short-Term Memory
> \[2\] Mamba: Linear-Time Sequence Modeling with Selective State Spaces
> \[3\] Training Compute-Optimal Large Language Models
>
> **Q1.2: Have we attempted other explanations?**
> Yes, in our paper (Figure 4), we present a simple quadratic function illustrating the high-level idea of river-valley loss landscape and why the WSD learning rate schedule can lead to good performance. It is hard to determine how far we can extrapolate the implication of this toy function apriori. If we consider stochastic optimization over general convex smooth loss, it is known that the minimax optimal schedule requires decaying like 1/T so WSD will not be an optimal learning rate \[1\].
>
> Therefore,  our main contribution is a clean generalization of this toy function to a family of potentially non-convex loss with a novel and clean definition of the river and a sufficient set of conditions to show that WSD will track the river and perform well.
>
> \[1\] First-order and Stochastic Optimization Methods for Machine Learning, Theorem 4.2
>
> **Q2: Did we evaluate models on downstream tasks?**
>
> As the main goal of this paper is to speed up the optimization of pretraining, we didn’t include benchmarks in the previous versions. We have added an experiment to evaluate models trained with different learning rate schedules on heldout corpus including Penn Treebank and RedPajama. WSD-S continues to outperform WSD.
>
> **Q3: How to explain the loss spikes during training?**
>
> Due to the stochasticity of the update, there is a small probability on each step that the models may escape the neighborhood of the river and the loss may become very large. The probability of escaping monotonously increases with the learning rate. Hence, when the learning rate is too large, this event may happen often during training. After the happening of the event, the loss typically decreases back to the original level after a few steps. This suggests that the iterate may return to the local regime quickly after loss spikes, and hence it does not invalidate our theory.

---

### Official Review · Reviewer_A9Vb · 2024-11-01

**Soundness:** 4
**Presentation:** 4
**Contribution:** 4
**Rating:** 10
**Confidence:** 3

**Summary:**

In this paper, the authors explore a learning rate schedule named WSD-S, an enhancement of the previously introduced WSD model, which consists of three phases: Warm-up, Stable, and Decay. They provide a robust theoretical and empirical analysis supporting the effectiveness of WSD and propose its simplification in the new WSD-S variant.

The theoretical analysis presented in the study sets the mathematical definition of what the authors refer to as the "river valley" loss landscape. They demonstrate (Theorems 2.2 and 2.3) that in such landscapes, a higher learning rate yields greater progress over the same number of gradient descent steps. For stochastic gradient descent scenarios, while progress along the 'river' remains constant, an additional 'hill' component of loss emerges. The rapidly decaying learning rate phase of WSD effectively addresses this hill component (Theorem 2.4), suggesting that WSD is particularly beneficial for river valley-type losses. The authors further argue that language modeling pre-training losses resemble these river valley losses, thereby explaining the previously observed advantages of WSD.

For continual training scenarios, the authors highlight the importance of the transition from stable to decay phases. WSD continues the the training from the pre-decay checkpoint but authors argue that the progress made during the rapid decay phase is crucial and needs to be carried over. With this hypothesis, they present a simplified WSD where the same checkpoint (instead of the pre-decay checkpoint) after the decay is continued to be trained. This approach is operationalized in the WSD-S schedule, which the empirical analysis suggests outperforms other schedules like cosine, cyclic-cosine, and WSD.

**Strengths:**

The paper is excellently written for the complex nature of the analysis presented in the study. The information is organized well and easy to follow.

The authors have provided solid theoretical and empirical analysis to explain the effectiveness of the WSD and how simplification further benefits the optimization process.

**Weaknesses:**

None

**Questions:**

Minor edits:
Line 290: not sure if the eta_t is red on purpose
Line 399: missed a \ for textbf
Line 491: Figure 10 is difficult to read


Questions:
Line 236: What does it mean by “starting point w lies on the course of the river”? Does it mean that the starting point is on the manifold M or in some neighborhood of M?

Figure 8: Right bottom figure: The decay period is extended as the training progresses. If yes, why? Were there any experiments done to observe the effect of minimum LR (after the decay)?

I wonder, if more decay phases can consistently keep the hill component smaller and ultimately make more progress?  Or is there any sweet spot of number decays that helps WSD-S to achieve the best results?

---

> ### Author Response · Authors · 2024-11-27
>
> We thank the reviewer for the strong support.  We would like to address the concerns here.
>
> **Q1.1: Red learning rate on line 290**
>
> We try to highlight the learning rate here to separate it from the previous Theorem.
>
> **Q1.2: Typoes and formatting issues**
>
> We thank the reviewer for pointing out these problems and have fixed them.
>
> **Q2: What does it mean by “starting point w lies on the course of the river”?**
>
> We mean the starting point is inside a neighbor of the manifold.
>
> **Q3: Why do we need an extended decay period for a longer training duration?**
>
> Thank you for the good question. This is indeed beyond the scope of our current theory. If the Hessian of the loss landscape stays constant, our theory predicts that the decay period should be kept constant. Therefore, we believe the extended decay period is connected with the change of Hessian during the course of training and would love to examine this further in future works.
>
> **Q4: Can more decaying periods keep the hill component smaller? Can we determine how often to decay using WSD-S?**
>
> Empirically we observe that the loss will return to the original level soon after the learning rate returns to the peaked learning rate. This may suggest that the decrease of the hill component is not sustainable after the learning rate is resumed. Therefore, it is beyond the scope of our current theory to determine a way to utilize the decaying phase on purpose to get better performance.

---

### Official Review · Reviewer_UHrP · 2024-11-04

**Soundness:** 2
**Presentation:** 3
**Contribution:** 2
**Rating:** 6
**Confidence:** 3

**Summary:**

This paper compares the optimization space of language models to a river, thereby demonstrating the effectiveness of the WSD method. Based on this method, a simplified version, WSD-S, is proposed, which reduces computational complexity by reusing checkpoint weights without compromising performance.

**Strengths:**

1. The optimization algorithms supported by theoretical foundations are a crucial research direction.
2. Despite involving complex theoretical proofs, the writing of the article remains clear and easy to understand.
3. The proposed WSD-S performs no worse than WSD.

**Weaknesses:**

1. The theoretical proof in the paper relies on many assumptions, and the validity of these assumptions in complex large models is difficult to ascertain. For example, the paper assumes that the loss function is analytic, which seems unreliable in models like GPT2 that are filled with GELU functions.
2. The paper assumes that the optimization space of the loss function is a river and uses a toy model from Allen-Zhu's paper to verify this. However, neither the model assumptions in Zhu's paper nor the toy model in this paper are reflective of natural language situations, making it somewhat far-fetched to use such models to verify the existence of a river.
3. Compared to WSD, WSD-S does not offer any substantial innovative improvements, neither in terms of computational complexity savings nor in the final performance.

Minor comments：

The excessive length of the paper results in cramped formatting and sections. Especially in Figure 7.

**Questions:**

1. Could the authors provide more evidence that the optimization space might be a river?
2. Regarding the valley optimization space, do the authors have any ideas for developing more optimizers? For instance, how can more steps be focused on progressing along the river rather than crossing the river? Does Adam offer better descent speed compared to SGD in such a river valley? Additionally, I did not find the optimizer used by the authors mentioned, which should be specified in section 4.1.

---

> ### Author Response · Authors · 2024-11-27
>
> We thank the reviewer for acknowledging the contribution of our paper. We would like to address the concerns below.
>
> **W1: The theoretical assumptions in the paper are hard to ascertain. For example, the loss may not be analytical.**
>
> As argued in our introduction, our work provides a *conceptual picture.* The goal of the river valley landscape is not to claim that the pretraining loss **is** a river valley. Instead, we argue that river valley is a useful abstraction in which WSD can obtain near-optimal performance and we can provide useful predictions.
>
> We believe that these assumptions, in this sense, are faithful abstractions of the original loss landscape. As an example, here we assume the loss is analytical in the sense that it is infinitely differentiable, and its Taylor series converges to the function. Hence, both GeLU functions and the Transformers model composed with analytical activations are analytical. Therefore, at least a large family of models is trained on a smooth landscape.
>
> **W2: We use a toy synthetic language to illustrate how a river can emerge in the loss landscape. The language may not be reflective of natural language property and thus the argument does not verify the existence of river in natural language.**
>
> 1. The synthetic language we studied captures one property of natural language: there are vastly different uncertainties for the next word with respect to different contexts.
>    1. As argued on line 332, some of the tokens in natural languages are highly deterministic given the context while some are naturally ambiguous.
>    2. Our synthetic language is designed to abstract this particular property.
> 2. The goal of Section 3 is not to directly verify the existence of rivers in natural language. Instead, we aim to show that a river can emerge when a model is trained with data with different uncertainty levels.
>
> **W3: WSD-S does not offer any substantial innovative improvements, neither in terms of computational complexity savings nor in the final performance.**
>
> The main focus of our paper is the river valley theory and WSD-S can be viewed as a product of the river valley landscape theory.
>
> While the performance improvement of WSD-S over WSD is not a large margin, we have shown that it is a consistent improvement over different models and data scales. To further validate the effectiveness of WSD-S, we have tried rerunning our experiments on a new dataset called DCLM \[1\] for models with sizes 100M and 600M. Our results show that WSD-S yields better validation loss compared with WSD. The results are presented in Appendix B of the updated draft.
>
> \[1\] DataComp-LM: In search of the next generation of training sets for language models
>
> **W4: Formatting issues concerning Figure 7**
>
> We apologize for the mistakes and have fixed the issue in the updated draft.
>
> **Q1: Will there be more proof that there exists a river in the loss landscape?**
> Thank the reviewer for the suggestions. We have added the following evidence to our paper. The new results are listed in Appendix B.
>
> 1. **Theoretical justification**. We prove that the river must exist locally when (1) the Hessian has eigengap, and (2) the flattest direction spins slowly.
> 2. **Empirical evidence.** Beyond the original mode connectivity result, we present a 2-dimensional loss landscape visualization, showing that on the line segment connecting decayed checkpoints, the losses are consistently low and decrease slowly. On the direction connecting checkpoints in stable and decay checkpoints, the losses change sharply. This is consistent with our definition of flat rivers and sharp hillsides.
>
> **Q2.1: Do we have any ideas for new optimizers?**
> Given the river valley analogy, we think separating the directions of the river and hillside and using different adaptive learning rates will be a promising idea. We are planning to explore this further in future works.
>
> **Q2.2: Will Adam progress faster than SGD in the river valley landscape?**
> While we didn’t study Adam in this paper, we conjecture that the river valley landscape is connected with the effectiveness of Adam because Adam’s preconditioning may allow the usage of a larger learning rate while stabilizing the iterates between the sharp hillsides. This larger learning rate can then lead to faster progress along the river. We plan to explore this further in future works.
>
> **Q2.3: What is the optimizer we used empirically?**
> We use Adam and have now included this in our main text.

---

> > ### Comment · Reviewer_UHrP · 2024-12-03
> >
> > Thank you to the author for responding to my questions. I will maintain my positive score.

---

### Official Review · Reviewer_Cgph · 2024-11-04

**Soundness:** 3
**Presentation:** 3
**Contribution:** 2
**Rating:** 6
**Confidence:** 2

**Summary:**

This work proposes a perspective on pretraining loss landscape drawing a metaphor to a river valley. The authors run toy experiments in support of this persepective, and present theoretical analysis as well. The authors present WD-S, a variant of Warmup-Stable-Decay, which they favorably compare against WSD and cosine-derived learning rate schedules across a number of model sizes.

**Strengths:**

The paper is well-written, with clear prose and clean figures. The elucidation of the river-valley loss landscape is intriguing and intuitive. The empirical results are good, though they could cover a broader range of experiment setups. The work is of potential interest to model-training practitioners who may desire to train large models without pre-specifying a compute budget.

**Weaknesses:**

The related works is deferred to the appendix. This makes it difficult to contextualize the work presented. If space is a concern, please still present a truncated related work section and defer a more extended discussion to the appendix. Of course the optimization literature is large but the most relevant works need to be discussed in the main text to provide context.

The originality of the technical contribution of this extension of WDS is marginal, though the empirical results suggest it is a useful one.

The loss curves presented in Fig 9 are very spiky, and the field of view is highly zoomed in. These should be re-run with a larger number of training runs and averaged, otherwise the result seems plausibly artifactual. Same for Figures 10 and 11.

**Questions:**

Nit:
Fig 7 occludes the caption of Fig 8. Please fix. Fig 7 captions includes 'textbf'

---

> ### Author Response · Authors · 2024-11-27
>
> We thank the reviewer for finding our paper both intriguing and intuitive. We address the concerns below.
>
> **W1: Related work is deferred to the appendix. A concise related work should be put in the main text.**
> We thank the reviewer for the suggestion and have now included a concise related work section in the main text.
>
> **W2: The technical originality of WSD-S is marginal.**
> The main focus of our paper is the river valley theory, and WSD-S is a product of this theory. The effectiveness of WSD-S, as acknowledged by the reviewer, serves as evidence supporting our theory.
>
> **W3: Loss curves on Figures are too spiky.**
> To further validate the effectiveness of WSD-S, we reran our experiments on a new dataset, DCLM \[1\], using models of sizes 100M and 600M. Our results continue to hold without loss spikes. The updated results are presented in Appendix B of the revised draft. Additionally, we plan to replace the original Pile experiments with these new results in future versions.
>
> \[1\] DataComp-LM: In search of the next generation of training sets for language models
>
> **Q1: Formatting issues concerning Figure 7**
> We apologize for the error and have corrected it in the updated draft.

---

### Meta-Review · Area_Chair_TDCf · 2024-12-19

**Metareview:**

This paper proposes a theoretical and intuitional view of why the Warmup-Stable-Decay (WSD) learning rate schedule is so effective in model training.

Pros:
If these intuitions apply to real world models, they will greatly help reasoning about the training process. They present a useful explanation for how learning rate decay acts after a loss spike. Their theory has explanatory power for why learning rate schedules act as they do. They introduce a new learning raid schedule that seems to improve training speed in their small settings.

Cons:
The claimed connection to large language models or practical models is weak, in that the conceptual intuition might not apply. The empirical results are on very small models and unlikely to generalize to real world settings. The assumptions that lead to the river valley phenomenon are not necessarily guaranteed on large models, and are not tested. Furthermore, the proposed improved training schedule also might not work in realistic settings; it is unlikely that the contributions of this paper are practical.

 The literature review is still weak and requires better contextualization of their analysis in the wider background literature on loss landscapes and training dynamics.

**Additional Comments On Reviewer Discussion:**

bscX and H6GK have a valid objection to the weakness of the literature review. The authors have moved the related work section into the appendix, which is increasingly common for space limitations, but even as it is, the authors absolutely should expand it to better reflect the current state of understanding of the loss landscape. I did not consider this objection to be sufficient to reject the paper, however.

They also maintain an objection to the realism of the experiments, and whether the conclusions of the paper would scale. While I agree that the claims need to be checked in a larger model setting, the paper proposes a way to think about the loss surface which can be tested in any given setting. I think there is value to the proposal.

bscX had a variety of confounders that they felt the authors should address, and the authors ran additional experiments to control for them. They also ran additional experiments to study whether this conceptual model works for training runs that don't include loss spikes. They also added evaluations for downstream tasks. These generally have improved the paper.

---

### Decision · Program_Chairs · 2025-01-22

Accept (Poster)